# Collapsing Taylor Mode Automatic Differentiation

**Felix Dangel**[*]
Vector Institute
Toronto, Canada
fdangel@vectorinstitute.ai

**Tim Siebert**[*]
Humboldt-Universität zu Berlin and
Zuse Institute Berlin
Berlin, Germany
tim.siebert@hu-berlin.de

**Marius Zeinhofer**
ETH Zurich
Zurich, Switzerland,
marius.zeinhofer@sam.math.ethz.ch

**Andrea Walther**
Humboldt-Universität zu Berlin and
Zuse Institute Berlin
Berlin, Germany
andrea.walther@math.hu-berlin.de

## Abstract

Computing partial differential equation (PDE) operators via nested backpropagation is expensive, yet popular, and severely restricts their utility for scientific machine learning. Recent advances, like the forward Laplacian and randomizing Taylor mode automatic differentiation (AD), propose forward schemes to address this. We introduce an optimization technique for Taylor mode that "collapses" derivatives by rewriting the computational graph, and demonstrate how to apply it to general linear PDE operators, and randomized Taylor mode. The modifications simply require propagating a sum up the computational graph, which could—or should—be done by a machine learning compiler, without exposing complexity to users. We implement our collapsing procedure and evaluate it on popular PDE operators, confirming it accelerates Taylor mode and outperforms nested backpropagation.

## 1 Introduction

Using neural networks to learn functions constrained by physical laws is a popular trend in scientific machine learning [4, 16, 17, 19, 24, 25, 29]. Typically, the Physics is encoded through partial differential equations (PDEs) that the neural net must satisfy. The associated loss functions require evaluating differential operators w.r.t. the net's input, rather than weights. Evaluating these differential operators remains a computational challenge, especially if they contain high-order derivatives.

**Computing PDE operators.** Two important fields that build on PDE operators are variational Monte-Carlo (VMC) simulations and Physics-informed neural networks (PINNs). VMC employs neural networks as ansatz for the Schrödinger equation [4, 16, 24] and demands computing the net's Laplacian (the Hessian trace) for the Hamiltonian's kinetic term. PINNs represent PDE solutions as a neural net and train it by minimizing the residuals of the governing equations [19, 25]. For instance, Kolmogorov-type equations like the Fokker-Planck and Black-Scholes equation require weighted second-order derivatives on high-dimensional spatial domains [17, 28]. Other PINNs for elasticity problems use the biharmonic operator [7, 17, 27, 31], which contains fourth-order derivatives.

**Is backpropagation all we need?** Although nesting first-order automatic differentiation (AD) to compute high-order derivatives scales exponentially w.r.t. the degree in time and memory [27, §3.2],

39th Conference on Neural Information Processing Systems (NeurIPS 2025).

---

[*]Equal contribution

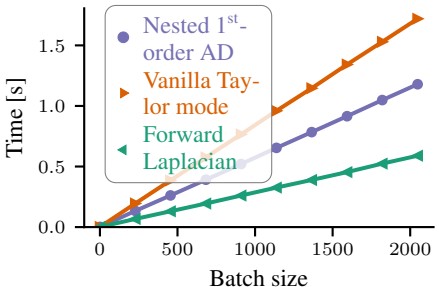

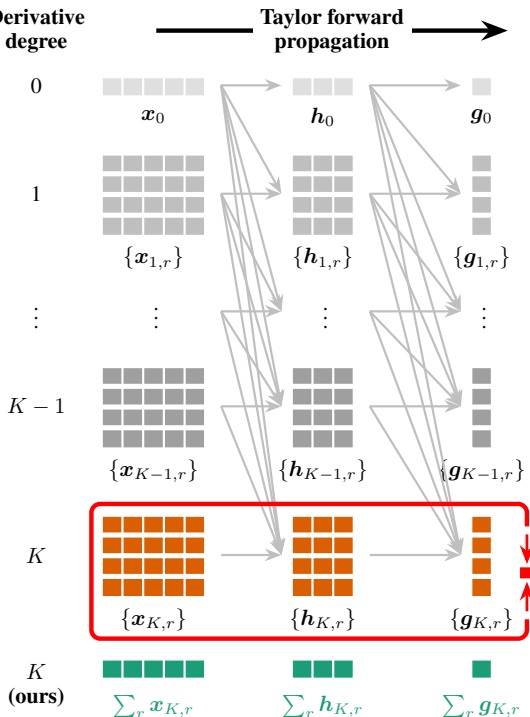

Figure 1: ▲ **Vanilla Taylor mode is not enough to beat nested 1st-order AD.** Illustrated for computing the Laplacian of a $\tanh$-activated $50 \to 768 \to 768 \to 512 \to 512 \to 1$ MLP with JAX (+ `jit`) on GPU (details in §G). We show how to automatically obtain the specialized forward Laplacian through simple graph transformations that "collapse" vanilla Taylor mode.

Figure 2: ▶ **Collapsed Taylor mode directly propagates the sum of highest degree coefficients.** Visualized for pushing 4 $K$-jets through a $\mathbb{R}^5 \to \mathbb{R}^3 \to \mathbb{R}$ function ($K = 2$ yields the forward Laplacian).

this approach is common practice: it is easy to implement in ML libraries, and their backpropagation is highly optimized. A promising alternative is *Taylor mode AD* [or simply *Taylor mode*, 13, §13], introduced to the ML community in 2019, which scales polynomially w.r.t. the degree in time and memory [14]. However, we observe empirically that vanilla Taylor mode may not be enough to beat nesting (fig. 1): evaluating the Laplacian of a 5-layer MLP, using JAX's *Taylor mode is 50% slower than nested backpropagation* that computes, then traces, the Hessian via Hessian-vector products [23]. This calls into question the relevance of Taylor mode for computing common PDE operators.

**The advent of forward schemes.** Recent works have successfully demonstrated the potential of modified forward propagation schemes, though. For the Laplacian, Li et al. [20, 21] developed a special forward propagation framework called the *forward Laplacian*, whose JAX implementation [12] is *roughly twice as fast as nested first-order AD* (fig. 1). While the forward Laplacian does not rely on Taylor mode, recent work pointed out a connection [6]; it remains unclear, though, if efficient forward schemes exist for other differential operators, and how they relate to Taylor mode. Concurrently, Shi et al. [27] derived stochastic approximations of differential operators in high dimensions by evaluating Taylor mode along suitably sampled random directions.

Irrespective of stochastic or exact computation, at their core, these popular PDE operators are *linear*: we must evaluate derivatives along multiple directions, then sum them. Based on this linearity, we identify an optimization technique to rewrite the computational graph of standard Taylor mode that is applicable to general linear PDE operators and randomized Taylor mode:

1. **We propose optimizing standard Taylor mode by collapsing the highest Taylor coefficients,** directly **propagating their sum**, rather than **propagating then summing** (fig. 2). Our approach contains the forward Laplacian as special case, is applicable to randomized Taylor mode, and also general linear PDE operators, which we show using the techniques from Griewank et al. [14].

2. **We show how to collapse standard Taylor mode by simple graph rewrites based on linearity.** This leads to a clean separation of concepts: Users can build their computational graph using standard Taylor mode, then rewrite it to collapse it. Due to the simple nature of our proposed rewrites, they could easily be absorbed into the just-in-time (JIT) compilation of ML frameworks without introducing a new interface or exposing complexity to users.

3. **We empirically demonstrate that collapsing Taylor mode accelerates standard Taylor mode.** We implement a Taylor mode library[2] for PyTorch [22] that realizes the graph simplifications with `torch.fx` [26]. On popular PDE operators, we empirically find that, compared to standard Taylor mode, collapsed Taylor mode achieves superior performance that is well-aligned with the theoretical expectation, while consistently outperforming nested first-order AD.

Our work takes an important step towards the broader adoption of Taylor mode as viable alternative to nested first-order AD for computing PDE operators, while being as easy to use.

## 2 Background: Introduction to Taylor Mode AD

Taylor mode AD (or, simply, Taylor mode) computes higher-order derivatives—as needed, e.g., for PDE operators—through propagation of Taylor coefficients according to the chain rule.

**Scalar case.** To illustrate Taylor mode, consider the scalar function $f : \mathbb{R} \to \mathbb{R}$ and extend the input variable $x$ to a path $x(t)$ with $x(0) = x_0$, whose form is a univariate Taylor polynomial of degree $K$, $x(t) = \sum_{k=0}^{K} \frac{t^k}{k!} x_k$ with $x_k$ the $k$-th Taylor coefficient. If $f$ is smooth enough, we can evaluate Taylor coefficients of the transformed path $f(x(t)) = \sum_{k=0}^{K} \frac{t^k}{k!} f_k$ with $f_k := \frac{\mathrm{d}^k}{\mathrm{d}t^k} f(x(t))|_{t=0}$. The chain rule provides the coefficients' propagation rules. E.g., for degree $K = 3$ we get

$$
\begin{aligned}
&f_0 = f(x_0)\,, &&f_2 = \partial^2 f(x_0) x_1^2 + \partial f(x_0) x_2\,, \\
&f_1 = \partial f(x_0) x_1\,, &&f_3 = \partial^3 f(x_0) x_1^3 + 3\partial^2 f(x_0) x_1 x_2 + \partial f(x_0) x_3\,.
\end{aligned}
\tag{1}
$$

Faà Di Bruno [9] provided the general formula for $f_k$, and Fraenkel [11] extended it to the multivariate case [see also 1, 15]. It serves as foundation for Taylor mode to compute higher-order derivatives [e.g., 13, §13]: setting $x_1 = 1, x_2 = x_3 = 0$ yields $f_1 = \partial f(x_0), f_2 = \partial^2 f(x_0), f_3 = \partial^3 f(x_0)$. We call the univariate Taylor polynomial of a function $x(t)$ of degree $K$, represented by the coefficients $(x_0, \dots, x_K)$, the $K$-*jet of* $x$, following the terminology of JAX's Taylor mode [2].

**Notation for multivariate case.** We consider the general case of computing higher-order derivatives, e.g., PDE operators, of a vector-to-vector function $\boldsymbol{f} : \mathbb{R}^D \to \mathbb{R}^C$. This requires additional notation to generalize eq. (1). Given $K$ vectors $\boldsymbol{v}_1, \dots, \boldsymbol{v}_K \in \mathbb{R}^D$, we write their tensor product as

$$
\otimes_{k=1}^{K} \boldsymbol{v}_k = \boldsymbol{v}_1 \otimes \dots \otimes \boldsymbol{v}_K \in (\mathbb{R}^D)^{\otimes K} \quad \text{with entries} \quad \left[\otimes_{k=1}^{K} \boldsymbol{v}_k\right]_{d_1, \dots, d_K} = [\boldsymbol{v}_1]_{d_1} \cdot \ldots \cdot [\boldsymbol{v}_K]_{d_K}
$$

for $d_1, \dots, d_K \in \{1, \dots, D\}$, and compactly write $\boldsymbol{v}^{\otimes K} = \otimes_{k=1}^{K} \boldsymbol{v}$. We define the inner product of two tensors $\mathbf{A}, \mathbf{B} \in (\mathbb{R}^D)^{\otimes K}$ as the Euclidean inner product of their flattened versions

$$
\langle \mathbf{A}, \mathbf{B} \rangle := \sum_{d_1} \sum_{d_2} \cdots \sum_{d_K} [\mathbf{A}]_{d_1, d_2, \dots, d_K} [\mathbf{B}]_{d_1, d_2, \dots, d_K} \in \mathbb{R}\,.
\tag{2}
$$

We allow broadcasting in eq. (2): if one tensor has more dimensions but matching trailing dimensions, we take the inner product for each component of the leading dimensions. This allows to express contractions with derivative tensors of vector-valued functions, e.g., contracting the $k$-th derivative tensor $\partial^k \boldsymbol{f}(\boldsymbol{x}_0) \in \mathbb{R}^C \times (\mathbb{R}^D)^{\otimes k}$, such that $\langle \mathbf{A}, \partial^k \boldsymbol{f}(\boldsymbol{x}_0) \rangle \in \mathbb{R}^C$.

**Multivariate case & composition.** Evaluating the $K$-jet of $\boldsymbol{f}$ at $\boldsymbol{x}_0 \in \mathbb{R}^D$ starts with the extension of $\boldsymbol{x}_0$ to a smooth path $\boldsymbol{x} : \mathbb{R} \to \mathbb{R}^D$ with $\boldsymbol{x}(0) = \boldsymbol{x}_0$. Formally, the $K$-jet of $\boldsymbol{f}$ is defined as

$$
J^K \boldsymbol{f} : \mathbb{R} \to \mathbb{R}^C\,, \quad (J^K \boldsymbol{f})(t) := \sum_{k=0}^{K} \frac{t^k}{k!} \boldsymbol{f}_k \quad \text{with} \quad \boldsymbol{f}_k := \frac{\mathrm{d}^k}{\mathrm{d}t^k} \boldsymbol{f}(\boldsymbol{x}(t))|_{t=0}
$$

and requires the $K$-jet of $\boldsymbol{x}$, $(J^K \boldsymbol{x})(t) := \sum_{k=0}^{K} \frac{t^k}{k!} \boldsymbol{x}_k$. As we are interested in the coefficients, we will slightly abuse the $K$-jet as mapping $(\boldsymbol{x}_0, \dots, \boldsymbol{x}_K) \mapsto (\boldsymbol{f}_0, \dots, \boldsymbol{f}_K)$ (see fig. 3 for an illustration).

As is common for AD, propagating the coefficients is broken down into composing $\boldsymbol{f}$ of atomic functions with known derivatives and the chain rule. In the simplest case, let $\boldsymbol{f} = \boldsymbol{g} \circ \boldsymbol{h} : \mathbb{R}^D \to$

---

[2]Available at https://github.com/f-dangel/torch-jet.

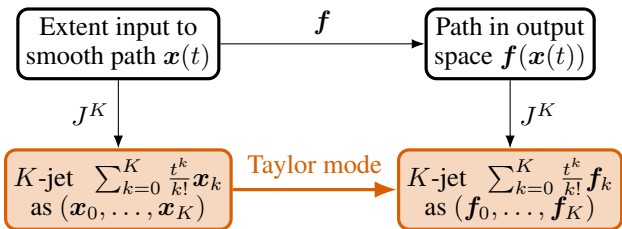

Figure 3: **Taylor mode propagates Taylor coefficients of a path in input space.** This results in the function-transformed path's Taylor coefficients. The Taylor expansion of degree $K$ is called a $K$-jet; hence Taylor mode propagates the input $K$-jet to the output $K$-jet.

$\mathbb{R}^I \to \mathbb{R}^C$ for two elemental functions $\boldsymbol{g}, \boldsymbol{h}$. Given the input $K$-jet for $\boldsymbol{x}$, the coefficients $\boldsymbol{h}_k = \frac{\mathrm{d}^k}{\mathrm{d}t^k}\boldsymbol{h}(\boldsymbol{x}(t))|_{t=0}$ follow from the generalized Faà di Bruno formula (spelled out for some $k$s in §A)

$$\boldsymbol{h}_k = \sum_{\sigma \in \mathrm{part}(k)} \nu(\sigma) \left\langle \partial^{|\sigma|}\boldsymbol{h}, \bigotimes_{s \in \sigma} \boldsymbol{x}_s \right\rangle \quad \text{with} \quad \nu(\sigma) = \frac{k!}{\left(\prod_{s \in \sigma} n_s!\right)\left(\prod_{s \in \sigma} s!\right)}. \tag{3}$$

Here, $\mathrm{part}(k)$ is the integer partitioning of $k$ (a set of sets), $\nu$ is a multiplicity function, and $n_s$ counts occurrences of $s$ in a set $\sigma$ (e.g., $n_1(\{1, 1, 3\}) = 2$ and $n_3 = 1$). Propagating the $\boldsymbol{h}_k$s through $\boldsymbol{g}$ results in the $K$-jet for $\boldsymbol{f}$. In summary, the propagation scheme is (with $\boldsymbol{x}_k \in \mathbb{R}^D$, $\boldsymbol{h}_k \in \mathbb{R}^I$, $\boldsymbol{f}_k \in \mathbb{R}^C$)

$$\begin{pmatrix} \boldsymbol{x}_0 \\ \boldsymbol{x}_1 \\ \boldsymbol{x}_2 \\ \vdots \\ \boldsymbol{x}_K \end{pmatrix} \overset{(3)}{\to} \begin{pmatrix} \boldsymbol{h}_0 = \boldsymbol{h}(\boldsymbol{x}_0) \\ \boldsymbol{h}_1 = \langle \partial \boldsymbol{h}(\boldsymbol{x}_0), \boldsymbol{x}_1 \rangle \\ \boldsymbol{h}_2 = \langle \partial^2 \boldsymbol{h}(\boldsymbol{x}_0), \boldsymbol{x}_1^{\otimes 2} \rangle + \langle \partial \boldsymbol{h}(\boldsymbol{x}_0), \boldsymbol{x}_2 \rangle \\ \vdots \\ \boldsymbol{h}_K = \sum_{\sigma \in \mathrm{part}(K)} \nu(\sigma) \left\langle \partial^{|\sigma|}\boldsymbol{h}(\boldsymbol{x}_0), \bigotimes_{s \in \sigma} \boldsymbol{x}_s \right\rangle \end{pmatrix}$$

$$\overset{(3)}{\to} \begin{pmatrix} \boldsymbol{g}_0 = \boldsymbol{g}(\boldsymbol{h}_0) \\ \boldsymbol{g}_1 = \langle \partial \boldsymbol{g}(\boldsymbol{h}_0), \boldsymbol{h}_1 \rangle \\ \boldsymbol{g}_2 = \langle \partial^2 \boldsymbol{g}(\boldsymbol{h}_0), \boldsymbol{h}_1^{\otimes 2} \rangle + \langle \partial \boldsymbol{g}(\boldsymbol{h}_0), \boldsymbol{h}_2 \rangle \\ \vdots \\ \boldsymbol{g}_K = \sum_{\sigma \in \mathrm{part}(K)} \nu(\sigma) \left\langle \partial^{|\sigma|}\boldsymbol{g}(\boldsymbol{h}_0), \bigotimes_{s \in \sigma} \boldsymbol{h}_s \right\rangle \end{pmatrix} \overset{(1)}{=} \begin{pmatrix} \boldsymbol{f}_0 = \boldsymbol{f}(\boldsymbol{x}_0) \\ \boldsymbol{f}_1 = \langle \partial \boldsymbol{f}(\boldsymbol{x}_0), \boldsymbol{x}_1 \rangle \\ \boldsymbol{f}_2 = \langle \partial^2 \boldsymbol{f}(\boldsymbol{x}_0), \boldsymbol{x}_1^{\otimes 2} \rangle + \langle \partial \boldsymbol{f}(\boldsymbol{x}_0), \boldsymbol{x}_2 \rangle \\ \vdots \\ \boldsymbol{f}_K = \sum_{\sigma \in \mathrm{part}(K)} \nu(\sigma) \left\langle \partial^{|\sigma|}\boldsymbol{f}(\boldsymbol{x}_0), \bigotimes_{s \in \sigma} \boldsymbol{x}_s \right\rangle \end{pmatrix} \tag{4}$$

which describes the forward propagation of a *single* $K$-jet. However, computing popular PDE operators requires propagating *multiple* $K$-jets in parallel, then summing their results. We propose to pull this accumulation inside Taylor mode's propagation scheme, thereby collapsing it.

## 3 Collapsing Taylor Mode AD

We now describe how to collapse the Taylor mode AD computation of popular linear PDE operators and their stochastic approximations proposed in [27], and provide a general recipe for computing and collapsing general linear differential operators by interpolation, using earlier work from Griewank et al. [14]. At its core, our procedure uses the linearity of the highest Taylor coefficient's propagation rule. It allows to collapse coefficients along multiple directions and directly **propagate their sum**, rather than **propagating then summing**, yielding substantial reductions in computational cost.

### 3.1 Exploiting Linearity to Collapse Taylor Mode AD

To derive our proposed method, we start with a sum of $K$th-order directional derivatives of the function $\boldsymbol{f}$ along $R$ directions $\{\boldsymbol{v}_r\}_{r=1}^R$, which is a common building block for all our PDE operators:

$$\sum_{r=1}^{R} \left\langle \partial^K \boldsymbol{f}(\boldsymbol{x}_0), \boldsymbol{v}_r^{\otimes K} \right\rangle \in \mathbb{R}^C \tag{5}$$

(e.g., the exact Laplacian uses $K = 2$, $R = \dim(\boldsymbol{x}_0) = D$, and the unit vectors $\boldsymbol{v}_r = \boldsymbol{e}_r \in \mathbb{R}^D$ as directions; see §3.2 below). Instead of nesting $K$ calls to $1^{\text{st}}$-order AD, we can use $K$-jets to calculate

each summand of eq. (5) with Taylor mode. In total, we need $R$ $K$-jets, and have to set the $r$-th jet's coefficients to $\boldsymbol{x}_{0,r} = \boldsymbol{x}_0, \boldsymbol{x}_{1,r} = \boldsymbol{v}_r$ and $\boldsymbol{x}_{2,r} = \ldots = \boldsymbol{x}_{K,r} = \boldsymbol{0}$ (eq. (D13) applies this to eq. (4)).

Standard Taylor mode propagates $1 + KR$ vectors through every node of the computational graph (the 0th component is shared across all jets, see fig. 2). This gives the output jets $\{\{\boldsymbol{f}_{k,r}\}_{k=1}^K\}_{r=1}^R$, from which we only select the highest-degree coefficients $\{\boldsymbol{f}_{K,r}\}_{r=1}^R$, then sum them to obtain eq. (5).

The approach we propose here exploits that the $K$-th derivative of $\boldsymbol{g} \circ \boldsymbol{h}$ is $\partial\boldsymbol{g}$ times the $K$-th derivative of $\boldsymbol{h}$ plus other lower-order terms in $\boldsymbol{h}$. Therefore, $\boldsymbol{g} \circ \boldsymbol{h}$ is linear in the $K$-th derivative of $\boldsymbol{h}$. Mathematically speaking, there is a special element in the set of integer partitions $\mathrm{part}(K)$, namely the trivial partition $\tilde{\sigma} = \{K\}$, which contributes the term $\nu(\tilde{\sigma}) \langle \partial\boldsymbol{g}(\boldsymbol{h}_0), \boldsymbol{h}_{K,r}\rangle$ to eq. (3). This is the only term that uses the input jet's highest coefficient $\boldsymbol{h}_{K,r}$, and its dependency is *linear*. Separating it in the highest coefficient's forward propagation, we get (using $\nu(\tilde{\sigma}) = 1$)

$$\sum_{r=1}^R \boldsymbol{f}_{K,r} = \sum_{r=1}^R \boldsymbol{g}_{K,r} = \sum_{r=1}^R \sum_{\sigma\in\mathrm{part}(K)\setminus\{\tilde{\sigma}\}} \nu(\sigma) \left\langle \partial^{|\sigma|}\boldsymbol{g}(\boldsymbol{h}_0), \underset{s\in\sigma}{\otimes} \boldsymbol{h}_{s,r}\right\rangle + \sum_{r=1}^R \langle \partial\boldsymbol{g}(\boldsymbol{h}_0), \boldsymbol{h}_{K,r}\rangle$$

and since the propagation rule is linear w.r.t. $\boldsymbol{h}_{K,r}$, we can pull the summation inside:

$$\sum_{r=1}^R \boldsymbol{g}_{K,r} = \sum_{r=1}^R \sum_{\sigma\in\mathrm{part}(K)\setminus\{\tilde{\sigma}\}} \nu(\sigma) \left\langle \partial^{|\sigma|}\boldsymbol{g}(\boldsymbol{h}_0), \underset{s\in\sigma}{\otimes} \boldsymbol{h}_{s,r}\right\rangle + \left\langle \partial\boldsymbol{g}(\boldsymbol{h}_0), \sum_{r=1}^R \boldsymbol{h}_{K,r}\right\rangle . \quad (6)$$

This is the key insight of our work: *The summed highest-degree output coefficients depend on the summed highest-degree input coefficients* (as well as all lower-degree coefficients). The reason is *linearity* in Faà di Bruno's formula. Hence, to compute the sum $\sum_r \boldsymbol{g}_{K,r}$ we can directly propagate the sum $\sum_r \boldsymbol{h}_{K,r}$, collapsing coefficients over all directions. We call this *collapsed Taylor mode AD*.

Collapsed Taylor mode propagates only $1 + (K-1)R + 1$ vectors through every node in the computational graph (see fig. 2 and eq. (D14) which applies this to eq. (3)). These savings of $R - 1$ coefficients are significant improvements over standard Taylor mode, as we show below. In the following, we discuss how to collapse the Taylor mode computation of various PDE operators.

## 3.2 Linear Second-order Operators

**Laplacian.** The Laplace operator plays a central role in Physics and engineering, including electrostatics, fluid dynamics, heat conduction, and quantum mechanics [10, 24]. It contains the Hessian trace of each element of a function, i.e., for $\boldsymbol{f}: \mathbb{R}^D \to \mathbb{R}^C$, it is

$$\underbrace{\Delta\boldsymbol{f}(\boldsymbol{x}_0)}_{\in\mathbb{R}^C} := \langle \partial^2\boldsymbol{f}(\boldsymbol{x}_0), \boldsymbol{I}_D\rangle \begin{cases} = \sum_{d=1}^D \langle \partial^2\boldsymbol{f}(\boldsymbol{x}_0), \boldsymbol{e}_d^{\otimes 2}\rangle & \text{(exact)} \\ \overset{[27]}{\approx} \dfrac{1}{S}\sum_{s=1}^S \langle \partial^2\boldsymbol{f}(\boldsymbol{x}_0), \boldsymbol{v}_s^{\otimes 2}\rangle & \text{(stochastic)} \end{cases} \quad (7a)$$

with the $d$-th standard basis vector $\boldsymbol{e}_d$ used for exact computation, and $S$ random vectors $\boldsymbol{v}_s$ drawn i.i.d. from a distribution with unit variance (e.g. Rademacher or standard Gaussian) for stochastic estimation. By pattern-matching eq. (7a) with eq. (5) we conclude that $K = 2$, and the following choices for computing the Laplacian with standard Taylor mode:

$$\begin{matrix} \{(\boldsymbol{x}_{0,d} = \boldsymbol{x}_0, & \boldsymbol{x}_{1,d} = \boldsymbol{e}_d, & \boldsymbol{x}_{2,d} = \boldsymbol{0})\}_{d=1}^D & 1 + D + D \text{ vectors} & \text{(exact)} \\ \{(\boldsymbol{x}_{0,s} = \boldsymbol{x}_0, & \boldsymbol{x}_{1,s} = \boldsymbol{v}_s, & \boldsymbol{x}_{2,s} = \boldsymbol{0})\}_{s=1}^S & 1 + S + S \text{ vectors} & \text{(stochastic)} \end{matrix} . \quad (7b)$$

Collapsing standard Taylor mode yields $1 + D + 1$ (exact) and $1 + S + 1$ (stochastic) vectors. In fact, the collapsed Taylor mode for the exact Laplacian is the forward Laplacian from Li et al. [21] (see eq. (D16) for detailed presentation of the forward propagation). Note how we can seamlessly also collapse the stochastic approximation over the sampled directions, which is currently not done.

**Weighted Laplacian.** A natural generalization of the Laplacian involves contracting with a positive semi-definite matrix $\boldsymbol{D} = \boldsymbol{\sigma}\boldsymbol{\sigma}^\top \in \mathbb{R}^{D\times D}$ rather than the identity. $\boldsymbol{D}$ may represent the diffusion tensor in Kolmogorov-type PDEs like the Fokker-Planck equation [17], and $\boldsymbol{\sigma}$ can depend on $\boldsymbol{x}_0$

[8]. The weighted Laplacian contains the weighted Hessian's trace $\text{Tr}(\boldsymbol{\sigma}\boldsymbol{\sigma}^\top \partial^2[\boldsymbol{f}]_c)$ for each output element $c$ of $\boldsymbol{f}$. If $\text{rank}(\boldsymbol{D}) = R$ and therefore $\boldsymbol{\sigma} = (\boldsymbol{s}_1, \ldots, \boldsymbol{s}_R) \in \mathbb{R}^{D \times R}$, it is

$$\underbrace{\Delta_{\boldsymbol{D}} \boldsymbol{f}(\boldsymbol{x}_0)}_{\in \mathbb{R}^C} := \langle \partial^2 \boldsymbol{f}(\boldsymbol{x}_0), \boldsymbol{D} \rangle \quad \begin{cases} = \sum_{r=1}^{R} \langle \partial^2 \boldsymbol{f}(\boldsymbol{x}_0), \boldsymbol{s}_r^{\otimes 2} \rangle & \text{(exact)} \\ \overset{[17]}{\approx} \dfrac{1}{S} \sum_{s=1}^{S} \langle \partial^2 \boldsymbol{f}(\boldsymbol{x}_0), (\boldsymbol{\sigma}\boldsymbol{v}_s)^{\otimes 2} \rangle & \text{(stochastic)} \end{cases} . \tag{8a}$$

Computing it requires evaluating the following 2-jets with standard Taylor mode:

$$\begin{array}{llll} \{(\boldsymbol{x}_{0,r} = \boldsymbol{x}_0, & \boldsymbol{x}_{1,r} = \boldsymbol{s}_r, & \boldsymbol{x}_{2,r} = \boldsymbol{0})\}_{r=1}^{R} & 1 + R + R \text{ vectors} & \text{(exact)} \\ \{(\boldsymbol{x}_{0,s} = \boldsymbol{x}_0, & \boldsymbol{x}_{1,s} = \boldsymbol{\sigma}\boldsymbol{v}_s, & \boldsymbol{x}_{2,s} = \boldsymbol{0})\}_{s=1}^{S} & 1 + S + S \text{ vectors} & \text{(stochastic)} \end{array} . \tag{8b}$$

Our collapsed Taylor mode uses $1 + R + 1$ (exact) and $1 + S + 1$ (stochastic) vectors. This yields the modified forward Laplacian from Li et al. [20]; collapsing the stochastic variant speeds up the Hutchinson trace estimator from Hu et al. [17]. For indefinite $\boldsymbol{D}$, we can simply apply this scheme to the positive and negative eigen-spaces (however, such weightings are not used in practise).

### 3.3 Collapsed Taylor Mode for Arbitrary Mixed Partial Derivatives

So far, we discussed operators that result from contracting the second-order derivative tensor with a coefficient matrix ($\boldsymbol{I}$ or $\boldsymbol{D}$) that can conveniently be written as sum of vector outer products. For orders higher than two, the coefficient tensor can in general *not* easily be decomposed as such. Hence, we extend our framework to also cover differential operators containing mixed-partial derivatives by evaluating a suitable family of jets using the interpolation result of Griewank et al. [14]. As illustrative example, we will use the biharmonic operator with a 4-dimensional coefficient tensor:

$$\underbrace{\Delta^2 \boldsymbol{f}(\boldsymbol{x}_0)}_{\in \mathbb{R}^C} \quad \begin{cases} = \sum_{d_1=1}^{D} \sum_{d_2=1}^{D} \langle \partial^4 \boldsymbol{f}(\boldsymbol{x}_0), \boldsymbol{e}_{d_1}^{\otimes 2} \otimes \boldsymbol{e}_{d_2}^{\otimes 2} \rangle & \text{(exact)} \\ \overset{[27]}{\approx} \dfrac{1}{3S} \sum_{s=1}^{S} \langle \partial^4 \boldsymbol{f}(\boldsymbol{x}_0), \boldsymbol{v}_s^{\otimes 4} \rangle & \text{(stochastic)} \end{cases} . \tag{9}$$

We can directly collapse the stochastic version: draw $S$ standard normal vectors $\boldsymbol{v}_1, \ldots, \boldsymbol{v}_S$ and propagate the coefficients $\{(\boldsymbol{x}_{0,s} = \boldsymbol{x}_0, \boldsymbol{x}_{1,s} = \boldsymbol{v}_s, \boldsymbol{x}_{2,s} = \boldsymbol{x}_{3,s} = \boldsymbol{x}_{4,s} = \boldsymbol{0})\}_{s=1}^{S}$. With standard Taylor mode, this uses $1 + 4S$ vectors; collapsed Taylor mode uses $1 + 3S + 1$ vectors. For the exact biharmonic operator, however, we need to develop an approach to compute mixed partials.

**General approach.** Assume we want to compute a linear differential operator of degree $K$. We can do so by contracting the $K$-th order derivative tensor $\partial^K \boldsymbol{f}(\boldsymbol{x}_0)$ with a coefficient tensor $\mathbf{C} \in (\mathbb{R}^D)^{\otimes K}$. We can always express this tensor in a tensor product basis, such that

$$\langle \partial^K \boldsymbol{f}(\boldsymbol{x}_0), \mathbf{C} \rangle = \sum_{d_1=1}^{D_1} \cdots \sum_{d_I=1}^{D_I} \langle \partial^K \boldsymbol{f}(\boldsymbol{x}_0), \boldsymbol{v}_{d_1}^{\otimes i_1} \otimes \ldots \otimes \boldsymbol{v}_{d_I}^{\otimes i_I} \rangle \in \mathbb{R}^C , \tag{10}$$

where the multi-index entries $\boldsymbol{i} = (i_1, \ldots, i_I)$ sum to $K$ and $D_j \leq D$. For the exact biharmonic operator (eq. (9)), we identify $K = 4, I = 2, \boldsymbol{i} = (2, 2), D_1 = D_2 = D, \boldsymbol{v}_{d_1} = \boldsymbol{e}_{d_1}$, and $\boldsymbol{v}_{d_2} = \boldsymbol{e}_{d_2}$. From the Faá di Bruno formula, we know that we can only compute terms of the form $\langle \partial^K \boldsymbol{f}(\boldsymbol{x}_0), \boldsymbol{v}^{\otimes K} \rangle$ with a $K$-jet. The challenge in eq. (10) is that it includes terms where *not* all directions coincide (e.g., for the biharmonic we have $I = 2$ different directions).

Fortunately, Griewank et al. [14] derived an approach to reconstruct such mixed-direction terms by linearly combining a *family* of $K$-jets that is determined by all vectors $\boldsymbol{j} \in \mathbb{N}^I$ whose entries sum to $K$, see fig. 4 for an illustration for the biharmonic (5 members). The $K$-jets along these directions are then combined with coefficients $\gamma_{\boldsymbol{i},\boldsymbol{j}} \in \mathbb{R}$, whose definition we provide in §E. In summary, we get

$$\langle \partial^K \boldsymbol{f}(\boldsymbol{x}_0), \boldsymbol{v}_{d_1}^{\otimes i_1} \otimes \ldots \otimes \boldsymbol{v}_{d_I}^{\otimes i_I} \rangle = \sum_{\boldsymbol{j} \in \mathbb{N}^I, \|\boldsymbol{j}\|_1 = K} \frac{\gamma_{\boldsymbol{i},\boldsymbol{j}}}{K!} \left\langle \partial^K \boldsymbol{f}(\boldsymbol{x}_0), \left( \sum_{i=1}^{I} \boldsymbol{v}_{d_i}[\boldsymbol{j}]_i \right)^{\otimes K} \right\rangle . \tag{11}$$

This construction allows us to rewrite eq. (10) as

$$\sum_{d_1=1}^{D_1} \cdots \sum_{d_I=1}^{D_I} \sum_{\boldsymbol{j} \in \mathbb{N}^I, \|\boldsymbol{j}\|_1 = K} \frac{\gamma_{\boldsymbol{i},\boldsymbol{j}}}{K!} \left\langle \partial^K \boldsymbol{f}(\boldsymbol{x}_0), \left(\sum_{i=1}^{I} \boldsymbol{v}_{d_i}[\boldsymbol{j}]_i\right)^{\otimes K} \right\rangle,$$

and—since the coefficients $\gamma_{\boldsymbol{i},\boldsymbol{j}}$ only depend on the problem structure ($K$, $I$ and $\boldsymbol{i}$) and *not* on the function $\boldsymbol{f}$ and the directions $\boldsymbol{v}_{d_i}$ [14]—we can pull out the inner sum to obtain the final expression

$$\sum_{\boldsymbol{j} \in \mathbb{N}^I, \|\boldsymbol{j}\|_1 = K} \frac{\gamma_{\boldsymbol{i},\boldsymbol{j}}}{K!} \sum_{d_1=1}^{D_1} \cdots \sum_{d_I=1}^{D_I} \left\langle \partial^K \boldsymbol{f}(\boldsymbol{x}_0), \left(\sum_{i=1}^{I} \boldsymbol{v}_{d_i}[\boldsymbol{j}]_i\right)^{\otimes K} \right\rangle. \tag{12}$$

We can evaluate eq. (12) with standard Taylor mode: For each $\boldsymbol{j}$, compute $\prod_{i=1}^{I} D_i$ $K$-jets with coefficients $\boldsymbol{x}_0, \boldsymbol{x}_1 = \sum_i \boldsymbol{v}_{d_i}[\boldsymbol{j}]_i, \boldsymbol{x}_2 = \ldots = \boldsymbol{x}_K = \boldsymbol{0}$. The sums from the tensor basis expansion can be collapsed with our proposed optimization, removing $\prod_{i=1}^{I} D_i$ vectors from the propagation for each $\boldsymbol{j}$. After repeating for each member $\boldsymbol{j}$ of the interpolation family, we form the linear combination using the $\gamma_{\boldsymbol{i},\boldsymbol{j}}$s, which yields the desired differential operator. We can often exploit symmetries in the $\gamma_{\boldsymbol{i},\boldsymbol{j}}$s and basis vectors to further reduce the number of $K$-jets (see §E.1 for a complete example).

**Applied to the biharmonic operator.**  Let us now illustrate the key steps of applying eq. (12) to the exact biharmonic operator eq. (9) (full procedure in §E.1). Figure 4 illustrates the 5 multi-indices $\boldsymbol{j}$ characterizing the 4-jets we need to interpolate $\langle \partial^4 \boldsymbol{f}(\boldsymbol{x}_0), \boldsymbol{e}_{d_1}^{\otimes 2} \otimes \boldsymbol{e}_{d_2}^{\otimes 2} \rangle$, and their coefficients $\gamma_{\boldsymbol{i},\boldsymbol{j}}$. Their definition, see eq. (E17), shows the equality of $\gamma_{\boldsymbol{i},\boldsymbol{j}}$ for $\boldsymbol{j} = (4,0)$ and $\boldsymbol{j} = (0,4)$, as well as $\boldsymbol{j} = (3,1)$ and $\boldsymbol{j} = (1,3)$. Exploiting those symmetries reduces the number of interpolation terms from 5 to 3 (eq. (E19)), corresponding to $D + D^2 + D^2$ 4-jets. Removing doubly-computed terms brings down the number of 4-jets to $D + D(D-1) + \tfrac{1}{2}D(D-1)$ (eq. (E21)). Translated to vectors, standard Taylor mode propagates $1 + 4D + 4D(D-1) + \tfrac{4}{2}D(D-1) = 6D^2 - 2D + 1$ vectors. After collapsing, we get $1 + 3D + 1 + 3D(D-1) + 1 + \tfrac{3}{2}D(D-1) + 1 = \tfrac{9}{2}D^2 - \tfrac{3}{2}D + 4$ vectors. This demonstrates the relevance of collapsing: it achieves a 25 % reduction in the quadratic coefficient.

**Summary & relation to other approaches for computing mixed partials.**  The scheme we propose based on Griewank et al. [14]'s interpolation result allows to calculate *general* linear differential operators beyond Laplacians, and is amenable to collapsing. Admittedly, eq. (12) seems daunting at first glance. However, it (i) offers a one-fits-all recipe to construct schemes for general linear PDE operators, and (ii) does not use jets of order $K' > K$ to compute $K$-th order derivatives. It is possible to derive more "pedagogical" approaches, which however require hand-crafted interpolation rules case by case, and propagation of higher-order jets which is costly (see §E.2 for a pedagogical example using less efficient 6-jets to compute the biharmonic operator, or [27, §F] for other operators).

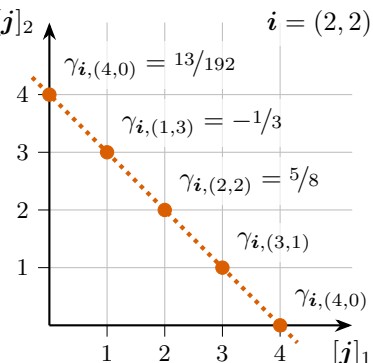

Figure 4: **Illustration of eq. (12) for the biharmonic operator**, i.e., the 5 values of $\boldsymbol{j}$ with $\|\boldsymbol{j}\|_1 = 4$ and their coefficients $\gamma_{\boldsymbol{i},\boldsymbol{j}}$ to interpolate the desired mixed partials.

## 4   Implementation & Experiments

Here, we describe our implementation of the Taylor mode collapsing process and empirically validate its performance improvements on the previously discussed operators.

**Design decisions & limitations.**  JAX [3] already offers an—albeit experimental—Taylor mode implementation [2]. However, we found it challenging to capture the computation graph and modify it using JAX's public interface. In contrast, PyTorch [22] provides `torch.fx` [26], which offers a user-friendly interface to capture and transform computational graphs purely in Python. Hence, we re-implemented Taylor mode in PyTorch, taking heavy inspiration from the JAX implementation.

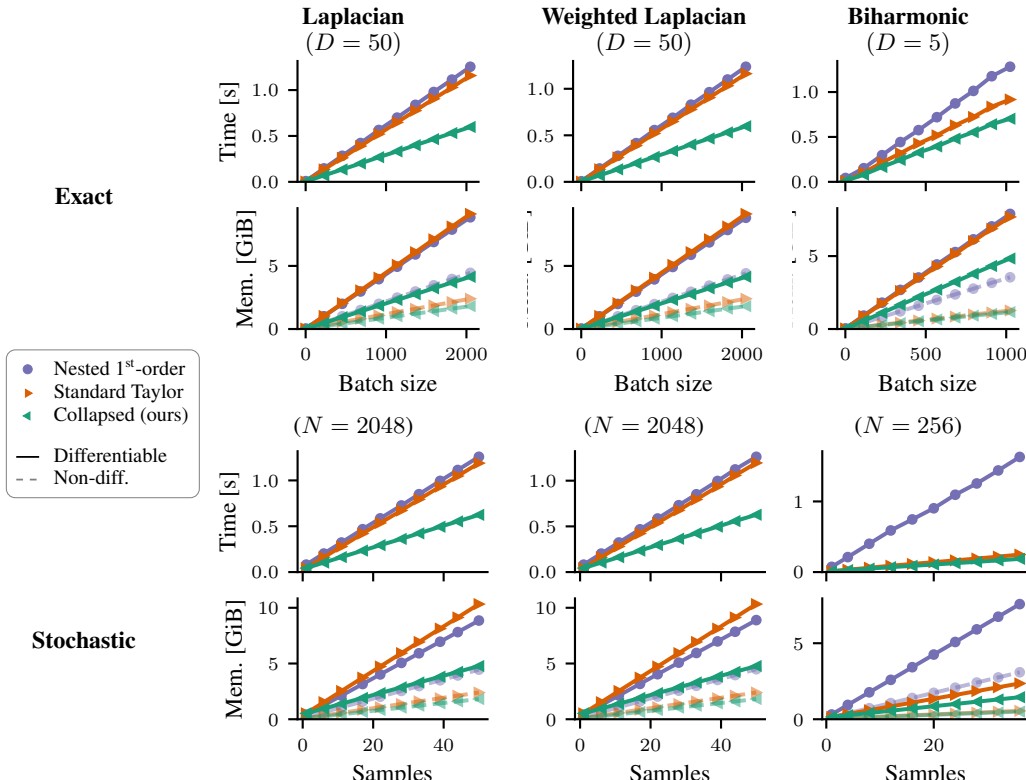

Figure 5: **Collapsed Taylor mode** accelerates **standard Taylor mode** and outperforms **nested 1st-order AD**. Exact computation varies the batch size, stochastic computation fixes a batch size and varies the samples such that $S < D$ (Laplacians), and $2 + 3S < \frac{9}{2}D^2 - \frac{3}{2}D + 4$ (biharmonic operator); we could compute exactly otherwise. Opaque markers are non-differentiable computations.

This deliberate choice imposes certain limitations. First, as of now, our Taylor mode in PyTorch supports only a small number of primitives, because the Taylor arithmetic in eq. (3) needs to be implemented case by case (this of course also applies to JAX's Taylor mode, which has broader operator coverage). Second, while our Taylor mode implementation is competitive with JAX's, we did not fully optimize it (e.g., we do *not* use in-place operations, and we do *not* implement the efficient schemes from Griewank & Walther [13, §13], but stick to Faà di Bruno (eq. (3))). Given our implementation's superiority compared to nested first-order AD that we demonstrate below, these are promising future efforts that will further improve performance, and we believe that making Taylor mode available to the PyTorch community is also an important step towards establishing its use.

**Usage (overview in §B).** Our implementation takes a PyTorch function (e.g., a neural net) and first captures its computational graph using `torch.fx`'s symbolic tracing mechanism. Then, it replaces each operation with its Taylor arithmetic, which yields the computational graph of the function's $K$-jet. Users can then write a function to compute their differential operator with this vanilla Taylor mode. Collapsing is achieved using a function `simplify`, which traces the computation again, rewrites the graph, and propagates the summation of highest coefficients up to its leafs. This requires one backward traversal through the graph (§C presents a detailed example). The simplified graph produces the same result, but propagates summed coefficients, i.e., uses collapsed Taylor mode.

**Experimental setup.** We empirically validate our proposed collapsing approach in PyTorch. We compare standard Taylor mode with collapsed Taylor mode and nested 1st-order AD on an Nvidia RTX 6000 GPU with 24 GiB memory. To implement the (weighted) Laplacian and its stochastic counterpart, we use vector-Hessian-vector products (VHVPs) in forward-over-reverse order, as recommended [5, 13]. For the biharmonic operator, we simply nest two VHVPs. For the weighted Laplacian's coefficient matrix, we choose a full-rank diagonal matrix (§F.2 shows results for rank-

Table 1: **Benchmark from fig. 5 in numbers.** We fit linear functions and report their slopes, i.e., how much runtime and memory increase when incrementing the batch size or random samples. We show two significant digits and bold values are best according to parenthesized values.

| Mode | Per-datum or -sample cost | Implementation | Laplacian | Weighted Laplacian | Biharmonic |
|---|---|---|---|---|---|
| **Exact** | Time [ms] | Nested 1$^{st}$-order | 0.61 (1.0x) | 0.60 (1.0x) | 1.2 (1.0x) |
| | | Standard Taylor | 0.56 (0.93x) | 0.57 (0.94x) | 0.90 (0.72x) |
| | | Collapsed (ours) | **0.29 (0.48x)** | **0.29 (0.48x)** | **0.69 (0.55x)** |
| | Mem. [MiB] (differentiable) | Nested 1$^{st}$-order | 4.4 (1.0x) | 4.4 (1.0x) | 7.9 (1.0x) |
| | | Standard Taylor | 4.6 (1.0x) | 4.6 (1.0x) | 7.7 (0.98x) |
| | | Collapsed (ours) | **2.1 (0.47x)** | **2.1 (0.47x)** | **4.8 (0.61x)** |
| | Mem. [MiB] (non-diff.) | Nested 1$^{st}$-order | 2.2 (1.0x) | 2.2 (1.0x) | 3.5 (1.0x) |
| | | Standard Taylor | 1.2 (0.54x) | 1.2 (0.54x) | 1.2 (0.36x) |
| | | Collapsed (ours) | **0.90 (0.41x)** | **0.90 (0.41x)** | **1.1 (0.33x)** |
| **Stochastic** | Time [ms] | Nested 1$^{st}$-order | 24 (1.0x) | 24 (1.0x) | 44 (1.0x) |
| | | Standard Taylor | 23 (0.97x) | 23 (0.97x) | 6.6 (0.15x) |
| | | Collapsed (ours) | **12 (0.49x)** | **12 (0.49x)** | **4.9 (0.11x)** |
| | Mem. [MiB] (differentiable) | Nested 1$^{st}$-order | 180 (1.0x) | 180 (1.0x) | 210 (1.0x) |
| | | Standard Taylor | 200 (1.2x) | 200 (1.2x) | 64 (0.30x) |
| | | Collapsed (ours) | **89 (0.50x)** | **89 (0.50x)** | **38 (0.18x)** |
| | Mem. [MiB] (non-diff.) | Nested 1$^{st}$-order | 89 (1.0x) | 90 (1.0x) | 86 (1.0x) |
| | | Standard Taylor | 48 (0.54x) | 48 (0.53x) | 15 (0.17x) |
| | | Collapsed (ours) | **36 (0.40x)** | **36 (0.40x)** | **13 (0.16x)** |

deficient weightings). To avoid confounding factors, all implementations are executed without compilation (our JAX experiments with the Laplacian in §G confirm that `jit` does not affect the relative performance). As common for PINNs [e.g., 6, 27], we use a 5-layer MLP $f_{\theta} : D \to 768 \to 768 \to 512 \to 512 \to 1$ with $\tanh$ activations and trainable parameters $\theta$, and compute the PDE operators on batches of size $N$. We measure three performance metrics: **(1) runtime** reports the smallest execution time of 50 repetitions. **(2) Peak memory (non-differentiable)** measures the maximum allocated GPU memory when computing the PDE operator's value (e.g., used in VMC [24]) inside a `torch.no_grad` context. **(3) Peak memory (differentiable)** is the maximum memory usage when computing the PDE operator inside a `torch.enable_grad` context, which allows backpropagation to $\theta$ (required for training PINNs, or alternative VMC works [30, 32]). This demands saving intermediates, which uses more memory but does not affect runtime. As memory allocation does not fluctuate much, we measure it in a single run.

**Results.** Figure 5 visualizes the growth in computational resources w.r.t. the batch size (exact) and random samples (stochastic) for fixed dimensions $D$. Runtime and memory increase linearly in both, as expected. We quantify the results by fitting linear functions and reporting their slopes (i.e., time and memory added per datum/sample) in table 1. We make the following observations:

- **Collapsed Taylor mode accelerates standard Taylor mode.** The measured performance differences correspond well with the theoretical estimate from counting the number of forward-propagated vectors. E.g., for the exact Laplacian, adding one datum introduces $2 + D$ versus $1 + 2D$ new vectors. For $D = 50$, their ratio is $(2+D)/(1+2D) \approx 0.51$. Empirically, we measure that adding one datum adds 0.56 ms to standard, and 0.29 ms to collapsed, Taylor mode (table 1); the ratio of $\approx 0.52$ is close. Similar arguments hold for peak memory of differentiable computation, stochastic approximation, and the other PDE operators (see table F2 for all comparisons).

- **Collapsed Taylor mode outperforms nested 1$^{st}$-order AD.** For the exact and stochastic (weighted) Laplacians, collapsed Taylor mode is roughly twice as fast (consistent with the JAX results in fig. 1) while using only 40-50% memory. For the biharmonic operator, we also observe speed-ups; in the stochastic case up to 9x in time, and 5x in memory (differentiable).

**Comparison with JAX.** We also conducted experiments with JAX (+ `jit`) to rule out artifacts from choosing PyTorch, implementation mistakes in our Taylor mode library, or unexpected simplifications from the JIT compiler. We find that the choice of the ML framework does not affect the results. E.g., when computing the exact Laplacian with nested first-order AD, PyTorch consumes $0.61$ ms per datum (table 1), while JAX uses $0.58$ ms (fig. 1 and table G5). We find the same trend when comparing our collapsed Taylor mode and JAX's forward Laplacian. Interestingly, we noticed that JAX's Taylor mode was consistently slower than our PyTorch implementation, despite using `jit`. We hypothesize that this could stem from algorithmic differences in the Taylor mode implementations and conclude from these results that (both ours, as well as the existing JAX) Taylor mode still has potential for improvements that may further increase the margin to nested first-order.

## 5   Conclusion

Computing differential operators is a critical component in scientific machine learning, particularly for Physics-informed neural networks and variational Monte-Carlo. Our work introduces collapsed Taylor mode, a simple yet effective optimization based on linearity in Faà di Bruno's formula, that propagates the sum of highest-order Taylor coefficients, rather than propagating then summing. It contains recent advances in forward-mode schemes, recovering the forward Laplacian [21], while being applicable to stochastic Taylor mode [17, 27]. We demonstrated that collapsed Taylor mode is useful to compute general linear differential operators, leveraging Griewank et al. [14]'s interpolation formula. Empirically, we confirmed speed-ups and memory savings for computing (randomized) Laplacians and biharmonic operators after collapsing Taylor mode, in accordance with our theoretical analysis, and confirmed its superiority to nesting first-order automatic differentiation. As the optimizations are achieved through simple graph rewrites based on linearity, we believe they could be integrated into existing just-in-time compilers without requiring a new interface or burdening users.

Our work takes an important step towards making Taylor mode a practical alternative to nested first-order differentiation in scientific machine learning, while maintaining ease of use. Future work could focus on integrating these optimizations directly into ML compilers, broadening operator coverage of our PyTorch implementation, and exploring additional graph optimizations for AD.

## Acknowledgments and Disclosure of Funding

Resources used in preparing this research were provided, in part, by the Province of Ontario, the Government of Canada through CIFAR, and companies sponsoring the Vector Institute. The research was funded partly by the DFG under Germany's Excellence Strategy – The Berlin Mathematics Research Center MATH+ (EXC-2046/1, project ID:390685689). M.Z. acknowledges support from an ETH Postdoctoral Fellowship for the project "Reliable, Efficient, and Scalable Methods for Scientific Machine Learning".

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

# Collapsing Taylor Mode Automatic Differentiation (Supplementary Material)

# A   Faà Di Bruno Formula Cheat Sheet

To give some intuition on the Faà di Bruno formula, we illustrate eq. (4) for higher orders here:

$$x_0 \to \qquad h_0 = h(x_0) \to \qquad g_0 = g(h_0) = f_0 = f(x_0)$$

$$x_1 \to \qquad h_1 = \langle \partial h, x_1 \rangle \to \qquad g_1 = \langle \partial g, h_1 \rangle = f_1 = \langle \partial f, x_1 \rangle$$

$$x_2 \to \qquad h_2 = \begin{matrix} \langle \partial^2 h, x_1^{\otimes 2} \rangle \\ +\langle \partial h, x_2 \rangle \end{matrix} \to \qquad g_2 = \begin{matrix} \langle \partial^2 g, h_1^{\otimes 2} \rangle \\ +\langle \partial g, h_2 \rangle \end{matrix} = f_2 = \begin{matrix} \langle \partial^2 f, x_1^{\otimes 2} \rangle \\ +\langle \partial f, x_2 \rangle \end{matrix}$$

$$x_3 \to \quad h_3 = \begin{matrix} \langle \partial^3 h, x_1^{\otimes 3} \rangle \\ +3\langle \partial^2 h, x_1 \otimes x_2 \rangle \\ +\langle \partial h, x_3 \rangle \end{matrix} \to \quad g_3 = \begin{matrix} \langle \partial^3 g, h_1^{\otimes 3} \rangle \\ +3\langle \partial^2 g, h_1 \otimes h_2 \rangle \\ +\langle \partial g, h_3 \rangle \end{matrix} = f_3 = \begin{matrix} \langle \partial^3 f, x_1^{\otimes 3} \rangle \\ +3\langle \partial^2 f, x_1 \otimes x_2 \rangle \\ +\langle \partial f, x_3 \rangle \end{matrix}$$

$$x_4 \to \quad h_4 = \begin{matrix} \langle \partial^4 h, x_1^{\otimes 4} \rangle \\ +6\langle \partial^3 h, x_1^{\otimes 2} \otimes x_2 \rangle \\ +4\langle \partial^2 h, x_1 \otimes x_3 \rangle \\ +3\langle \partial^2 h, x_2^{\otimes 2} \rangle \\ +\langle \partial h, x_4 \rangle \end{matrix} \to \quad g_4 = \begin{matrix} \langle \partial^4 g, h_1^{\otimes 4} \rangle \\ +6\langle \partial^3 g, h_1^{\otimes 2} \otimes h_2 \rangle \\ +4\langle \partial^2 g, h_1 \otimes h_3 \rangle \\ +3\langle \partial^2 g, h_2^{\otimes 2} \rangle \\ +\langle \partial g, h_4 \rangle \end{matrix} = f_4 = \begin{matrix} \langle \partial^4 f, x_1^{\otimes 4} \rangle \\ +6\langle \partial^3 f, x_1^{\otimes 2} \otimes x_2 \rangle \\ +4\langle \partial^2 f, x_1 \otimes x_3 \rangle \\ +3\langle \partial^2 f, x_2^{\otimes 2} \rangle \\ +\langle \partial f, x_4 \rangle \end{matrix}$$

$$x_5 \to \quad h_5 = \begin{matrix} \langle \partial^5 h, x_1^{\otimes 5} \rangle \\ +10\langle \partial^4 h, x_1^{\otimes 3} \otimes x_2 \rangle \\ +10\langle \partial^3 h, x_1^{\otimes 2} \otimes x_3 \rangle \\ +15\langle \partial^3 h, x_1 \otimes x_2^{\otimes 2} \rangle \\ +5\langle \partial^2 h, x_1 \otimes x_4 \rangle \\ +10\langle \partial^2 h, x_2 \otimes x_3 \rangle \\ +\langle \partial h, x_5 \rangle \end{matrix} \to \quad g_5 = \begin{matrix} \langle \partial^5 g, h_1^{\otimes 5} \rangle \\ +10\langle \partial^4 g, h_1^{\otimes 3} \otimes h_2 \rangle \\ +10\langle \partial^3 g, h_1^{\otimes 2} \otimes h_3 \rangle \\ +15\langle \partial^3 g, h_1 \otimes h_2^{\otimes 2} \rangle \\ +5\langle \partial^2 g, h_1 \otimes h_4 \rangle \\ +10\langle \partial^2 g, h_2 \otimes h_3 \rangle \\ +\langle \partial g, h_5 \rangle \end{matrix} = f_5 = \begin{matrix} \langle \partial^5 f, x_1^{\otimes 5} \rangle \\ +10\langle \partial^4 f, x_1^{\otimes 3} \otimes x_2 \rangle \\ +10\langle \partial^3 f, x_1^{\otimes 2} \otimes x_3 \rangle \\ +15\langle \partial^3 f, x_1 \otimes x_2^{\otimes 2} \rangle \\ +5\langle \partial^2 f, x_1 \otimes x_4 \rangle \\ +10\langle \partial^2 f, x_2 \otimes x_3 \rangle \\ +\langle \partial f, x_5 \rangle \end{matrix}$$

$$x_6 \to \quad h_6 = \begin{matrix} \langle \partial^6 h, x_1^{\otimes 6} \rangle \\ +15\langle \partial^5 h, x_1^{\otimes 4} \otimes x_2 \rangle \\ +20\langle \partial^4 h, x_1^{\otimes 3} \otimes x_3 \rangle \\ +45\langle \partial^4 h, x_1^{\otimes 2} \otimes x_2^{\otimes 2} \rangle \\ +15\langle \partial^3 h, x_1^{\otimes 2} \otimes x_4 \rangle \\ +60\langle \partial^3 h, x_1 \otimes x_2 \otimes x_3 \rangle \\ +15\langle \partial^3 h, x_2^{\otimes 3} \rangle \\ +6\langle \partial^2 h, x_1 \otimes x_5 \rangle \\ +15\langle \partial^2 h, x_2 \otimes x_4 \rangle \\ +10\langle \partial^2 h, x_3^{\otimes 2} \rangle \\ +\langle \partial h, x_6 \rangle \end{matrix} \to \quad g_6 = \begin{matrix} \langle \partial^6 g, h_1^{\otimes 6} \rangle \\ +15\langle \partial^5 g, h_1^{\otimes 4} \otimes h_2 \rangle \\ +20\langle \partial^4 g, h_1^{\otimes 3} \otimes h_3 \rangle \\ +45\langle \partial^4 g, h_1^{\otimes 2} \otimes h_2^{\otimes 2} \rangle \\ +15\langle \partial^3 g, h_1^{\otimes 2} \otimes h_4 \rangle \\ +60\langle \partial^3 g, h_1 \otimes h_2 \otimes h_3 \rangle \\ +15\langle \partial^3 g, h_2^{\otimes 3} \rangle \\ +6\langle \partial^2 g, h_1 \otimes h_5 \rangle \\ +15\langle \partial^2 g, h_2 \otimes h_4 \rangle \\ +10\langle \partial^2 g, h_3^{\otimes 2} \rangle \\ +\langle \partial g, h_6 \rangle \end{matrix} = f_6 = \begin{matrix} \langle \partial^6 f, x_1^{\otimes 6} \rangle \\ +15\langle \partial^5 f, x_1^{\otimes 4} \otimes x_2 \rangle \\ +20\langle \partial^4 f, x_1^{\otimes 3} \otimes x_3 \rangle \\ +45\langle \partial^4 f, x_1^{\otimes 2} \otimes x_2^{\otimes 2} \rangle \\ +15\langle \partial^3 f, x_1^{\otimes 2} \otimes x_4 \rangle \\ +60\langle \partial^3 f, x_1 \otimes x_2 \otimes x_3 \rangle \\ +15\langle \partial^3 f, x_2^{\otimes 3} \rangle \\ +6\langle \partial^2 f, x_1 \otimes x_5 \rangle \\ +15\langle \partial^2 f, x_2 \otimes x_4 \rangle \\ +10\langle \partial^2 f, x_3^{\otimes 2} \rangle \\ +\langle \partial f, x_6 \rangle \end{matrix}$$

$$x_7 \to \quad h_7 = \begin{matrix} \langle \partial^7 h, x_1^{\otimes 7} \rangle \\ +21\langle \partial^6 h, x_1^{\otimes 5} \otimes x_2 \rangle \\ +35\langle \partial^5 h, x_1^{\otimes 4} \otimes x_3 \rangle \\ +105\langle \partial^5 h, x_1^{\otimes 3} \otimes x_2^{\otimes 2} \rangle \\ +35\langle \partial^4 h, x_1^{\otimes 3} \otimes x_4 \rangle \\ +210\langle \partial^4 h, x_1^{\otimes 2} \otimes x_2 \otimes x_3 \rangle \\ +105\langle \partial^4 h, x_1 \otimes x_2^{\otimes 3} \rangle \\ +21\langle \partial^3 h, x_1^{\otimes 2} \otimes x_5 \rangle \\ +105\langle \partial^3 h, x_1 \otimes x_2 \otimes x_4 \rangle \\ +70\langle \partial^3 h, x_1 \otimes x_3^{\otimes 2} \rangle \\ +105\langle \partial^3 h, x_2^{\otimes 2} \otimes x_3 \rangle \\ +7\langle \partial^2 h, x_1 \otimes x_6 \rangle \\ +21\langle \partial^2 h, x_2 \otimes x_5 \rangle \\ +35\langle \partial^2 h, x_3 \otimes x_4 \rangle \\ +\langle \partial h, x_7 \rangle \end{matrix} \to \quad g_7 = \begin{matrix} \langle \partial^7 g, h_1^{\otimes 7} \rangle \\ +21\langle \partial^6 g, h_1^{\otimes 5} \otimes h_2 \rangle \\ +35\langle \partial^5 g, h_1^{\otimes 4} \otimes h_3 \rangle \\ +105\langle \partial^5 g, h_1^{\otimes 3} \otimes h_2^{\otimes 2} \rangle \\ +35\langle \partial^4 g, h_1^{\otimes 3} \otimes h_4 \rangle \\ +210\langle \partial^4 g, h_1^{\otimes 2} \otimes h_2 \otimes h_3 \rangle \\ +105\langle \partial^4 g, h_1 \otimes h_2^{\otimes 3} \rangle \\ +21\langle \partial^3 g, h_1^{\otimes 2} \otimes h_5 \rangle \\ +105\langle \partial^3 g, h_1 \otimes h_2 \otimes h_4 \rangle \\ +70\langle \partial^3 g, h_1 \otimes h_3^{\otimes 2} \rangle \\ +105\langle \partial^3 g, h_2^{\otimes 2} \otimes h_3 \rangle \\ +7\langle \partial^2 g, h_1 \otimes h_6 \rangle \\ +21\langle \partial^2 g, h_2 \otimes h_5 \rangle \\ +35\langle \partial^2 g, h_3 \otimes h_4 \rangle \\ +\langle \partial g, h_7 \rangle \end{matrix} = f_7 = \begin{matrix} \langle \partial^7 f, x_1^{\otimes 7} \rangle \\ +21\langle \partial^6 f, x_1^{\otimes 5} \otimes x_2 \rangle \\ +35\langle \partial^5 f, x_1^{\otimes 4} \otimes x_3 \rangle \\ +105\langle \partial^5 f, x_1^{\otimes 3} \otimes x_2^{\otimes 2} \rangle \\ +35\langle \partial^4 f, x_1^{\otimes 3} \otimes x_4 \rangle \\ +210\langle \partial^4 f, x_1^{\otimes 2} \otimes x_2 \otimes x_3 \rangle \\ +105\langle \partial^4 f, x_1 \otimes x_2^{\otimes 3} \rangle \\ +21\langle \partial^3 f, x_1^{\otimes 2} \otimes x_5 \rangle \\ +105\langle \partial^3 f, x_1 \otimes x_2 \otimes x_4 \rangle \\ +70\langle \partial^3 f, x_1 \otimes x_3^{\otimes 2} \rangle \\ +105\langle \partial^3 f, x_2^{\otimes 2} \otimes x_3 \rangle \\ +7\langle \partial^2 f, x_1 \otimes x_6 \rangle \\ +21\langle \partial^2 f, x_2 \otimes x_5 \rangle \\ +35\langle \partial^2 f, x_3 \otimes x_4 \rangle \\ +\langle \partial f, x_7 \rangle \end{matrix}$$

$$x_8 \to \quad h_8 = \begin{matrix} \langle \partial^8 h, x_1^{\otimes 8} \rangle \\ +28\langle \partial^7 h, x_1^{\otimes 6} \otimes x_2 \rangle \\ +56\langle \partial^6 h, x_1^{\otimes 5} \otimes x_3 \rangle \\ +210\langle \partial^6 h, x_1^{\otimes 4} \otimes x_2^{\otimes 2} \rangle \\ +70\langle \partial^5 h, x_1^{\otimes 4} \otimes x_4 \rangle \\ +560\langle \partial^5 h, x_1^{\otimes 3} \otimes x_2 \otimes x_3 \rangle \\ +420\langle \partial^5 h, x_1^{\otimes 2} \otimes x_2^{\otimes 3} \rangle \\ +56\langle \partial^4 h, x_1^{\otimes 3} \otimes x_5 \rangle \\ +420\langle \partial^4 h, x_1^{\otimes 2} \otimes x_2 \otimes x_4 \rangle \\ +280\langle \partial^4 h, x_1^{\otimes 2} \otimes x_3^{\otimes 2} \rangle \\ +840\langle \partial^4 h, x_1 \otimes x_2^{\otimes 2} \otimes x_3 \rangle \\ +105\langle \partial^4 h, x_2^{\otimes 4} \rangle \\ +28\langle \partial^3 h, x_1^{\otimes 2} \otimes x_6 \rangle \\ +168\langle \partial^3 h, x_1 \otimes x_2 \otimes x_5 \rangle \\ +280\langle \partial^3 h, x_1 \otimes x_3 \otimes x_4 \rangle \\ +210\langle \partial^3 h, x_2^{\otimes 2} \otimes x_4 \rangle \\ +280\langle \partial^3 h, x_2 \otimes x_3^{\otimes 2} \rangle \\ +8\langle \partial^2 h, x_1 \otimes x_7 \rangle \\ +28\langle \partial^2 h, x_2 \otimes x_6 \rangle \\ +56\langle \partial^2 h, x_3 \otimes x_5 \rangle \\ +35\langle \partial^2 h, x_4^{\otimes 2} \rangle \\ +\langle \partial h, x_8 \rangle \end{matrix} \to \quad g_8 = \begin{matrix} \langle \partial^8 g, h_1^{\otimes 8} \rangle \\ +28\langle \partial^7 g, h_1^{\otimes 6} \otimes h_2 \rangle \\ +56\langle \partial^6 g, h_1^{\otimes 5} \otimes h_3 \rangle \\ +210\langle \partial^6 g, h_1^{\otimes 4} \otimes h_2^{\otimes 2} \rangle \\ +70\langle \partial^5 g, h_1^{\otimes 4} \otimes h_4 \rangle \\ +560\langle \partial^5 g, h_1^{\otimes 3} \otimes h_2 \otimes h_3 \rangle \\ +420\langle \partial^5 g, h_1^{\otimes 2} \otimes h_2^{\otimes 3} \rangle \\ +56\langle \partial^4 g, h_1^{\otimes 3} \otimes h_5 \rangle \\ +420\langle \partial^4 g, h_1^{\otimes 2} \otimes h_2 \otimes h_4 \rangle \\ +280\langle \partial^4 g, h_1^{\otimes 2} \otimes h_3^{\otimes 2} \rangle \\ +840\langle \partial^4 g, h_1 \otimes h_2^{\otimes 2} \otimes h_3 \rangle \\ +105\langle \partial^4 g, h_2^{\otimes 4} \rangle \\ +28\langle \partial^3 g, h_1^{\otimes 2} \otimes h_6 \rangle \\ +168\langle \partial^3 g, h_1 \otimes h_2 \otimes h_5 \rangle \\ +280\langle \partial^3 g, h_1 \otimes h_3 \otimes h_4 \rangle \\ +210\langle \partial^3 g, h_2^{\otimes 2} \otimes h_4 \rangle \\ +280\langle \partial^3 g, h_2 \otimes h_3^{\otimes 2} \rangle \\ +8\langle \partial^2 g, h_1 \otimes h_7 \rangle \\ +28\langle \partial^2 g, h_2 \otimes h_6 \rangle \\ +56\langle \partial^2 g, h_3 \otimes h_5 \rangle \\ +35\langle \partial^2 g, h_4^{\otimes 2} \rangle \\ +\langle \partial g, h_8 \rangle \end{matrix} = f_8 = \begin{matrix} \langle \partial^8 f, x_1^{\otimes 8} \rangle \\ +28\langle \partial^7 f, x_1^{\otimes 6} \otimes x_2 \rangle \\ +56\langle \partial^6 f, x_1^{\otimes 5} \otimes x_3 \rangle \\ +210\langle \partial^6 f, x_1^{\otimes 4} \otimes x_2^{\otimes 2} \rangle \\ +70\langle \partial^5 f, x_1^{\otimes 4} \otimes x_4 \rangle \\ +560\langle \partial^5 f, x_1^{\otimes 3} \otimes x_2 \otimes x_3 \rangle \\ +420\langle \partial^5 f, x_1^{\otimes 2} \otimes x_2^{\otimes 3} \rangle \\ +56\langle \partial^4 f, x_1^{\otimes 3} \otimes x_5 \rangle \\ +420\langle \partial^4 f, x_1^{\otimes 2} \otimes x_2 \otimes x_4 \rangle \\ +280\langle \partial^4 f, x_1^{\otimes 2} \otimes x_3^{\otimes 2} \rangle \\ +840\langle \partial^4 f, x_1 \otimes x_2^{\otimes 2} \otimes x_3 \rangle \\ +105\langle \partial^4 f, x_2^{\otimes 4} \rangle \\ +28\langle \partial^3 f, x_1^{\otimes 2} \otimes x_6 \rangle \\ +168\langle \partial^3 f, x_1 \otimes x_2 \otimes x_5 \rangle \\ +280\langle \partial^3 f, x_1 \otimes x_3 \otimes x_4 \rangle \\ +210\langle \partial^3 f, x_2^{\otimes 2} \otimes x_4 \rangle \\ +280\langle \partial^3 f, x_2 \otimes x_3^{\otimes 2} \rangle \\ +8\langle \partial^2 f, x_1 \otimes x_7 \rangle \\ +28\langle \partial^2 f, x_2 \otimes x_6 \rangle \\ +56\langle \partial^2 f, x_3 \otimes x_5 \rangle \\ +35\langle \partial^2 f, x_4^{\otimes 2} \rangle \\ +\langle \partial f, x_8 \rangle \end{matrix}$$

# B Visual Tour: From Function to Collapsed Taylor Mode

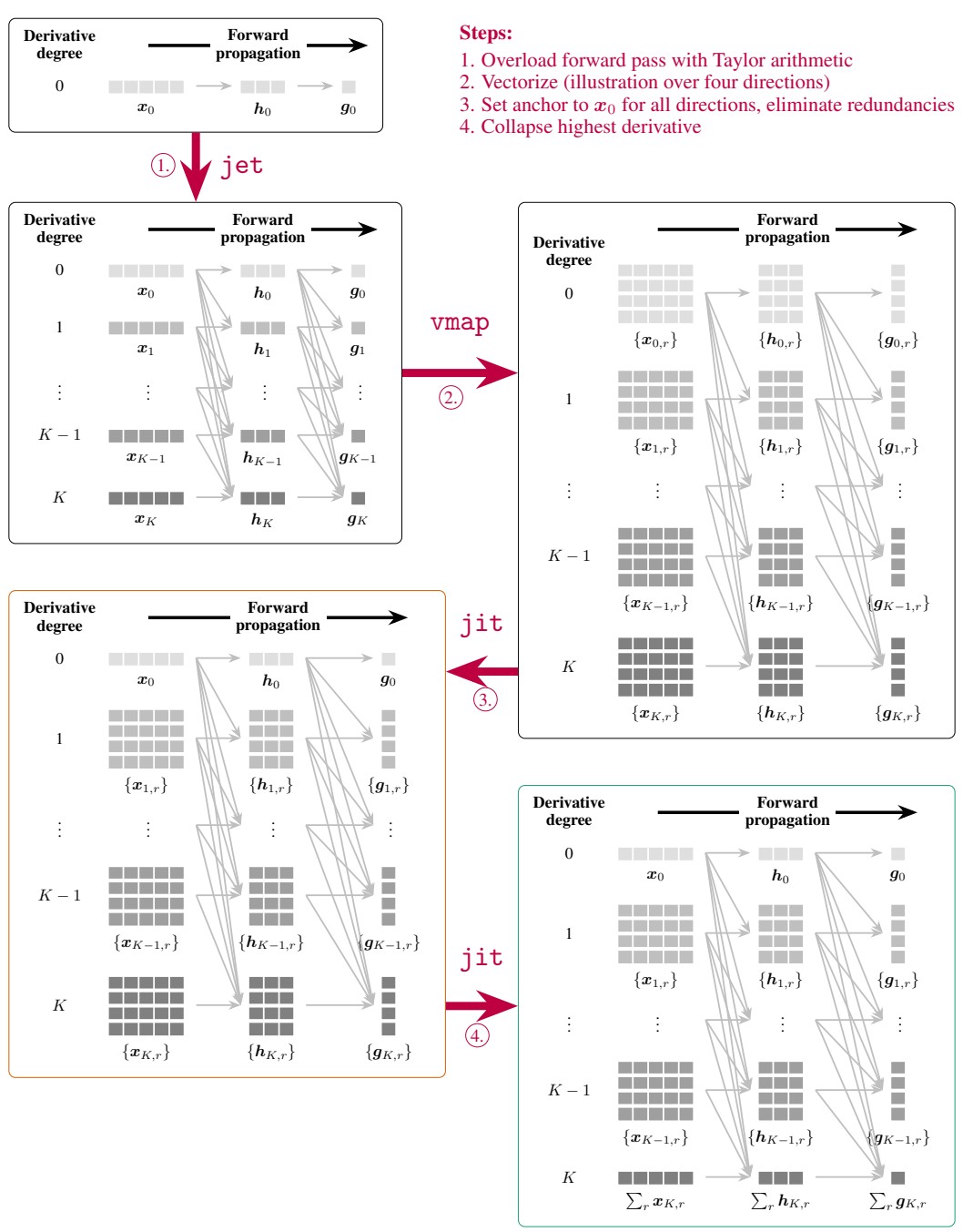

Figure B6: **Step-by-step transformation of a function into its collapsed Taylor mode.** Collapsed Taylor mode can be implemented via the already existing `jet` (standard Taylor mode), `vmap` (vectorization), and `jit` (graph transformation) interfaces. Therefore, users do not need to learn a new interface to benefit from our proposed method and can simply rely on standard Taylor mode and just-in-time compilation (after incorporating our proposed simplifications). Coloured boxes correspond to our PyTorch implementations of standard and collapsed Taylor mode from our experiments.

## C  Graph Simplifications

In this section, we illustrate the two graph simplifications that are required to collapse Taylor mode.

We will consider collapsing the 2-jet of $f = \sin$ as an example. Recall the propagation scheme eq. (D13) and assume that the Taylor coefficients are given by $\{\boldsymbol{x}_{0,r} = \boldsymbol{x}_0\}$, $\{\boldsymbol{x}_{1,r}\}$, and $\{\boldsymbol{x}_{2,r}\}$ where $r$ indexes the directions along which we evaluate the sum:

$$
\begin{matrix}
\boldsymbol{x}_0 \\
\{\boldsymbol{x}_{1,r}\} \\
\{\boldsymbol{x}_{2,r}\}
\end{matrix}
\overset{\text{replicate } \boldsymbol{x}_0}{\rightarrow}
\left\{
\begin{matrix}
\boldsymbol{x}_{0,r} = \boldsymbol{x}_0 \\
\boldsymbol{x}_{1,r} \\
\boldsymbol{x}_{2,r}
\end{matrix}
\right\}
\overset{\text{(D13)}}{\rightarrow}
\left\{
\begin{matrix}
\boldsymbol{f}_{0,r} = \sin(\boldsymbol{x}_0) \\
\boldsymbol{f}_{1,r} = \cos(\boldsymbol{x}_0) \odot \boldsymbol{x}_{1,r} \\
\boldsymbol{f}_{2,r} = -\sin(\boldsymbol{x}_0) \odot \boldsymbol{x}_{1,r} \odot \boldsymbol{x}_{1,r} + \cos(\boldsymbol{x}_0) \odot \boldsymbol{x}_{2,r}
\end{matrix}
\right\}
$$
$$
\overset{\text{sum highest component}}{\rightarrow}
\begin{matrix}
\left\{
\begin{matrix}
\boldsymbol{f}_{0,r} \\
\boldsymbol{f}_{1,r}
\end{matrix}
\right\} \\
\sum_r \boldsymbol{f}_{2,r}
\end{matrix}
$$

Here, $\sin$ applies element-wise and $\odot$ denotes element-wise multiplication. The computational graph for this procedure is displayed in the following diagram, with input and output nodes highlighted in dark and light gray. The suffix `_r` means that all $R$ corresponding tensors are stacked along their leading axis. `replicate` is a function that replicates a tensor $R$ times along a new leading axis, which is in PyTorch usually for free and without additional memory overhead (using `torch.expand`). All other functions refer to those of the PyTorch API:

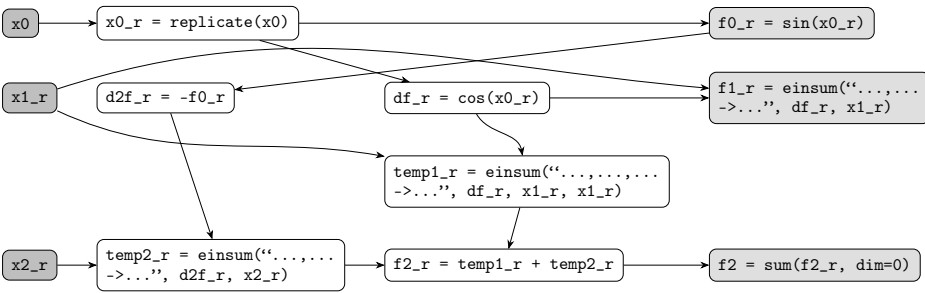

Our simplification proceeds in two steps. First, propagate `replicate` nodes down the graph to remove repeated computations on the same tensors. This is done in a forward traversal through the graph. Second, in a single backward traversal through the graph, we propagate the `sum` node up. After applying both steps, the graph looks as follows:

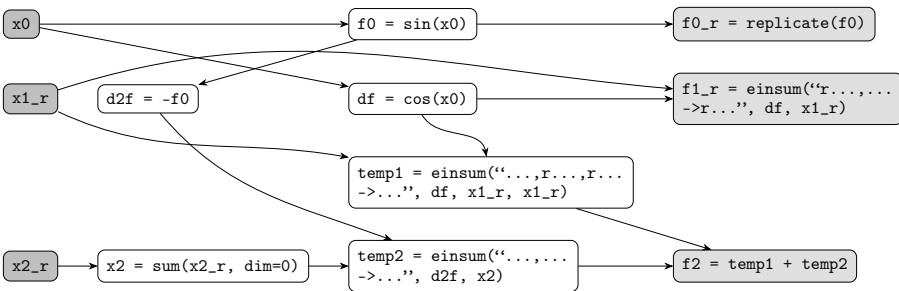

Two important properties of the new graph are (i) the `replicate` node moved to an output node, hence the corresponding redundant computation was successfully removed (ii) the highest component `x2_r` is immediately summed then propagated, i.e., we collapsed Taylor mode and avoid the separate propagation for all `x2_r`.

We will now illustrate the two simplification steps in full detail. The first stage starts from the original graph and pushes forward the replicate node, as illustrated step-by-step in fig. C7. The second stage starts from the graph produced by the replicate-push procedure, and propagates the final sum node up the graph, illustrated by fig. C8. This yields the final computation graph shown above.

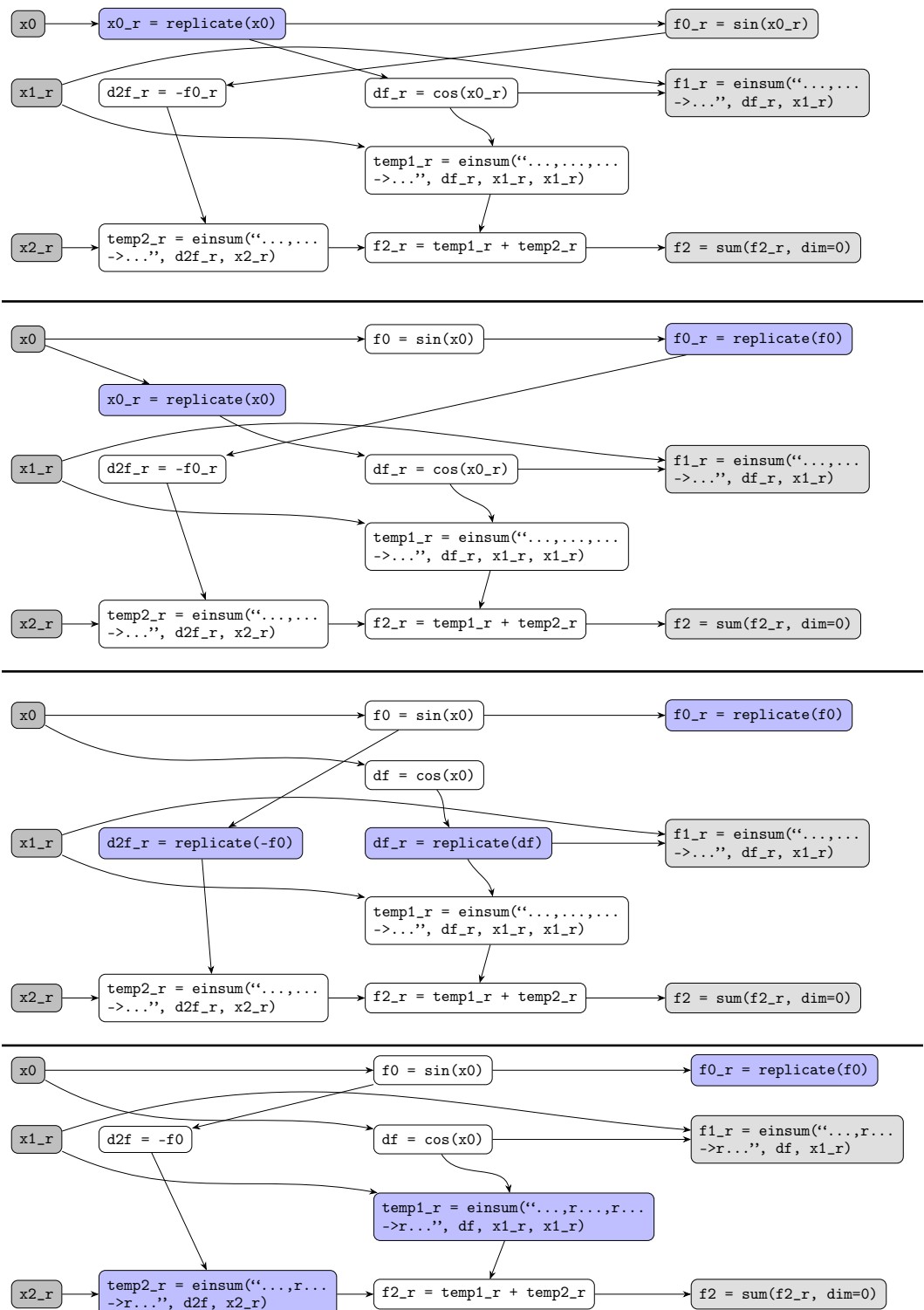

Figure C7: **Step-by-step illustration of pushing** `replicate` **nodes down a computation graph.**

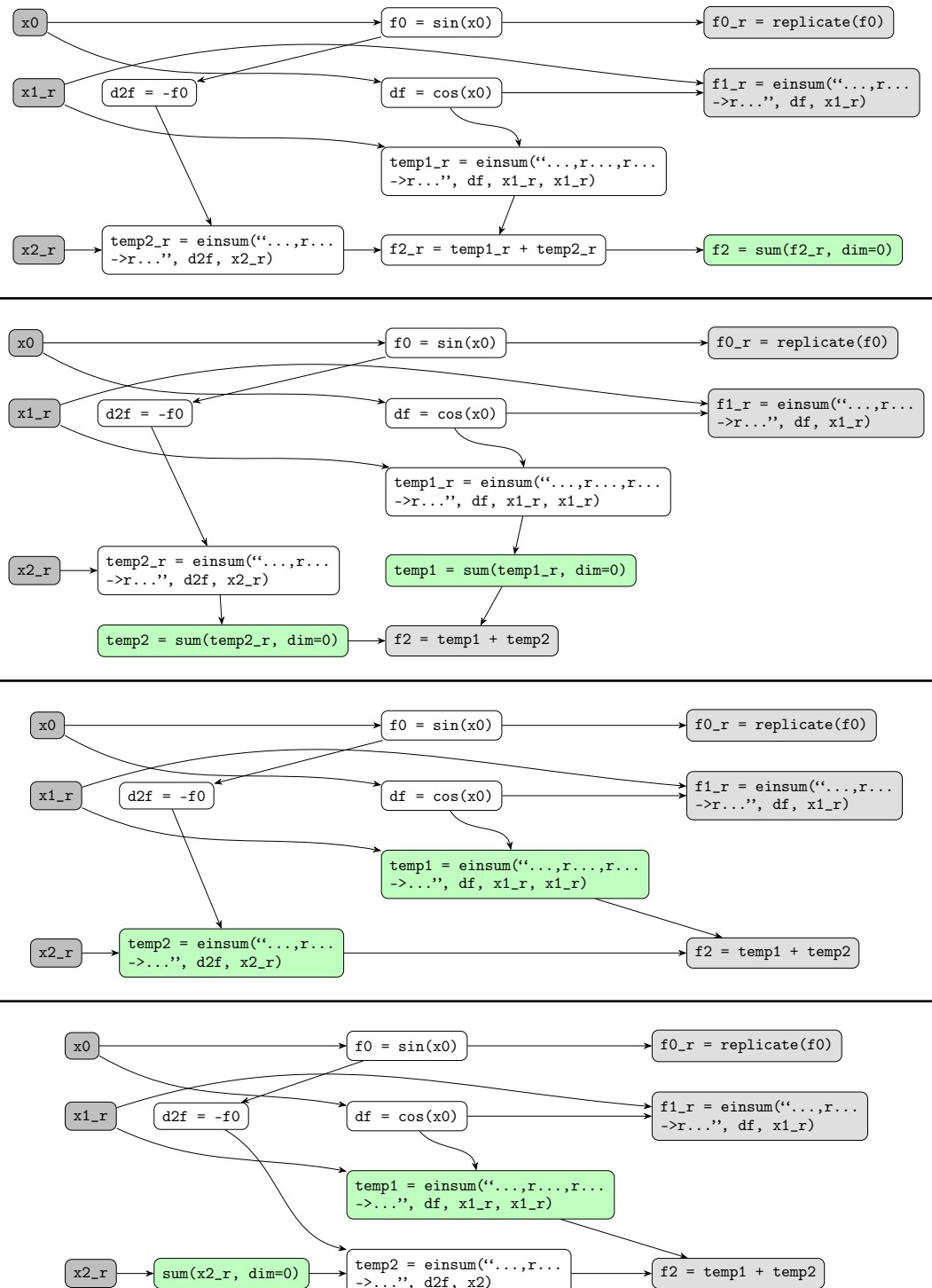

Figure C8: **Step-by-step illustration of propagating** sum **nodes up a computation graph.**

## D   Exploiting Linearity to Collapse Taylor Mode

Here, we illustrate the idea behind propagating $R$ $K$-jets through $\boldsymbol{f} = \boldsymbol{g} \circ \boldsymbol{h}$ with input jets $\left(J^K \boldsymbol{x}\right)_r (t) = \sum_{k=0}^{K} \frac{t^k}{k!} \boldsymbol{x}_{k,r}$. The Taylor mode scheme results from inserting eq. (5) into eq. (4):

$$
\begin{pmatrix} \boldsymbol{x}_0 \\ \{\boldsymbol{x}_{1,r}\} \\ \{\boldsymbol{x}_{2,r}\} \\ \vdots \\ \{\boldsymbol{x}_{K,r}\} \end{pmatrix} \xrightarrow{(3)} \begin{pmatrix} \boldsymbol{h}_0 = \boldsymbol{h}(\boldsymbol{x}_0) \\ \{\boldsymbol{h}_{1,r}\} = \{\langle \partial \boldsymbol{h}(\boldsymbol{x}_0), \boldsymbol{x}_{1,r}\rangle\} \\ \{\boldsymbol{h}_{2,r}\} = \{\langle \partial^2 \boldsymbol{h}(\boldsymbol{x}_0), \boldsymbol{x}_{1,r} \otimes \boldsymbol{x}_{1,r}\rangle + \langle \partial \boldsymbol{h}(\boldsymbol{x}_0), \boldsymbol{x}_{2,r}\rangle\} \\ \vdots \\ \{\boldsymbol{h}_{K,r}\} = \left\{ \sum_{\sigma \in \text{part}(K)} \nu(\sigma) \left\langle \partial^{|\sigma|} \boldsymbol{h}(\boldsymbol{x}_0), \underset{s \in \sigma}{\otimes} \boldsymbol{x}_{s,r}\right\rangle \right\} \end{pmatrix}
$$

$$
\xrightarrow{(3)} \begin{pmatrix} \boldsymbol{g}_0 = \boldsymbol{g}(\boldsymbol{h}_0) \\ \{\boldsymbol{g}_{1,r}\} = \{\langle \partial \boldsymbol{g}(\boldsymbol{h}_0), \boldsymbol{h}_{1,r}\rangle\} \\ \{\boldsymbol{g}_{2,r}\} = \{\langle \partial^2 \boldsymbol{g}(\boldsymbol{h}_0), \boldsymbol{h}_{1,r} \otimes \boldsymbol{h}_{1,r}\rangle + \langle \partial \boldsymbol{g}(\boldsymbol{h}_0), \boldsymbol{h}_{2,r}\rangle\} \\ \vdots \\ \{\boldsymbol{g}_{K,r}\} = \left\{ \sum_{\sigma \in \text{part}(K)} \nu(\sigma) \left\langle \partial^{|\sigma|} \boldsymbol{g}(\boldsymbol{h}_0), \underset{s \in \sigma}{\otimes} \boldsymbol{h}_{s,r}\right\rangle \right\} \end{pmatrix} \tag{D13}
$$

$$
\overset{(3)}{=} \begin{pmatrix} \boldsymbol{f}_0 = \boldsymbol{f}(\boldsymbol{x}_0) \\ \{\boldsymbol{f}_{1,r}\} = \{\langle \partial \boldsymbol{f}(\boldsymbol{x}_0), \boldsymbol{x}_{1,r}\rangle\} \\ \{\boldsymbol{f}_{2,r}\} = \{\langle \partial^2 \boldsymbol{f}(\boldsymbol{x}_0), \boldsymbol{x}_{1,r} \otimes \boldsymbol{x}_{1,r}\rangle + \langle \partial \boldsymbol{f}(\boldsymbol{x}_0), \boldsymbol{x}_{2,r}\rangle\} \\ \vdots \\ \{\boldsymbol{f}_{K,r}\} = \left\{ \sum_{\sigma \in \text{part}(K)} \nu(\sigma) \left\langle \partial^{|\sigma|} \boldsymbol{f}(\boldsymbol{x}_0), \underset{s \in \sigma}{\otimes} \boldsymbol{x}_{s,r}\right\rangle \right\} \end{pmatrix}
$$

$$
\xrightarrow{\text{slice}} \{\boldsymbol{g}_{K,r}\}
$$

$$
\xrightarrow{\text{sum}} \sum_{r=1}^{R} \boldsymbol{g}_{K,r} \overset{\{\boldsymbol{x}_{1,r}=\boldsymbol{v}_r, \boldsymbol{x}_{2,r}=\dots=\boldsymbol{x}_{K,r}=\boldsymbol{0}\}}{=} \sum_{r=1}^{R} \langle \partial^K \boldsymbol{f}(\boldsymbol{x}_0), \otimes_{k=1}^{K} \boldsymbol{v}_r \rangle
$$

Leveraging linearity in certain terms (in green) of the highest coefficient, as explained in eq. (6), instead leads to

$$
\begin{pmatrix} \boldsymbol{x}_0 \\ \{\boldsymbol{x}_{1,r}\} \\ \{\boldsymbol{x}_{2,r}\} \\ \vdots \\ \sum_{r=1}^{R} \boldsymbol{x}_{K,r} \end{pmatrix} \xrightarrow{(3)} \begin{pmatrix} \boldsymbol{h}_0 \quad = \boldsymbol{h}(\boldsymbol{x}_0) \\ \{\boldsymbol{h}_{1,r}\} \quad = \{\langle \partial \boldsymbol{h}(\boldsymbol{x}_0), \boldsymbol{x}_{1,r}\rangle\} \\ \{\boldsymbol{h}_{2,r}\} \quad = \{\langle \partial^2 \boldsymbol{h}(\boldsymbol{x}_0), \boldsymbol{x}_{1,r} \otimes \boldsymbol{x}_{1,r}\rangle + \langle \partial \boldsymbol{h}(\boldsymbol{x}_0), \boldsymbol{x}_{2,r}\rangle\} \\ \vdots \\ \sum_{r=1}^{R} \boldsymbol{h}_{K,r} \quad = \sum_{r=1}^{R} \sum_{\sigma \in \text{part}(K) \backslash \{\tilde{\sigma}\}} \nu(\sigma) \left\langle \partial^{|\sigma|} \boldsymbol{h}(\boldsymbol{x}_0), \underset{s \in \sigma}{\otimes} \boldsymbol{x}_{s,r}\right\rangle \\ \qquad\qquad + \left\langle \partial \boldsymbol{h}(\boldsymbol{x}_0), \sum_{r=1}^{R} \boldsymbol{x}_{K,r}\right\rangle \end{pmatrix}
$$

$$
\xrightarrow{(3)} \begin{pmatrix} \boldsymbol{g}_0 \quad = \boldsymbol{g}(\boldsymbol{h}_0) \\ \{\boldsymbol{g}_{1,r}\} \quad = \{\langle \partial \boldsymbol{g}(\boldsymbol{h}_0), \boldsymbol{h}_{1,r}\rangle\} \\ \{\boldsymbol{g}_{2,r}\} \quad = \{\langle \partial^2 \boldsymbol{g}(\boldsymbol{h}_0), \boldsymbol{h}_{1,r} \otimes \boldsymbol{h}_{1,r}\rangle + \langle \partial \boldsymbol{g}(\boldsymbol{h}_0), \boldsymbol{h}_{2,r}\rangle\} \\ \vdots \\ \sum_{r=1}^{R} \boldsymbol{g}_{K,r} \quad = \sum_{r=1}^{R} \sum_{\sigma \in \text{part}(K) \backslash \{\tilde{\sigma}\}} \nu(\sigma) \left\langle \partial^{|\sigma|} \boldsymbol{g}(\boldsymbol{h}_0), \underset{s \in \sigma}{\otimes} \boldsymbol{h}_{s,r}\right\rangle \\ \qquad\qquad + \left\langle \partial \boldsymbol{g}(\boldsymbol{h}_0), \sum_{r=1}^{R} \boldsymbol{h}_{K,r}\right\rangle \end{pmatrix} \tag{D14}
$$

$$\overset{(3)}{=} \begin{pmatrix} \boldsymbol{f}_0 & = \boldsymbol{f}(\boldsymbol{x}_0) \\ \{\boldsymbol{f}_{1,r}\} & = \{\langle \partial \boldsymbol{f}(\boldsymbol{x}_0), \boldsymbol{x}_{1,r}\rangle\} \\ \{\boldsymbol{f}_{2,r}\} & = \{\langle \partial^2 \boldsymbol{f}(\boldsymbol{x}_0), \boldsymbol{x}_{1,r} \otimes \boldsymbol{x}_{1,r}\rangle + \langle \partial \boldsymbol{f}(\boldsymbol{x}_0), \boldsymbol{x}_{2,r}\rangle\} \\ \vdots \\ \displaystyle\sum_{r=1}^{R} \boldsymbol{f}_{K,r} & = \displaystyle\sum_{r=1}^{R} \sum_{\sigma \in \mathrm{part}(K) \setminus \{\tilde\sigma\}} \nu(\sigma) \left\langle \partial^{|\sigma|} \boldsymbol{f}(\boldsymbol{x}_0), \underset{s\in\sigma}{\otimes} \boldsymbol{x}_{s,r}\right\rangle \\ & \quad + \left\langle \partial \boldsymbol{f}(\boldsymbol{x}_0), \displaystyle\sum_{r=1}^{R} \boldsymbol{x}_{K,r}\right\rangle \end{pmatrix}$$

$$\overset{\text{slice}}{\to} \sum_{r=1}^{R} \boldsymbol{g}_{K,r} \overset{\{\boldsymbol{x}_{1,r}=\boldsymbol{v}_r, \boldsymbol{x}_{2,r}=\ldots=\boldsymbol{x}_{K,r}=\boldsymbol{0}\}}{=} \sum_{r=1}^{R} \langle \partial^K \boldsymbol{f}(\boldsymbol{x}_0), \otimes_{k=1}^{K} \boldsymbol{v}_r\rangle$$

### D.1 Second-order Operators: Laplacian

Here, we show details about the propagation schemes of standard Taylor mode and collapsed Taylor mode for the computation of the Laplacian of $\boldsymbol{f}$. We consider the decomposition $\boldsymbol{f} = \boldsymbol{g} \circ \boldsymbol{h}$.

**Standard Taylor mode.** Using standard Taylor mode (eq. (D13)) to compute the Laplacian yields

$$\begin{pmatrix} \boldsymbol{x}_0 \\ \{\boldsymbol{x}_{1,d}\} \\ \{\boldsymbol{x}_{2,d}\} \end{pmatrix} \overset{(3)}{\to} \begin{pmatrix} \boldsymbol{h}_0 = \boldsymbol{h}(\boldsymbol{x}_0) \\ \{\boldsymbol{h}_{1,d}\} = \{\langle \partial \boldsymbol{h}(\boldsymbol{x}_0), \boldsymbol{x}_{1,d}\rangle\} \\ \{\boldsymbol{h}_{2,d}\} = \{\langle \partial^2 \boldsymbol{h}(\boldsymbol{x}_0), \boldsymbol{x}_{1,d} \otimes \boldsymbol{x}_{1,d}\rangle + \langle \partial \boldsymbol{h}(\boldsymbol{x}_0), \boldsymbol{x}_{2,d}\rangle\} \end{pmatrix}$$

$$\overset{(3)}{\to} \begin{pmatrix} \boldsymbol{g}_0 = \boldsymbol{g}(\boldsymbol{h}_0) \\ \{\boldsymbol{g}_{1,d}\} = \{\langle \partial \boldsymbol{g}(\boldsymbol{h}_0), \boldsymbol{h}_{1,d}\rangle\} \\ \{\boldsymbol{g}_{2,d}\} = \{\langle \partial^2 \boldsymbol{g}(\boldsymbol{h}_0), \boldsymbol{h}_{1,d} \otimes \boldsymbol{h}_{1,d}\rangle + \langle \partial \boldsymbol{g}(\boldsymbol{h}_0), \boldsymbol{h}_{2,d}\rangle\} \end{pmatrix}$$

$$\overset{(3)}{=} \begin{pmatrix} \boldsymbol{f}_0 = \boldsymbol{f}(\boldsymbol{x}_0) \\ \{\boldsymbol{f}_{1,d}\} = \{\langle \partial \boldsymbol{f}(\boldsymbol{x}_0), \boldsymbol{x}_{1,d}\rangle\} \\ \{\boldsymbol{f}_{2,d}\} = \{\langle \partial^2 \boldsymbol{f}(\boldsymbol{x}_0), \boldsymbol{x}_{1,d} \otimes \boldsymbol{x}_{1,d}\rangle + \langle \partial \boldsymbol{f}(\boldsymbol{x}_0), \boldsymbol{x}_{2,d}\rangle\} \end{pmatrix} \tag{D15}$$

$$\overset{\text{slice}}{\to} \{\boldsymbol{g}_{2,d}\}$$

$$\overset{\text{sum}}{\to} \sum_{d=1}^{D} \{\boldsymbol{g}_{2,d}\} \overset{\{\boldsymbol{x}_{1,d}=\boldsymbol{e}_d, \boldsymbol{x}_{2,d}=\boldsymbol{0}\}}{=} \Delta \boldsymbol{f}(\boldsymbol{x}_0).$$

**Collapsed Taylor Mode AD** Using our proposed collapsed Taylor mode, we get

$$\begin{pmatrix} \boldsymbol{x}_0 \\ \{\boldsymbol{x}_{1,d}\} \\ \displaystyle\sum_{d=1}^{D} \boldsymbol{x}_{2,d} \end{pmatrix} \overset{(1)}{\to} \begin{pmatrix} \boldsymbol{h}_0 = \boldsymbol{h}(\boldsymbol{x}_0) \\ \{\boldsymbol{h}_{1,d}\} = \{\langle \partial \boldsymbol{h}(\boldsymbol{x}_0), \boldsymbol{x}_{1,d}\rangle\} \\ \displaystyle\sum_{d=1}^{D} \boldsymbol{h}_{2,d} = \sum_{d=1}^{D} \langle \partial^2 \boldsymbol{h}(\boldsymbol{x}_0), \boldsymbol{x}_{1,d} \otimes \boldsymbol{x}_{1,d}\rangle + \left\langle \partial \boldsymbol{h}(\boldsymbol{x}_0), \displaystyle\sum_{d=1}^{D} \boldsymbol{x}_{2,d}\right\rangle \end{pmatrix}$$

$$\overset{(1)}{\to} \begin{pmatrix} \boldsymbol{g}_0 = \boldsymbol{g}(\boldsymbol{h}_0) \\ \{\boldsymbol{g}_{1,d}\} = \{\langle \partial \boldsymbol{g}(\boldsymbol{h}_0), \boldsymbol{h}_{1,d}\rangle\} \\ \displaystyle\sum_{d=1}^{D} \boldsymbol{g}_{2,d} = \sum_{d=1}^{D} \langle \partial^2 \boldsymbol{g}(\boldsymbol{h}_0), \boldsymbol{h}_{1,d} \otimes \boldsymbol{h}_{1,d}\rangle + \left\langle \partial \boldsymbol{g}(\boldsymbol{h}_0), \displaystyle\sum_{d=1}^{D} \boldsymbol{h}_{2,d}\right\rangle \end{pmatrix} \tag{D16}$$

$$\overset{(1)}{=} \begin{pmatrix} \boldsymbol{f}_0 = \boldsymbol{f}(\boldsymbol{x}_0) \\ \{\boldsymbol{f}_{1,d}\} = \{\langle \partial \boldsymbol{f}(\boldsymbol{x}_0), \boldsymbol{x}_{1,d}\rangle\} \\ \displaystyle\sum_{d=1}^{D} \boldsymbol{f}_{2,d} = \sum_{d=1}^{D} \langle \partial^2 \boldsymbol{f}(\boldsymbol{x}_0), \boldsymbol{x}_{1,d} \otimes \boldsymbol{x}_{1,d}\rangle + \left\langle \partial \boldsymbol{f}(\boldsymbol{x}_0), \displaystyle\sum_{i=1}^{D} \boldsymbol{x}_{2,d}\right\rangle \end{pmatrix}$$

$$\overset{\text{slice}}{\to} \sum_{d=1}^{D} \{\boldsymbol{g}_{2,d}\} \overset{\{(\boldsymbol{x}_{1,d}=\boldsymbol{e}_d, \boldsymbol{x}_{2,d}=\boldsymbol{0})\}}{=} \Delta \boldsymbol{f}(\boldsymbol{x}_0)$$

# E (Collapsed) Taylor Mode for Arbitrary Mixed Partial Derivatives

Here, we introduce the notation of eq. (11). The right side of the formula sums over the set of all $j \in \mathbb{N}^I$ such that $\|j\|_1 := \sum_i [j]_i = K$. For example, if $I = 2$ and $\|j\|_1 = 4$, this set is $\{(4,0),(0,4),(3,1),(1,3),(2,2)\}$.

The coefficient $\gamma_{i,j}$ is defined as

$$\gamma_{i,j} := \sum_{0 < m \leq i} (-1)^{\|i-m\|_1} \binom{i}{m} \binom{\|i\|_1 \frac{m}{\|m\|_1}}{j} \left(\frac{\|m\|_1}{\|i\|_1}\right)^{\|i\|_1}. \tag{E17}$$

The summation ranges over the set $\{m \in \mathbb{N}^I \mid [m]_1 \leq [i]_1, \ldots, [m]_I \leq [i]_I, \|m\|_1 > 0\}$. Furthermore, we utilize the generalized binomial coefficient

$$\binom{a}{b} := \prod_{l=0}^{b-1} \frac{a-l}{b-l}$$

to allow the computation for all $a \in \mathbb{R}$ and $b \in \mathbb{N}$, which is defined to be 1 if $b = 0$. The generalized binomial coefficient of vectors is the component-wise product of generalized binomial coefficients:

$$\binom{a}{b} := \prod_{i=1}^{I} \binom{[a]_i}{[b]_i}.$$

This notation also includes cases where the vector $a$ has components of $\mathbb{R}$.

**Example computation.** Let us compute the coefficient $\gamma_{(2,2),(3,1)}$, used by the biharmonic operator:

$$\gamma_{(2,2),(3,1)} = \sum_{\substack{m \in \mathbb{N}, \|m\|_1 > 0 \\ [m]_1 \leq 2, [m]_2 \leq 2}} (-1)^{2-[m]_1 + 2 - [m]_2} \binom{2}{[m]_1} \binom{2}{[m]_2} \binom{4 \frac{[m]_1}{\|m\|_1}}{3} \binom{4 \frac{[m]_2}{\|m\|_1}}{1} \left(\frac{\|m\|_1}{4}\right)^4.$$

We have $m \in \{(1,0),(2,0),(1,1),(2,1),(2,2),(1,2),(0,1),(0,2)\}$, which results in the terms

$$= \quad (-1)^{2-1+2-0} \binom{2}{1} \binom{2}{0} \binom{4\frac{1}{1}}{3} \binom{4\frac{0}{1}}{1} \left(\frac{1}{4}\right)^4$$

$$+ (-1)^{2-2+2-0} \binom{2}{2} \binom{2}{0} \binom{4\frac{2}{2}}{3} \binom{4\frac{0}{2}}{1} \left(\frac{2}{4}\right)^4$$

$$+ (-1)^{2-1+2-1} \binom{2}{1} \binom{2}{1} \binom{4\frac{1}{2}}{3} \binom{4\frac{1}{2}}{1} \left(\frac{2}{4}\right)^4$$

$$+ (-1)^{2-2+2-1} \binom{2}{2} \binom{2}{1} \binom{4\frac{2}{3}}{3} \binom{4\frac{1}{3}}{1} \left(\frac{3}{4}\right)^4$$

$$+ (-1)^{2-2+2-2} \binom{2}{2} \binom{2}{2} \binom{4\frac{2}{4}}{3} \binom{4\frac{2}{4}}{1} \left(\frac{4}{4}\right)^4$$

$$+ (-1)^{2-1+2-2} \binom{2}{1} \binom{2}{2} \binom{4\frac{1}{3}}{3} \binom{4\frac{2}{3}}{1} \left(\frac{3}{4}\right)^4$$

$$+ (-1)^{2-0+2-1} \binom{2}{0} \binom{2}{1} \binom{4\frac{0}{1}}{3} \binom{4\frac{1}{1}}{1} \left(\frac{1}{4}\right)^4$$

$$+ (-1)^{2-0+2-2} \binom{2}{0} \binom{2}{2} \binom{4\frac{0}{2}}{3} \binom{4\frac{2}{2}}{1} \left(\frac{2}{4}\right)^4.$$

The next step is to evaluate the binomial coefficients:

$$
\begin{aligned}
= \quad & (-1) \cdot 2 \cdot 1 \cdot 4 \cdot 0 \cdot \left(\frac{1}{4}\right)^4 \\
& +1 \cdot 1 \cdot 4 \cdot 0 \cdot \left(\frac{2}{4}\right)^4 \\
& +2 \cdot 2 \cdot 0 \cdot 2 \cdot \left(\frac{2}{4}\right)^4 \\
& -1 \cdot 2 \cdot \frac{8}{9}\frac{5}{6}\frac{2}{3} \cdot \frac{4}{3} \cdot \left(\frac{3}{4}\right)^4 \\
& +1 \cdot 1 \cdot 0 \cdot 2 \cdot \left(\frac{4}{4}\right)^4 \\
& -2 \cdot 1 \cdot \frac{4}{9}\frac{1}{6}\frac{-2}{3} \cdot \frac{8}{3} \cdot \left(\frac{3}{4}\right)^4 \\
& -1 \cdot 2 \cdot 0 \cdot 4 \cdot \left(\frac{1}{4}\right)^4 \\
& +1 \cdot 1 \cdot 1 \cdot 0 \cdot 4 \cdot \left(\frac{2}{4}\right)^4 .
\end{aligned}
$$

After simplification, the final result is

$$
\begin{aligned}
\gamma_{(2,2),(3,1)} &= (-1) \cdot 2 \cdot \frac{8}{9}\frac{5}{6}\frac{2}{3} \cdot \frac{4}{3} \cdot \left(\frac{3}{4}\right)^4 - 2\left(\frac{4}{9} \cdot \frac{1}{6} \cdot \frac{-2}{3}\right) \cdot \frac{8}{3} \cdot \left(\frac{3}{4}\right)^4 \\
&= \frac{-640}{486}\frac{81}{256} + \frac{128}{486}\frac{81}{256} = \frac{-5}{12} + \frac{1}{12} = -\frac{1}{3}.
\end{aligned}
$$

### E.1 Applied to the Biharmonic Operator

To compute eq. (9) with eq. (11), we first select $K = 4, I = 2, D_1 = D_2 = D, \boldsymbol{i} = (2,2), \boldsymbol{v}_{d_1} = \boldsymbol{e}_{d_1}$ and $\boldsymbol{e}_{d_2} = \boldsymbol{e}_{d_2}$. Then we insert these parameters into the general equation eq. (11) and get

$$
\begin{aligned}
\Delta^2 \boldsymbol{f}(\boldsymbol{x}_0) = & \sum_{\boldsymbol{j}\in\mathbb{N}^2, \|\boldsymbol{j}\|_1=4} \gamma_{(2,2),\boldsymbol{j}} \frac{1}{4!} \sum_{d_1=1}^{D}\sum_{d_2=1}^{D} \left\langle \partial^4 \boldsymbol{f}(\boldsymbol{x}_0), (\boldsymbol{e}_{d_1}[\boldsymbol{j}]_1 + \boldsymbol{e}_{d_2}[\boldsymbol{j}]_2)^{\otimes 4} \right\rangle \\
= & \frac{1}{24}\Bigg(\gamma_{(2,2),(4,0)} \sum_{d_1=1}^{D}\sum_{d_2=1}^{D} \left\langle \partial^4 \boldsymbol{f}(\boldsymbol{x}_0), (4\boldsymbol{e}_{d_1})^{\otimes 4} \right\rangle \\
& + \gamma_{(2,2),(0,4)} \sum_{d_1=1}^{D}\sum_{d_2=1}^{D} \left\langle \partial^4 \boldsymbol{f}(\boldsymbol{x}_0), (4\boldsymbol{e}_{d_2})^{\otimes 4} \right\rangle \\
& + \gamma_{(2,2),(3,1)} \sum_{d_1=1}^{D}\sum_{d_2=1}^{D} \left\langle \partial^4 \boldsymbol{f}(\boldsymbol{x}_0), (3\boldsymbol{e}_{d_1} + \boldsymbol{e}_{d_2})^{\otimes 4} \right\rangle \\
& + \gamma_{(2,2),(1,3)} \sum_{d_1=1}^{D}\sum_{d_2=1}^{D} \left\langle \partial^4 \boldsymbol{f}(\boldsymbol{x}_0), (\boldsymbol{e}_{d_1} + 3\boldsymbol{e}_{d_2})^{\otimes 4} \right\rangle \\
& + \gamma_{(2,2),(2,2)} \sum_{d_1=1}^{D}\sum_{d_2=1}^{D} \left\langle \partial^4 \boldsymbol{f}(\boldsymbol{x}_0), (2\boldsymbol{e}_{d_1} + 2\boldsymbol{e}_{d_2})^{\otimes 4} \right\rangle \Bigg).
\end{aligned}
\tag{E18}
$$

Now, exploit the symmetry of the coefficients $\gamma_{(2,2),(4,0)} = \gamma_{(2,2),(0,4)}$ and $\gamma_{(2,2),(3,1)} = \gamma_{(2,2),(1,3)}$ and the corresponding tensor basis expansion:

$$
\begin{aligned}
= \frac{1}{24} \Big( & 2D\gamma_{(2,2),(4,0)} \sum_{d_1=1}^{D} \left\langle \partial^4 \boldsymbol{f}(\boldsymbol{x}_0), (4\boldsymbol{e}_{d_1})^{\otimes 4} \right\rangle \\
& + 2\gamma_{(2,2),(3,1)} \sum_{d_1=1}^{D} \sum_{d_2=1}^{D} \left\langle \partial^4 \boldsymbol{f}(\boldsymbol{x}_0), (3\boldsymbol{e}_{d_1} + \boldsymbol{e}_{d_2})^{\otimes 4} \right\rangle \\
& + \gamma_{(2,2),(2,2)} \sum_{d_1=1}^{D} \sum_{d_2=1}^{D} \left\langle \partial^4 \boldsymbol{f}(\boldsymbol{x}_0), (2\boldsymbol{e}_{d_1} + 2\boldsymbol{e}_{d_2})^{\otimes 4} \right\rangle \Big).
\end{aligned}
\tag{E19}
$$

Since the first sum captures all diagonal directions $\boldsymbol{e}_{d_1} = \boldsymbol{e}_{d_2}$, we extract this from the second and third sums to further reduce the computational effort:

$$
\begin{aligned}
= \frac{1}{24} \Big( & \left( 2D\gamma_{(2,2),(4,0)} + 2\gamma_{(2,2),(3,1)} + \gamma_{(2,2),(2,2)} \right) \sum_{d_1=1}^{D} \left\langle \partial^4 \boldsymbol{f}(\boldsymbol{x}_0), (4\boldsymbol{e}_{d_1})^{\otimes 4} \right\rangle \\
& + 2\gamma_{(2,2),(3,1)} \sum_{d_1=1}^{D} \sum_{\substack{d_2=1 \\ d_2 \neq d_1}}^{D} \left\langle \partial^4 \boldsymbol{f}(\boldsymbol{x}_0), (3\boldsymbol{e}_{d_1} + \boldsymbol{e}_{d_2})^{\otimes 4} \right\rangle \\
& + \gamma_{(2,2),(2,2)} \sum_{d_1=1}^{D} \sum_{\substack{d_2=1 \\ d_2 \neq d_1}}^{D} \left\langle \partial^4 \boldsymbol{f}(\boldsymbol{x}_0), (2\boldsymbol{e}_{d_1} + 2\boldsymbol{e}_{d_2})^{\otimes 4} \right\rangle \Big).
\end{aligned}
\tag{E20}
$$

Exploiting further symmetries in the last term's summation, we obtain

$$
\begin{aligned}
\Delta^2 \boldsymbol{f}(\boldsymbol{x}_0) = \frac{1}{24} \Big( & \left( 2D\gamma_{(2,2),(4,0)} + 2\gamma_{(2,2),(3,1)} + \gamma_{(2,2),(2,2)} \right) \sum_{d_1=1}^{D} \left\langle \partial^4 \boldsymbol{f}(\boldsymbol{x}_0), (4\boldsymbol{e}_{d_1})^{\otimes 4} \right\rangle \\
& + 2\gamma_{(2,2),(3,1)} \sum_{d_1=1}^{D} \sum_{\substack{d_2=1 \\ d_2 \neq d_1}}^{D} \left\langle \partial^4 \boldsymbol{f}(\boldsymbol{x}_0), (3\boldsymbol{e}_{d_1} + \boldsymbol{e}_{d_2})^{\otimes 4} \right\rangle \\
& + 2\gamma_{(2,2),(2,2)} \sum_{d_1=1}^{D-1} \sum_{d_2=d_1+1}^{D} \left\langle \partial^4 \boldsymbol{f}(\boldsymbol{x}_0), (2\boldsymbol{e}_{d_1} + 2\boldsymbol{e}_{d_2})^{\otimes 4} \right\rangle \Big).
\end{aligned}
\tag{E21}
$$

### E.2 Pedagogical Approach for the Biharmonic Operator with 6-jets

A different approach to compute arbitrary-mixed derivatives was proposed in [27]. This approach relies, for the biharmonic operator, on the hand-selection of certain 6-jets to extract the required derivatives. The degree and directions for the jets are obtained by considering the Faà di Bruno formula for the 6-th coefficient $\boldsymbol{f}_6$ (see §A). Selecting coefficients of the input 6-jet to $\boldsymbol{x}_1 = \boldsymbol{e}_{d_1}, \boldsymbol{x}_2 = \boldsymbol{e}_{d_2}$ and $\boldsymbol{x}_3 = \boldsymbol{x}_4 = \boldsymbol{x}_5 = \boldsymbol{x}_6 = \boldsymbol{0}$ leads us to

$$
\begin{aligned}
\boldsymbol{f}_6 = & \left\langle \partial^6 \boldsymbol{f}(\boldsymbol{x}_0), \boldsymbol{e}_{d_1}^{\otimes 6} \right\rangle + 15 \left\langle \partial^5 \boldsymbol{f}(\boldsymbol{x}_0), \boldsymbol{e}_{d_1}^{\otimes 4} \otimes \boldsymbol{e}_{d_2} \right\rangle \\
& + 45 \left\langle \partial^4 \boldsymbol{f}(\boldsymbol{x}_0), \boldsymbol{e}_{d_1}^{\otimes 2} \otimes \boldsymbol{e}_{d_2}^{\otimes 2} \right\rangle + 15 \left\langle \partial^3 \boldsymbol{f}(\boldsymbol{x}_0), \boldsymbol{e}_{d_2}^{\otimes 3} \right\rangle.
\end{aligned}
\tag{E22}
$$

Notice the blue term, which has the same structure as the summands we want to compute for the biharmonic operator. Therefore, a first 6-jet is computed as explained above. To cancel out the unwanted terms, we evaluate another 6-jet with the same input except $\boldsymbol{x}_2 = -\boldsymbol{e}_{d_2}$ and adding the 6-th coefficient of this jet to eq. (E22) gives

$$
2\left\langle \partial^6 \boldsymbol{f}(\boldsymbol{x}_0), \boldsymbol{e}_{d_1}^{\otimes 6} \right\rangle + 90 \left\langle \partial^4 \boldsymbol{f}(\boldsymbol{x}_0), \boldsymbol{e}_{d_1}^{\otimes 2} \otimes \boldsymbol{e}_{d_2}^{\otimes 2} \right\rangle.
\tag{E23}
$$

Finally, a third 6-jet is computed with $\boldsymbol{x}_2 = \boldsymbol{0}$. The 6-th coefficient of this jet contains only

$$
\left\langle \partial^6 \boldsymbol{f}(\boldsymbol{x}_0), \boldsymbol{x}_1^{\otimes 6} \right\rangle.
\tag{E24}
$$

We obtain

$$90 \left\langle \partial^4 \boldsymbol{f}(\boldsymbol{x}_0), \boldsymbol{x}_1^{\otimes 2} \otimes \boldsymbol{x}_2^{\otimes 2} \right\rangle \tag{E25}$$

by subtracting twice of the 6-th coefficient of the third jet from eq. (E23).

To summarize the procedure, we evaluate the 6-jet three times. The first jet has the input $\boldsymbol{x}_1 = \boldsymbol{e}_{d_1}, \boldsymbol{x}_2 = \boldsymbol{e}_{d_2}$ and $\boldsymbol{x}_3 = \boldsymbol{x}_4 = \boldsymbol{x}_5 = \boldsymbol{x}_6 = \boldsymbol{0}$, the second jet has the same input jet apart from $\boldsymbol{x}_2 = -\boldsymbol{e}_{d_2}$, and the third 6-jet takes $\boldsymbol{x}_2 = \boldsymbol{0}$. Then we add the 6-th coefficient of the first and the second and subtract twice of the 6-th coefficient of the third jet. Dividing by 90 provides the derivative corresponding to the $d_1, d_2$ term of the biharmonic operator.

Standard Taylor mode would propagate $1 + 18D^2$ vectors through every node, if we already exploit that all jets share $\boldsymbol{x}_0$. our collapsed Taylor mode would pass $1 + 3 + 15D^2$ vectors through every node of the compute graph. This is more costly compared to our approach described before. In addition, until now, the selection of the jet degree and the input coefficients requires substantial human effort.

### E.3 Another Example: Mixed Third-order Derivatives

As an additional example, consider computing $\sum_{i=1}^{D} \sum_{j=1}^{D} \frac{\partial^3}{\partial x_i^2 x_j} \boldsymbol{f}(\boldsymbol{x})$. This example is from [27, §F.2], which describes how to compute these 3rd-order derivatives using 7-jets. The interpolation formula allows using multiple 3-jets instead. We expect it to be favorable as Taylor mode scales polynomially in the derivative order.

**Procedure.** The goal is to compute $\sum_{i=1}^{D} \sum_{j=1}^{D} \frac{\partial^3}{\partial x_i^2 \partial x_j} \boldsymbol{f}(\boldsymbol{x})$. We proceed as follows:

1. Formulate the operator in our notation:

$$\sum_{i=1}^{D} \sum_{j=1}^{D} \langle \partial^3 \boldsymbol{f}(\boldsymbol{x}), \boldsymbol{e}_i^{\otimes 2} \otimes \boldsymbol{e}_j \rangle$$

2. Compute the interpolation coefficients $\gamma_{\boldsymbol{p},\boldsymbol{q}}$ for $\boldsymbol{q} \in \{(3,0), (2,1), (1,2), (0,3)\}$ and $\boldsymbol{p} = (2,1)$: $\gamma_{(2,1)(0,3)} = -8/81, \gamma_{(2,1)(1,2)} = 16/27, \gamma_{(2,1)(2,1)} = -16/9, \gamma_{(2,1)(3,0)} = 32/81$.

3. Apply the interpolation equation (eq. (11)):

$$= \sum_{i=1}^{D} \sum_{j=1}^{D} \sum_{\boldsymbol{q} \in \mathbb{N}^2, \, \|\boldsymbol{q}\|_1 = 3} \langle \partial^3 \boldsymbol{f}(\boldsymbol{x}), ([\boldsymbol{q}]_1 \boldsymbol{e}_i + [\boldsymbol{q}]_2 \boldsymbol{e}_j)^{\otimes 3} \rangle$$

Collapsed Taylor mode can directly applied to these $4D^2$ 3-jets. However, to exploit the full potential some further steps that leverage the structure are required.

4. Expand and manually simplify, using symmetries. The sums for $\gamma_{(2,1)(3,0)}$ and $\gamma_{(2,1)(0,3)}$ are similar; same for $\gamma_{(2,1)(2,1)}$ and $\gamma_{(2,1)(1,2)}$. We only have $2D^2 - 3$ jets:

$$= (\gamma_{(2,1)(3,0)} + \gamma_{(2,1)(0,3)}) \sum_{i=1}^{D} \sum_{j=1}^{D} \langle \partial^3 \boldsymbol{f}(\boldsymbol{x}), (3\boldsymbol{e}_i)^{\otimes 3} \rangle$$

$$+ (\gamma_{(2,1)(2,1)} + \gamma_{(2,1)(1,2)}) \sum_{i=1}^{D} \sum_{j=1}^{D} \langle \partial^3 \boldsymbol{f}(\boldsymbol{x}), (2\boldsymbol{e}_i + \boldsymbol{e}_j)^{\otimes 3} \rangle .$$

We further observe that the first summation is independent of $j$:

$$= (\gamma_{(2,1)(3,0)} + \gamma_{(2,1)(0,3)}) D \sum_{i=1}^{D} \langle \partial^3 \boldsymbol{f}(\boldsymbol{x}), (3\boldsymbol{e}_i)^{\otimes 3} \rangle$$

$$+ (\gamma_{(2,1)(2,1)} + \gamma_{(2,1)(1,2)}) \sum_{i=1}^{D} \sum_{j=1}^{D} \langle \partial^3 \boldsymbol{f}(\boldsymbol{x}), (2\boldsymbol{e}_i + \boldsymbol{e}_j)^{\otimes 3} \rangle .$$

Extracting the case $i = j$ from the last term gives our final form

$$= ((\gamma_{(2,1)(3,0)}D + \gamma_{(2,1)(0,3)}D + \gamma_{(2,1)(2,1)} + \gamma_{(2,1)(1,2)}) \sum_{i=1}^{D} \langle \partial^3 \boldsymbol{f}(\boldsymbol{x}), (3\boldsymbol{e}_i)^{\otimes 3} \rangle$$

$$+ (\gamma_{(2,1)(2,1)} + \gamma_{(2,1)(1,2)}) \sum_{i=1}^{D} \sum_{j=1, j \neq i}^{D} \langle \partial^3 \boldsymbol{f}(\boldsymbol{x}), (2\boldsymbol{e}_i + \boldsymbol{e}_j)^{\otimes 3} \rangle.$$

This optimized version required $D^2$ 3-jets that can be collapsed.

# F  PyTorch Benchmark

## F.1  Additional Analysis and Impact of `torch.compile`

Here, we compare the theoretically estimated performance improvements based on counting the number of forward-propagated vectors with the empirically measured performance.

**The number of propagated vectors is a good empirical performance estimate.**  To estimate the performance ratio between standard and collapsed Taylor mode, we can use the number of additional vectors both modes propagate forward as we increase either the batch size or the number of Monte-Carlo samples. This is a relatively simplistic proxy; e.g., it assumes that each vector adds the same computational load, which is inaccurate as vectors corresponding to higher coefficients require more work and memory (as the Faà di Bruno formula contains more terms in general). Conversely, while incrementing the MC samples does add additional vectors that are propagated, it does not introduce additional cost to compute or store the derivatives, as they are already computed with just a single sample. Table F2 summarizes the theoretical and empirical ratios. We find them to align quite well, despite the overly simplistic assumptions.

**Concrete example.**  Consider the exact Laplacian. Adding one datum introduces $2 + D$ versus $1 + 2D$ new vectors. For $D = 50$, their ratio is ${(2+D)}/{(1+2D)} \approx 0.51$. Empirically, we measure that adding one datum adds $0.56\,\text{ms}$ to standard, and $0.29\,\text{ms}$ to collapsed, Taylor mode (table 1); the ratio of $\approx 0.52$ is close.

**Compilation reduces memory, but not runtime.**  In table F3, we repeat the benchmark from table 1 using `torch.compile`. We observe that compiling can further reduce the memory footprint of all approaches for computing the Laplacian and weighted Laplacian, while the runtime remains roughly identical. For the biharmonic operator, we observe that compilation leaves runtime and memory footprint unchanged.

Table F2: **Comparison of theoretical and empirical performance ratios between standard and collapsed Taylor mode.** We list the number of additional vectors that are used when adding another data point (exact) or another Monte-Carlo sample (stochastic). The ratio of vectors offers a good estimate of the empirically measured performance ratio.

| Mode | Add one datum or MC sample | Laplacian $(D = 50)$ | Weighted Laplacian $(D = R = 50)$ | Biharmonic $(D = 5)$ |
|---|---|---|---|---|
| **Exact** | # vectors (standard) | $1 + 2D$ | $1 + 2R$ | $6D^2 - 2D + 1$ |
| | # vectors (collapsed) | $2 + D$ | $2 + R$ | $^9/_2 D^2 - ^3/_2 D + 4$ |
| | Theoretical ratio #/# | 0.51 | 0.51 | 0.77 |
| | Empirical time ratio | 0.52 | 0.51 | 0.76 |
| | Empirical mem. ratio | 0.45 | 0.45 | 0.63 |
| **Stochastic** | # vectors (standard) | 2 | 2 | 4 |
| | # vectors (collapsed) | 1 | 1 | 3 |
| | Theoretical ratio #/# | 0.5 | 0.5 | 0.75 |
| | Empirical time ratio | 0.51 | 0.51 | 0.74 |
| | Empirical mem. ratio | 0.43 | 0.45 | 0.59 |

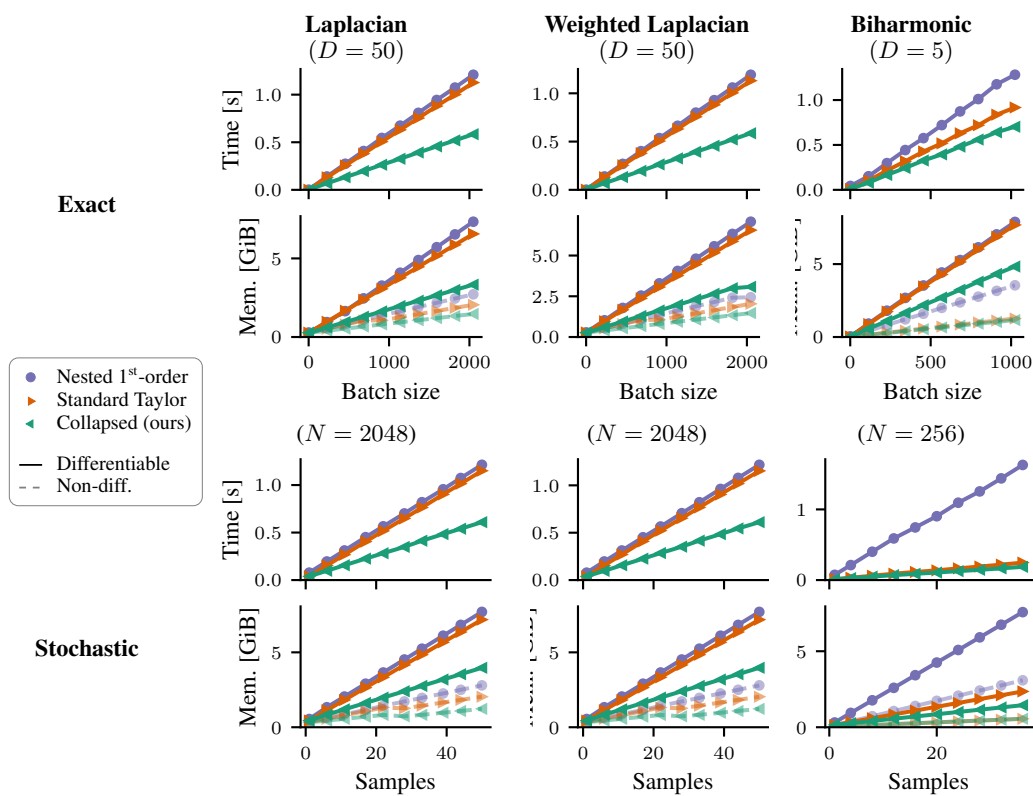

Figure F9: **Same as fig. 5, but using** `torch.compile`.

Table F3: **Same as table 1, but using** `torch.compile` **(i.e. fig. F9 in numbers).**

| Mode | Per-datum or -sample cost | Implementation | Laplacian | Weighted Laplacian | Biharmonic via interpolation (E21) |
|------|---------------------------|----------------|-----------|--------------------|------------------------------------|
| **Exact** | Time [ms] | Nested 1st-order | 0.59 (1.0x) | 0.58 (1.0x) | 1.2 (1.0x) |
| | | Standard Taylor | 0.55 (0.93x) | 0.55 (0.95x) | 0.90 (0.72x) |
| | | Collapsed (ours) | **0.28 (0.48x)** | **0.29 (0.49x)** | **0.69 (0.55x)** |
| | Mem. [MiB] (differentiable) | Nested 1st-order | 3.6 (1.0x) | 3.4 (1.0x) | 7.9 (1.0x) |
| | | Standard Taylor | 3.1 (0.88x) | 3.1 (0.92x) | 7.7 (0.98x) |
| | | Collapsed (ours) | **1.5 (0.43x)** | **1.5 (0.43x)** | **4.8 (0.61x)** |
| | Mem. [MiB] (non-diff.) | Nested 1st-order | 1.2 (1.0x) | 1.1 (1.0x) | 3.5 (1.0x) |
| | | Standard Taylor | 0.85 (0.69x) | 0.84 (0.73x) | 1.2 (0.36x) |
| | | Collapsed (ours) | **0.60 (0.49x)** | **0.60 (0.53x)** | **1.1 (0.33x)** |
| **Stochastic** | Time [ms] | Nested 1st-order | 23 (1.0x) | 23 (1.0x) | 44 (1.0x) |
| | | Standard Taylor | 23 (0.98x) | 23 (0.98x) | 6.6 (0.15x) |
| | | Collapsed (ours) | **12 (0.50x)** | **12 (0.50x)** | **4.9 (0.11x)** |
| | Mem. [MiB] (differentiable) | Nested 1st-order | 150 (1.0x) | 150 (1.0x) | 210 (1.0x) |
| | | Standard Taylor | 140 (0.95x) | 140 (0.95x) | 64 (0.30x) |
| | | Collapsed (ours) | **73 (0.49x)** | **73 (0.49x)** | **38 (0.18x)** |
| | Mem. [MiB] (non-diff.) | Nested 1st-order | 50 (1.0x) | 50 (1.0x) | 86 (1.0x) |
| | | Standard Taylor | 33 (0.67x) | 33 (0.67x) | **15 (0.17x)** |
| | | Collapsed (ours) | **17 (0.33x)** | **17 (0.33x)** | **15 (0.17x)** |

Table F4: **Exact weighted Laplacian for different ranks of the weightings.** The lower the rank, the lower the memory and time consumption. Both scale approximately linear with the rank—with stronger deviations for small ranks—as predicted by our theoretical analysis based on the number of forward-propagated coefficients. The 'Full-rank' column is identical to the 'Weighted Laplacian' columns in Tables 1 and F3.

| compile | Per-datum or -sample cost | Implementation | Weighted Laplacian ($D = 50$) | | |
| | | | Full-rank ($D$) | Half-full rank ($D/2$) | Low-rank ($D/10$) |
| --- | --- | --- | --- | --- | --- |
| ✓ | Time [ms] | Nested 1$^{st}$-order | 0.58 (1.0x) | 0.30 (1.0x) | 0.082 (1.0x) |
| | | Standard Taylor | 0.55 (0.95x) | 0.28 (0.92x) | 0.062 (0.76x) |
| | | Collapsed (ours) | **0.29 (0.49x)** | **0.15 (0.49x)** | **0.040 (0.49x)** |
| | Mem. [MiB] (differentiable) | Nested 1$^{st}$-order | 3.4 (1.0x) | 1.8 (1.0x) | 0.43 (1.0x) |
| | | Standard Taylor | 3.1 (0.92x) | 1.6 (0.90x) | 0.33 (0.78x) |
| | | Collapsed (ours) | **1.5 (0.43x)** | **0.80 (0.46x)** | **0.21 (0.50x)** |
| | Mem. [MiB] (non-diff.) | Nested 1$^{st}$-order | 1.1 (1.0x) | 0.62 (1.0x) | 0.15 (1.0x) |
| | | Standard Taylor | 0.84 (0.73x) | 0.47 (0.76x) | 0.12 (0.82x) |
| | | Collapsed (ours) | **0.60 (0.53x)** | **0.31 (0.49x)** | **0.073 (0.48x)** |
| ✗ | Time [ms] | Nested 1$^{st}$-order | 0.60 (1.0x) | 0.31 (1.0x) | 0.084 (1.0x) |
| | | Standard Taylor | 0.57 (0.94x) | 0.29 (0.92x) | 0.065 (0.78x) |
| | | Collapsed (ours) | **0.29 (0.48x)** | **0.15 (0.49x)** | **0.042 (0.51x)** |
| | Mem. [MiB] (differentiable) | Nested 1$^{st}$-order | 4.4 (1.0x) | 2.2 (1.0x) | 0.53 (1.0x) |
| | | Standard Taylor | 4.6 (1.0x) | 2.4 (1.0x) | 0.60 (1.1x) |
| | | Collapsed (ours) | **2.1 (0.47x)** | **1.1 (0.50x)** | **0.39 (0.74x)** |
| | Mem. [MiB] (non-diff.) | Nested 1$^{st}$-order | 2.2 (1.0x) | 1.1 (1.0x) | 0.26 (1.0x) |
| | | Standard Taylor | 1.2 (0.54x) | 0.60 (0.54x) | 0.14 (0.53x) |
| | | Collapsed (ours) | **0.90 (0.41x)** | **0.46 (0.41x)** | **0.11 (0.41x)** |

## F.2  Rank-deficient Weighted Laplacian

In the main text we use a weighted Laplacian with a full-rank weight matrix (i.e., $R := \text{rank}(\boldsymbol{D}) = D$). Since the weight matrix has full rank, the weighted Laplacian is as expensive as the unweighted Laplacian, and this is confirmed by our experiments. To show that the weight matrix's rank indeed affects the cost, we experiment with a rank-deficient weight matrix in this section and also consider the ranks $R \in \{D/2, D/10\}$. Table F4 contains the results of this analysis. We observe that going from full to half-full rank roughly halves both the runtime and memory consumption for all implementations. For small ranks, this linear relationship weakens because the fraction of computations that do not scale with $R$ grows.

# G  JAX Benchmark

This section presents experiments which show that the graph simplifications we propose to collapse standard Taylor mode are currently not applied by the `jit` compiler in JAX.

**Comparing JAX implementations.** Similar to our PyTorch experiment in §4, we compare three implementations of the Laplacian in JAX (all compiled with `jax.jit`):

1. **Nested 1$^{st}$-order AD** computes the Hessian using `jax.hessian`, which relies on forward-over-reverse mode, then traces it.

2. **Standard Taylor mode** propagates multiple univariate Taylor polynomials, each of which computes one element of the Hessian diagonal, then sums them to obtain the Laplacian. This is implemented with `jax.experimental.jet.jet` and `jax.vmap`.

3. **Collapsed Taylor mode** relies on the forward Laplacian implementation in JAX provided by the `folx` library [12] and implements our proposed collapsed Taylor mode for the specific case of the Laplacian. `folx` also enables leveraging sparsity in the tensors, which is

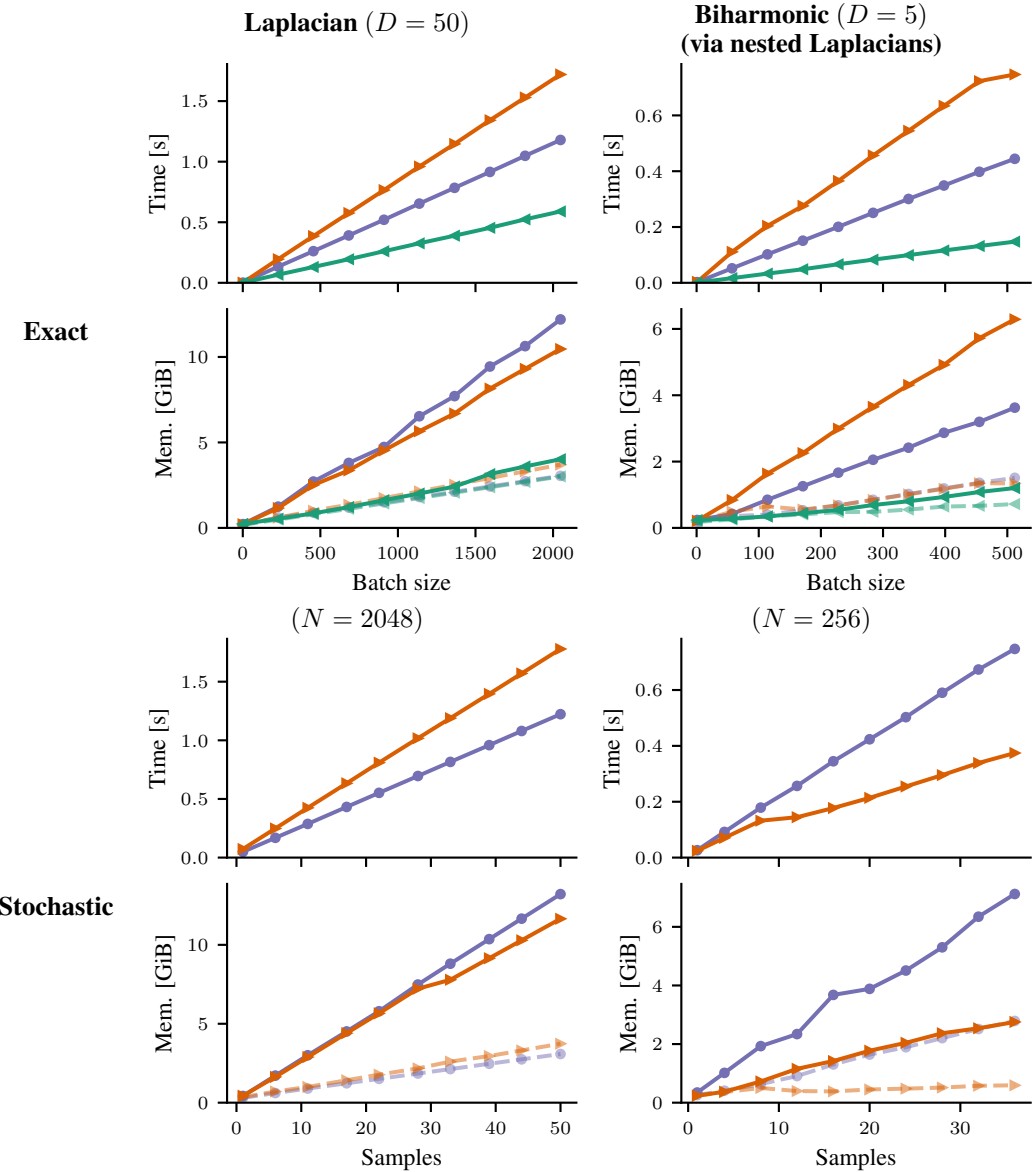

Figure G10: **JAX's `jit` compiler does not apply our graph simplifications to standard Taylor mode.** Colors: Collapsed Taylor mode, standard Taylor mode, and nested first-order automatic differentiaion, opaque memory consumptions are for non-differentiable computations. Results are on GPU and we use a $D \to 768 \to 768 \to 512 \to 512 \to 1$ MLP with tanh activations, varying the batch size. For each approach, we fit a line to the data and report the slope in table G5 to quantify the relative speedup and memory reduction.

beneficial for architectures in VMC. To disentangle runtime improvements from sparsity detection versus collapsing Taylor coefficient, we disable `folx`'s sparsity detection.

For the biharmonic operator, we simply nest the Laplacian implementations.

We only investigate computing the exact Laplacian, as the forward Laplacian in `folx` currently does not support stochastic computation. We use the same neural network architecture as for our PyTorch experiments, fix the input dimension to $D = 50$ and vary the batch size, recording the runtime and peak memory with the same protocol as described in the main text. JAX is purely functional and therefore does not have a mechanism to build up a differentiable computational graph similar to evaluating a function in PyTorch where some leafs have `requires_grad=True`. To approximate the

Table G5: **JAX Benchmark from fig. G10 in numbers.** We fit linear functions and report their slopes, i.e., how much runtime and memory increase when incrementing the batch size. All numbers are shown with two significant digits and bold values are best according to parenthesized values.

| Mode | Per-datum or sample cost | Implementation | Laplacian | Biharmonic (via nested Laplacians) |
|---|---|---|---|---|
| **Exact** | Time [ms] | Nested $1^{st}$-order | 0.58 (1.0x) | 0.87 (1.0x) |
| | | Standard Taylor | 0.84 (1.5x) | 1.5 (1.7x) |
| | | Collapsed (ours) | **0.29 (0.50x)** | **0.29 (0.33x)** |
| | Mem. [MiB] (differentiable) | Nested $1^{st}$-order | 6.0 (1.0x) | 7.0 (1.0x) |
| | | Standard Taylor | 5.1 (0.85x) | 12 (1.8x) |
| | | Collapsed (ours) | **1.9 (0.32x)** | **2.0 (0.29x)** |
| | Mem. [MiB] (non-diff.) | Nested $1^{st}$-order | 1.4 (1.0x) | 2.7 (1.0x) |
| | | Standard Taylor | 1.7 (1.2x) | 2.2 (0.83x) |
| | | Collapsed (ours) | **1.4 (1.0x)** | **1.1 (0.39x)** |
| **Stochastic** | Time [ms] | Nested $1^{st}$-order | **24 (1.0x)** | 21 (1.0x) |
| | | Standard Taylor | 35 (1.5x) | **9.5 (0.46x)** |
| | | Collapsed (ours) | Not implemented | Not implemented |
| | Mem. [MiB] (differentiable) | Nested $1^{st}$-order | 270 (1.0x) | 190 (1.0x) |
| | | Standard Taylor | **230 (0.87x)** | **77 (0.40x)** |
| | | Collapsed (ours) | Not implemented | Not implemented |
| | Mem. [MiB] (non-diff.) | Nested $1^{st}$-order | **58 (1.0x)** | 77 (1.0x) |
| | | Standard Taylor | 71 (1.2x) | **7.9 (0.10x)** |
| | | Collapsed (ours) | Not implemented | Not implemented |

peak memory of computing a differentiable Laplacian in JAX, we measure the peak memory of first computing the Laplacian, then evaluating the gradient w.r.t. the neural network's parameters which backpropagates through the same computation graph built by PyTorch.

**Results (Laplacian).** The left column of fig. G10 visualizes the performance of the three implementations. We fit linear functions to each of them and report the cost incurred by adding one more datum to the batch in table G5. From them, we draw the following conclusions:

1. **Performance is consistent between PyTorch and JAX.** Although our PyTorch implementation does not leverage compilation, the values reported in tables 1 and G5 are consistent and only in rare cases differ by a factor of more than two. This confirms that our PyTorch-based implementation of Taylor mode is reasonably efficient, and that the presented performance results in the main text are transferable to other frameworks like JAX.

2. **Our implementation of collapsed Taylor mode based on graph rewrites in PyTorch achieves consistent speed-up with the Laplacian-specific implementation in JAX.** Specifically, we observe that collapsed Taylor mode/forward Laplacian use roughly half the runtime of nested $1^{st}$-order AD (compare tables 1 and G5). This supports our argument that our collapsed Taylor is indeed a generalization of the forward Laplacian, i.e., the latter does not employ additional tricks (leveraging sparsity could also be applied to our approach but we are not aware of a drop-in implementation). It also illustrates that the savings we report in PyTorch carry over to other frameworks like JAX.

3. **JAX's `jit` compiler is unable to apply the graph rewrites we propose in this work.** If the JAX compiler was able to perform our proposed graph rewrites, then the `jit`-compiled standard Taylor mode should yield similar performance than the forward Laplacian. However, we observe a clear performance gap in runtime and memory, from which we conclude that the compilation did not collapse the Taylor coefficients. Our contribution is to point out that such rewrites could easily be added to the compiler's ability to unlock these performance gains at zero user overhead.

**Results (biharmonic operator).**  For the biharmonic operator (right column of fig. G10 and table G5), we conclude that (i) the most efficient way to compute biharmonics is by nesting Laplacians (compare with table 1 where Taylor mode uses the approach for general linear differential operators) and (ii) that nesting Taylor mode Laplacians is more efficient than nesting 1$^{\text{st}}$-order AD Laplacians, while also allowing to apply our collapsing technique.

## H  Numerical Complexity and Error Analysis

**Setup.**  To illustrate the numerical properties of our proposed collapsed Taylor mode, we consider a two-layered MLP with element-wise $\tanh$ activation $\boldsymbol{\phi} : \mathbb{R}^I \to \mathbb{R}^I$. The MLP is denoted by $\boldsymbol{f} := \boldsymbol{g} \circ \boldsymbol{\phi} \circ \boldsymbol{h}$. The two linear layers are given as $\boldsymbol{h} : \mathbb{R}^D \to \mathbb{R}^I, \boldsymbol{h}(\boldsymbol{x}_0) = \boldsymbol{W}_1 \boldsymbol{x}_0 + \boldsymbol{b}_1$ and $\boldsymbol{g} : \mathbb{R}^I \to \mathbb{R}^C$, $\boldsymbol{g}(\boldsymbol{\phi}_0) = \boldsymbol{W}_2 \boldsymbol{\phi}_0 + \boldsymbol{b}_2$, with weights $\boldsymbol{W}_1 \in \mathbb{R}^{I \times D}, \boldsymbol{W}_2 \in \mathbb{R}^{C \times I}$ and bias $\boldsymbol{b}_1 \in \mathbb{R}^I, \boldsymbol{b}_2 \in \mathbb{R}^C$. Below we compare the computational and storage complexity, as well as stability for evaluating the sum of the second coefficients $\sum_{r=1}^{R} \langle \partial^2 \boldsymbol{f}(\boldsymbol{x}_0), \boldsymbol{v}_i^{\otimes 2} \rangle = \sum_{r=1}^{R} \boldsymbol{g}_{2,r}$ (see eq. (5)) between collapsed and standard Taylor mode. For this toy example we show (i) collapsing uses less operations and (ii) both methods are similarly stable based on our simplified error analysis.

**Computational & storage complexity.**  Both vanilla and collapsed Taylor mode evaluate the function values $(\boldsymbol{h}_0, \boldsymbol{\phi}_0, \boldsymbol{g}_0)$ and the first derivatives $(\{\boldsymbol{h}_{1,r}, \boldsymbol{\phi}_{1,r}, \boldsymbol{g}_{1,r}\})$ by propagating $1 + R$ coefficients at each layer

$$\begin{pmatrix} \boldsymbol{h}_0 = \boldsymbol{W}_1 \boldsymbol{x}_0 + \boldsymbol{b}_1 \\ \{\boldsymbol{h}_{1,r}\} = \{\langle \boldsymbol{W}_1, \boldsymbol{x}_{1,r} \rangle\} \end{pmatrix} \overset{(3)}{\to} \begin{pmatrix} \boldsymbol{\phi}_0 = \boldsymbol{\phi}(\boldsymbol{h}_0) \\ \{\boldsymbol{\phi}_{1,r}\} = \{\langle \partial \boldsymbol{\phi}(\boldsymbol{h}_0), \boldsymbol{h}_{1,r} \rangle\} \end{pmatrix} \overset{(3)}{\to} \begin{pmatrix} \boldsymbol{g}_0 = \boldsymbol{W}_2 \boldsymbol{\phi}_0 + \boldsymbol{b}_2 \\ \{\boldsymbol{g}_{1,r}\} = \{\langle \boldsymbol{W}_2, \boldsymbol{\phi}_{1,r} \rangle\} \end{pmatrix}$$

with $\partial \boldsymbol{\phi}(\boldsymbol{h}_0) = \partial \tanh(\boldsymbol{h}_0) = \operatorname{diag}(\boldsymbol{1} - \boldsymbol{\phi}_0^{\odot 2}) \in \mathbb{R}^{I \times I}$ the $\tanh$-activation layer's Jacobian. The propagation costs $1 + R$ matrix-vector multiplications with $\boldsymbol{W}_1$, $1 + R$ matrix-vector multiplications with $\boldsymbol{W}_2$, $R$ Hadamard products with the derivative of $\boldsymbol{\phi}$ (since $\langle \operatorname{diag}(\boldsymbol{a}), \boldsymbol{h}_{1,r} \rangle = \boldsymbol{a} \odot \boldsymbol{h}_{1,r}$), and one Hadamard product to compute $\partial \boldsymbol{\phi}(\boldsymbol{h}_0)$. Additionally, there is one vector addition with the bias $\boldsymbol{b}_1$, one vector addition with $\boldsymbol{b}_2$, one vector subtraction in $\partial \boldsymbol{\phi}(\boldsymbol{h}_0)$ (counted as vector addition), as well as the evaluation of $\boldsymbol{\phi}(\boldsymbol{h}_0)$. $3 + 3R$ vectors are stored.

For the second derivatives, vanilla Taylor mode propagates $R$ vectors

$$(\{\boldsymbol{h}_{2,r}\} = \{\langle \boldsymbol{W}_1, \boldsymbol{x}_{2,r} \rangle\}) \overset{(3)}{\to} (\{\boldsymbol{\phi}_{2,r}\} = \{\langle \partial^2 \boldsymbol{\phi}(\boldsymbol{h}_0), \boldsymbol{h}_{1,r}^{\otimes 2} \rangle + \langle \partial \boldsymbol{\phi}(\boldsymbol{h}_0), \boldsymbol{h}_{2,r} \rangle\})$$

$$\overset{(3)}{\to} (\{\boldsymbol{g}_{2,r}\} = \{\langle \boldsymbol{W}_2, \boldsymbol{\phi}_{2,r} \rangle\})$$

with activation Hessian $\partial^2 \boldsymbol{\phi}(\boldsymbol{h}_0) \in \mathbb{R}^{I \times I \times I}$ of entries $[\partial^2 \boldsymbol{\phi}(\boldsymbol{h}_0)]_{i,j,k} = [-2\boldsymbol{\phi}_0 \odot (\boldsymbol{1} - \boldsymbol{\phi}_0^{\odot 2})]_i \delta_{i,j,k}$ and contraction $\langle \partial^2 \boldsymbol{\phi}(\boldsymbol{h}_0), \boldsymbol{h}_{1,r}^{\otimes 2} \rangle = -2\boldsymbol{\phi}_0 \odot (\boldsymbol{1} - \boldsymbol{\phi}_0^{\odot 2}) \odot \boldsymbol{h}_{1,r}^{\odot 2}$. These vectors are summed up to get the result $\sum_{r=1}^{R} \langle \partial^2 \boldsymbol{f}(\boldsymbol{x}_0), \boldsymbol{v}_i^{\otimes 2} \rangle = \sum_{r=1}^{R} \boldsymbol{g}_{2,r}$. This costs $2R$ matrix-vector products with the weights, $1 + 3R$ Hadamard products, $2R - 1$ vector additions, and a single scalar multiplication.

Table H6: **Comparison of theoretical computational and storage complexity** between standard Taylor mode and collapsed Taylor mode for a two-layer MLP computing the sum $\sum_{r=1}^{R} \langle \partial^2 \boldsymbol{f}(\boldsymbol{x}_0), \boldsymbol{v}_r^{\otimes 2} \rangle$.

| Computational Complexity | | |
| --- | --- | --- |
| **Operation** | Standard Taylor | Collapsed (ours) |
| # Matrix-vector products | $4R + 2$ | $2R + 4$ |
| # Hadamard products | $4R + 2$ | $3R + 3$ |
| # Vector additions | $2R + 2$ | $2R + 2$ |
| # Scalar multiplications | 1 | 1 |
| # Activation evaluations | $I$ | $I$ |
| Storage Complexity | | |
| # Vectors stored | $6R + 3$ | $3R + 6$ |

In contrast, collapsed Taylor mode propagates only a single summed vector

$$
\left(\sum_{r=1}^{R} \boldsymbol{h}_{2,r} = \left\langle \boldsymbol{W}_1, \sum_{r=1}^{R} \boldsymbol{x}_{2,r} \right\rangle\right) \overset{(3)}{\to} \left(\sum_{r=1}^{R} \boldsymbol{\phi}_{2,r} = \sum_{r=1}^{R} \left\langle \partial^2 \boldsymbol{\phi}(\boldsymbol{h}_0), \boldsymbol{h}_{1,r}^{\otimes 2} \right\rangle + \left\langle \partial \boldsymbol{\phi}(\boldsymbol{h}_0), \sum_{r=1}^{R} \boldsymbol{h}_{2,r} \right\rangle\right)
$$

$$
\overset{(3)}{\to} \left(\sum_{r=1}^{R} \boldsymbol{g}_{2,r} = \left\langle \boldsymbol{W}_2, \sum_{r=1}^{R} \boldsymbol{\phi}_{2,r} \right\rangle\right).
$$

This costs two matrix-vector products, $2 + 2R$ Hadamard products, $2R - 1$ vector additions, and a single scalar multiplication. Table H6 summarizes the accumulated costs.

**Error analysis.** For our numerical experiments, the result of all implementations (nested $1^{\text{st}}$-order and standard/collapsed Taylor mode) was always checked to be close. To supplement this experimental error analysis, we sketch a simplified error analysis below. We assume that there are error-prone first- and second-order inputs $\{\boldsymbol{x}_{1,r} + \boldsymbol{\varepsilon}_{1,r}\}_r$ and $\{\boldsymbol{x}_{2,r} + \boldsymbol{\varepsilon}_{2,r}\}_r$ with errors $\{\boldsymbol{\varepsilon}_{1,r}, \boldsymbol{\varepsilon}_{2,r}\}_{r=1}^{R}$ that can be seen as the error of previous propagation steps. An error-prone $\boldsymbol{x}_0$ would complicate our brief discussion too much and is ignored here. We consider again $\boldsymbol{f} = \boldsymbol{g} \circ \boldsymbol{\phi} \circ \boldsymbol{h}$. The error-influenced coefficients are denoted $\boldsymbol{g}_{2,r}^{\varepsilon}$.

Using vanilla Taylor mode, the erroneous result is

$$
\sum_{r=1}^{R} \boldsymbol{g}_{2,r}^{\varepsilon} = \sum_{r=1}^{R} \left( \left\langle \boldsymbol{W}_2, \left\langle \partial^2 \boldsymbol{\phi}(\boldsymbol{h}_0), \langle \boldsymbol{W}_1, \boldsymbol{x}_{1,r} + \boldsymbol{\varepsilon}_{1,r} \rangle^{\otimes 2} \right\rangle + \langle \partial \boldsymbol{\phi}(\boldsymbol{h}_0), \langle \boldsymbol{W}_1, \boldsymbol{x}_{2,r} + \boldsymbol{\varepsilon}_{2,r} \rangle \rangle \right\rangle \right)
$$

$$
= \sum_{r=1}^{R} \boldsymbol{g}_{2,r}
$$

$$
+ \sum_{r=1}^{R} \Big( \left\langle \boldsymbol{W}_2, \left\langle \partial^2 \boldsymbol{\phi}(\boldsymbol{h}_0), \langle \boldsymbol{W}_1, \boldsymbol{x}_{1,r} \rangle \otimes \langle \boldsymbol{W}_1, \boldsymbol{\varepsilon}_{1,r} \rangle \right\rangle \right\rangle
$$

$$
+ \left\langle \boldsymbol{W}_2, \left\langle \partial^2 \boldsymbol{\phi}(\boldsymbol{h}_0), \langle \boldsymbol{W}_1, \boldsymbol{\varepsilon}_{1,r} \rangle \otimes \langle \boldsymbol{W}_1, \boldsymbol{x}_{1,r} \rangle \right\rangle \right\rangle
$$

$$
+ \left\langle \boldsymbol{W}_2, \left\langle \partial^2 \boldsymbol{\phi}(\boldsymbol{h}_0), \langle \boldsymbol{W}_1, \boldsymbol{\varepsilon}_{1,r} \rangle^{\otimes 2} \right\rangle \right\rangle + \left\langle \boldsymbol{W}_2, \langle \partial \boldsymbol{\phi}(\boldsymbol{h}_0), \langle \boldsymbol{W}_1, \boldsymbol{\varepsilon}_{2,r} \rangle \rangle \right\rangle \Big)
$$

$$
= \sum_{r=1}^{R} \boldsymbol{g}_{2,r} + \Delta \boldsymbol{g}_{2,R}^{S} + \sum_{r=1}^{R} \langle \boldsymbol{W}_2 \langle \partial \boldsymbol{\phi}(\boldsymbol{h}_0), \langle \boldsymbol{W}_1, \boldsymbol{\varepsilon}_{2,r} \rangle \rangle \rangle .
$$

All errors related to the first-order coefficients are summarized in

$$
\Delta \boldsymbol{g}_{2,R}^{S} := \sum_{r=1}^{R} \Big( \left\langle \boldsymbol{W}_2, \left\langle \partial^2 \boldsymbol{\phi}(\boldsymbol{h}_0), \langle \boldsymbol{W}_1, \boldsymbol{x}_{1,r} \rangle \otimes \langle \boldsymbol{W}_1, \boldsymbol{\varepsilon}_{1,r} \rangle \right\rangle \right\rangle
$$

$$
+ \left\langle \boldsymbol{W}_2, \left\langle \partial^2 \boldsymbol{\phi}(\boldsymbol{h}_0), \langle \boldsymbol{W}_1, \boldsymbol{\varepsilon}_{1,r} \rangle \otimes \langle \boldsymbol{W}_1, \boldsymbol{x}_{1,r} \rangle \right\rangle \right\rangle
$$

$$
+ \left\langle \boldsymbol{W}_2, \left\langle \partial^2 \boldsymbol{\phi}(\boldsymbol{h}_0), \langle \boldsymbol{W}_1, \boldsymbol{\varepsilon}_{1,r} \rangle^{\otimes 2} \right\rangle \right\rangle \Big) .
$$

The collapsed Taylor mode results in

$$
\sum_{r=1}^{R} \boldsymbol{g}_{2,r}^{\varepsilon} = \left\langle \boldsymbol{W}_2, \sum_{r=1}^{R} \left( \left\langle \partial^2 \boldsymbol{\phi}(\boldsymbol{h}_0), \langle \boldsymbol{W}_1, \boldsymbol{x}_{1,r} + \boldsymbol{\varepsilon}_{1,r} \rangle^{\otimes 2} \right\rangle \right) \right\rangle
$$

$$
+ \left\langle \boldsymbol{W}_2, \left\langle \partial \boldsymbol{\phi}(\boldsymbol{h}_0), \left\langle \boldsymbol{W}_1, \sum_{r=1}^{R} (\boldsymbol{x}_{2,r} + \boldsymbol{\varepsilon}_{2,r}) \right\rangle \right\rangle \right\rangle
$$

$$
\begin{aligned}
=\;& \sum_{r=1}^{R} \boldsymbol{g}_{2,r} \\
&+ \Big\langle \boldsymbol{W}_2, \sum_{r=1}^{R} \Big( \langle \partial^2 \phi(\boldsymbol{h}_0), \langle \boldsymbol{W}_1, \boldsymbol{x}_{1,r} \rangle \otimes \langle \boldsymbol{W}_1, \boldsymbol{\varepsilon}_{1,r} \rangle \rangle + \langle \partial^2 \phi(\boldsymbol{h}_0), \langle \boldsymbol{W}_1, \boldsymbol{\varepsilon}_{1,r} \rangle \otimes \langle \boldsymbol{W}_1, \boldsymbol{x}_{1,r} \rangle \rangle \\
&\qquad\qquad + \langle \partial^2 \phi(\boldsymbol{h}_0), \langle \boldsymbol{W}_1, \boldsymbol{\varepsilon}_{1,r} \rangle^{\otimes 2} \rangle \Big) \Big\rangle \\
&+ \Big\langle \boldsymbol{W}_2, \Big\langle \partial \phi(\boldsymbol{h}_0), \Big\langle \boldsymbol{W}_1, \sum_{r=1}^{R} \boldsymbol{\varepsilon}_{2,r} \Big\rangle \Big\rangle \Big\rangle \\
=\;& \sum_{r=1}^{R} \boldsymbol{g}_{2,r} + \Delta \boldsymbol{g}_{2,R}^{C} + \Big\langle \boldsymbol{W}_2, \Big\langle \partial \phi(\boldsymbol{h}_0), \Big\langle \boldsymbol{W}_1, \sum_{r=1}^{R} \boldsymbol{\varepsilon}_{2,r} \Big\rangle \Big\rangle \Big\rangle,
\end{aligned}
$$

where the first-order errors are collected in

$$
\begin{aligned}
\Delta \boldsymbol{g}_{2,R}^{C} := \Big\langle \boldsymbol{W}_2, \sum_{r=1}^{R} \Big( &\langle \partial^2 \phi(\boldsymbol{h}_0), \langle \boldsymbol{W}_1, \boldsymbol{x}_{1,r} \rangle \otimes \langle \boldsymbol{W}_1, \boldsymbol{\varepsilon}_{1,r} \rangle \rangle \\
&+ \langle \partial^2 \phi(\boldsymbol{h}_0), \langle \boldsymbol{W}_1, \boldsymbol{\varepsilon}_{1,r} \rangle \otimes \langle \boldsymbol{W}_1, \boldsymbol{x}_{1,r} \rangle \rangle + \langle \partial^2 \phi(\boldsymbol{h}_0), \langle \boldsymbol{W}_1, \boldsymbol{\varepsilon}_{1,r} \rangle^{\otimes 2} \rangle \Big) \Big\rangle .
\end{aligned}
$$

**Error analysis (summary and discussion).** Without considering floating-point operations, the errors are equivalent. This is not surprising, since our collapsing method is mathematically equivalent to the standard Taylor mode on the same input coefficients.

Incorporating floating-point operations for the function evaluations, inner product, tensor product, and summations would greatly complicate the discussion, which is not part of the paper. Still, the error could be split into the same three parts for both vanilla and collapsed Taylor mode. For the first-order errors $\Delta \boldsymbol{g}_{2,R}^{S}$ and $\Delta \boldsymbol{g}_{2,R}^{C}$, however, even with floating-point operations, the errors are structurally similar, since apart from the most outer inner product (with $\boldsymbol{W}_2$) and the summation, all operations are done in the same order. In practice, we would expect smaller errors for the collapsing method due to the reduced number of operations. The second error term, which collects the error of the second-order coefficients, could also reduce the accumulation of error terms. Of course, the actual condition and input, and output dimensions of the matrices are crucial. Theoretically, this could even lead to a similar error asymptotically. If inputs are small, one could argue that catastrophic cancellations are more likely to happen in our case, since we sum first. But note that those cancellations are then also likely to happen in the standard Taylor mode, because weight matrices are often normalized, and the outputs of the activation functions are small if the input is small.

We plan to investigate this more rigorously in the future.

# I Connections of Collapsed Taylor Mode to Existing Methods

Here, we make the connection of collapsed Taylor mode to the forward Laplacian [21] and the randomized estimation of the Laplacian via Hutchinson's trace estimator [18] from [27] explicit.

## I.1 Connection to Randomized Laplacian via Hutchinson's Trace Estimator

For simplicity, we consider a vector-to-scalar function $f : \mathbb{R}^D \to \mathbb{R}, \boldsymbol{x} \mapsto f(\boldsymbol{x})$ (the general vector-to-vector case is straightforward but requires more notation) whose Laplacian is

$$
\Delta f(\boldsymbol{x}) = \mathrm{Tr}(\nabla^2 f(\boldsymbol{x})) = \sum_{d=1}^{D} [\nabla^2 f(\boldsymbol{x})]_{d,d} = \sum_{d=1}^{D} \boldsymbol{e}_d^\top \nabla^2 f(\boldsymbol{x}) \boldsymbol{e}_d ,
$$

with $\nabla^2 f(\boldsymbol{x}) \in \mathbb{R}^{D \times D}$ the Hessian of $f$ evaluated at $\boldsymbol{x}$. Because the Laplacian can be expressed as trace of the Hessian, we can use Hutchinson's trace estimator [18] to estimate it via Hessian-vector products with random vectors. Specifically, for any matrix $\boldsymbol{A} \in \mathbb{R}^{D \times D}$ and a distribution $p(\mathbf{v})$ over a

vector $\mathbf{v}$ with unit covariance ($\mathbb{E}[\mathbf{vv}^\top] = \boldsymbol{I}_D$) we can use the cyclic property of the trace and linearity of the expectation to write

$$\text{Tr}(\boldsymbol{A}) = \text{Tr}(\boldsymbol{A}\boldsymbol{I}_D) = \text{Tr}(\boldsymbol{A}\mathbb{E}[\mathbf{vv}^\top]) = \mathbb{E}[\text{Tr}(\boldsymbol{A}\mathbf{vv}^\top)] = \mathbb{E}[\text{Tr}(\mathbf{v}^\top \boldsymbol{A}\mathbf{v})].$$

Then, we can compute an unbiased estimate of the right hand side by drawing $S$ vectors $\boldsymbol{v}_1, \boldsymbol{v}_2, \ldots, \boldsymbol{v}_S \sim p(\mathbf{v})$ and evaluating the Monte-Carlo estimator

$$\text{Tr}(\boldsymbol{A}) \approx \frac{1}{S} \sum_{s=1}^{S} \boldsymbol{v}_s^\top \boldsymbol{A}\boldsymbol{v}_s.$$

Applied to the Hessian, we can estimate the Laplacian of $f$ as

$$\Delta f(\boldsymbol{x}) \approx \frac{1}{S} \sum_{s=1}^{S} \boldsymbol{v}_s^\top \nabla^2 f(\boldsymbol{x})\boldsymbol{v}_s.$$

Using our tensor notation, we can rewrite this into a sum of terms involving $\langle \partial^2 f(\boldsymbol{x}), \boldsymbol{v}_s^{\otimes 2} \rangle$, which can be computed with **vanilla Taylor mode** using $S$ 2-jets (see eq. (4)):

$$= \frac{1}{S} \sum_{s=1}^{S} \sum_{i,j=1}^{D} [\partial^2 f(\boldsymbol{x})]_{i,j} [\boldsymbol{v}_s]_i [\boldsymbol{v}_s]_j = \frac{1}{S} \sum_{s=1}^{S} \langle \partial^2 f(\boldsymbol{x}), \boldsymbol{v}_s^{\otimes 2} \rangle.$$

Instead of propagating then summing the 2-jets, we can also sum the vectors and then propagate the sum (assuming we have a composition, see eq. (D16)), which is our proposed **collapsed Taylor mode**:

$$= \frac{1}{S} \left\langle \partial^2 f(\boldsymbol{x}), \sum_{s=1}^{S} \boldsymbol{v}_s^{\otimes 2} \right\rangle.$$

### I.2 Connection to the Forward Laplacian

We start by writing out the propagation rules of the forward Laplacian (eqs. (5-7) in Li et al. [21]) for a function $f = g \circ h$ with $g : \mathbb{R}^C \to \mathbb{R}$ whose input we denote by $\boldsymbol{h}_0 \in \mathbb{R}^C$. The forward propagation consumes $\boldsymbol{h}_0 = \boldsymbol{h}(\boldsymbol{x}_0)$, the Jacobian $\nabla_{\boldsymbol{x}_0} \boldsymbol{h}_0 = \nabla_{\boldsymbol{x}_0} \boldsymbol{h}(\boldsymbol{x}_0) \in \mathbb{R}^{D \times C}$, and the Laplacian $\Delta_{\boldsymbol{x}_0} \boldsymbol{h}_0 = \Delta_{\boldsymbol{x}_0} \boldsymbol{h}(\boldsymbol{x}_0) \in \mathbb{R}^C$:

$$\begin{pmatrix} \boldsymbol{h}_0 & \in \mathbb{R}^C \\ \nabla_{\boldsymbol{x}_0} \boldsymbol{h}_0 & \in \mathbb{R}^{D \times C} \\ \Delta_{\boldsymbol{x}_0} \boldsymbol{h}_0 & \in \mathbb{R}^C \end{pmatrix} \to \begin{pmatrix} g_0 = g(\boldsymbol{h}_0) & \in \mathbb{R} \\ \nabla_{\boldsymbol{x}_0} g_0 = (\nabla_{\boldsymbol{x}_0} \boldsymbol{h}_0)(\nabla_{\boldsymbol{h}_0} g_0) & \in \mathbb{R}^D \\ \Delta_{\boldsymbol{x}_0} g_0 = (\nabla_{\boldsymbol{h}_0} g_0)^\top \Delta \boldsymbol{h}_0 + \text{Tr}\left((\nabla_{\boldsymbol{x}_0} \boldsymbol{h}_0)^\top (\nabla_{\boldsymbol{x}_0} \boldsymbol{h}_0) \nabla_{\boldsymbol{h}_0} g_0\right) & \in \mathbb{R} \end{pmatrix}.$$

Let us rewrite this in terms of rows of the Jacobian $[\nabla_{\boldsymbol{x}_0} \boldsymbol{h}_0]_{d,:} \in \mathbb{R}^C$ (where the colon subscript denotes a slice):

$$\begin{pmatrix} \boldsymbol{h}_0 \\ \{[\nabla_{\boldsymbol{x}_0} \boldsymbol{h}_0]_{d,:}\}_{d=1}^{D} \\ \Delta_{\boldsymbol{x}_0} \boldsymbol{h}_0 \end{pmatrix} \to \begin{pmatrix} g_0 = g(\boldsymbol{h}_0) \\ \{[\nabla_{\boldsymbol{x}_0} g_0]_{d,:} = [\nabla_{\boldsymbol{x}_0} \boldsymbol{h}_0]_{d,:}(\nabla_{\boldsymbol{h}_0} g_0)\}_{d=1}^{D} \\ \Delta_{\boldsymbol{x}_0} g_0 = (\nabla_{\boldsymbol{h}_0} g_0)^\top \Delta \boldsymbol{h}_0 + \sum_{d=1}^{D} [\nabla_{\boldsymbol{x}_0} \boldsymbol{h}_0]_{d,:} \nabla_{\boldsymbol{h}_0}^2 g_0 [\nabla_{\boldsymbol{x}_0} \boldsymbol{h}_0]_{d,:}^\top \end{pmatrix}.$$

In our tensor notation, this translates to

$$\begin{pmatrix} \boldsymbol{h}_0 \\ \{[\nabla_{\boldsymbol{x}_0} \boldsymbol{h}_0]_{d,:}\}_{d=1}^{D} \\ \Delta_{\boldsymbol{x}_0} \boldsymbol{h}_0 \end{pmatrix} \to \begin{pmatrix} g_0 = g(\boldsymbol{h}_0) \\ \{[\nabla_{\boldsymbol{x}_0} g_0]_{d,:} = \langle \partial g(\boldsymbol{h}_0), [\nabla_{\boldsymbol{x}_0} \boldsymbol{h}_0]_{d,:} \rangle\}_{d=1}^{D} \\ \Delta_{\boldsymbol{x}_0} g_0 = \langle \partial g(\boldsymbol{h}_0), \Delta \boldsymbol{h}_0 \rangle + \sum_{d=1}^{D} \langle \partial g(\boldsymbol{h}_0), [\nabla_{\boldsymbol{x}_0} \boldsymbol{h}_0]_{d,:}^{\otimes 2} \rangle \end{pmatrix}.$$

To obtain the connection to Taylor mode, we define $[\nabla_{\boldsymbol{x}_0} \boldsymbol{h}_0]_{d,:} = \boldsymbol{h}_{1,d}$ and $\Delta_{\boldsymbol{x}_0} \boldsymbol{h}_0 = \sum_d \boldsymbol{h}_{2,d}$ and $[\nabla_{\boldsymbol{x}_0} g_0]_{d,:} = g_{1,d}$ and $\Delta_{\boldsymbol{x}_0} g_0 = \sum_d g_{2,d}$, which allows us to rewrite the forward Laplacian as

$$\begin{pmatrix} \boldsymbol{h}_0 \\ \{\boldsymbol{h}_{1,d}\}_{d=1}^{D} \\ \sum_d \boldsymbol{h}_{2,d} \end{pmatrix} \to \begin{pmatrix} g_0 = g(\boldsymbol{h}_0) \\ \{g_{1,d} = \langle \partial g(\boldsymbol{h}_0), \boldsymbol{h}_{1,d} \rangle\}_{d=1}^{D} \\ \sum_d g_{2,d} = \langle \partial g(\boldsymbol{h}_0), \sum_d \boldsymbol{h}_{2,d} \rangle + \sum_{d=1}^{D} \langle \partial g(\boldsymbol{h}_0), \boldsymbol{h}_{1,d}^{\otimes 2} \rangle \end{pmatrix}.$$

This yields our collapsed Taylor mode propagation: the first equation is simply the forward pass, the second equation propagates the first-order derivatives along $D$ directions, and the last equation propagates the collapsed second-order derivatives, as described by setting $K = 2$ in Equation (6).

