# OpenReview forum: "Collapsing Taylor Mode Automatic Differentiation"
_NeurIPS.cc/2025/Conference — NeurIPS 2025 poster_

### Official Review · Reviewer_GZnh · 2025-06-17

**Clarity:** 3
**Significance:** 4
**Originality:** 4
**Rating:** 5
**Confidence:** 4

**Summary:**

Motivated by the evaluation of PDE operators, this paper introduces a collapsed Taylor mode forward algorithmic differentiation differentiation. The authors justify a form of chain rule for certain differential operators, represented as sums of high order derivatives, generalizing the Laplacian. The main idea is to propagate sums rather than high order tensors to be summed. This allows to avoid propagating the full highest order tensor and results in significant savings in terms of the number of arithmetic operations required to evaluate the operator.

The method is implemented in pytorch and tested on three differential operators. The authors report consistent improvements over the baseline nested AD both in terms of computation time and peak memory. The improvements are coherent with the arithmetic complexity estimates and the figures are confirmed using jax.

This is a convincing paper which adresses a relevant question and proposes an advanced automatic differentiation method which will likely reduce the training time of PINNs in the near future.

**Questions:**

(11), what does $[\mathbf{j}]_i$ mean? It seems that the notation is not self contained in this respect. I interpret it as just a number, is it correct?

Related to the previous question, please indicate which part of [14] is refered to in Section 3.3. In its present form, the submission requires to read another full paper, with different notations.

In (12), the outer sum is not pulled inside. I guess this is because of non-linearity of the tensor product. Any hope to pull this inside?

In the present form (12) should be re-implemented for each setting? Could this be automatized as well?

Could the authors provide a minimal working code example to reproduce some experiments? This would strengthen the submission.

Line 89: "composing f of atomic functions with known derivatives and the chain rule" what does it mean?

Does Figure 4 contain all coefficients?

**Ethical Concerns:**

["NO or VERY MINOR ethics concerns only"]

**Final Justification:**

See the "comment after rebutal".

**Limitations:**

Yes

**Quality:**

3

**Strengths And Weaknesses:**

### Strength

The reported improvements are very clean, consistent and convincing.

The proposed scheme can be directly combined with the stochastic estimator proposed in [26].

The authors suggest that the proposed approach could be implemented with minimal modifications of existing Taylor mode AD frameworks, at least from the user point of view.


### Weaknesses
While the authors propose a solution for general differential operators in (10), the proposed solution is only illustrated using the biharmonic operator. The paper would benefit from more such examples.

While equation (12) provides a one-fit-all solution for problems of the form (10), it requires to re-implement the equation in (12) for every new example.

The code is not part of the submission. The paper mentions that this will be open sourced, but the timeline and the actual content of this future library cannot be evaluated from the submission.

---

> ### Author Rebuttal · Authors · 2025-07-30
>
> Dear Reviewer GZnh,
>
> thanks for your strong support and constructive feedback.
> Please find our answers below and let us know if you have any follow-up questions.
>
> ---
> > (11), what does $[\mathbf{j}]_i$ mean? It seems that the notation is not self contained in this respect. I interpret it as just a number, is it correct?
>
> Thank you for pointing this out. As explained in the background section, we use square brackets to denote slicing.
> We will revise the text to clarify the meaning of $[\mathbf{j}]_i$ as $i$th entry of vector $\mathbf{j}$ and ensure that the notation is clearly defined.
>
> ---
>
> > The paper would benefit from more such examples.
>
> We agree that the interpolation formula we use to present a fully-general approach can be daunting to parse at first.
>
> To make this process more clear we further add an example in the appendix: computing $\sum_{i=1}^D\sum_{j=1}^D\frac{\partial^3}{\partial x_i^2 x_j}f(x)$.
> This example is inspired by Appendix F.1 of the stochastic Taylor derivative paper (Shi et al., 2024), which describes how to compute each mixed 3rd-order derivative using either one 7-jet, two 5-jets, or three 4-jets.
> Let us know if you would like to see all the details and we will follow up on this in a separate message.
>
> **TL;DR:** We show that computing this operator with the interpolation formula uses $D^2$ **3-jets (rather than multiples of $D^2$ jets of order $>3$)** that can also be collapsed.
> We expect the interpolation formula to be favourable as Taylor mode scales quadratically in the derivative order, and we will experimentally verify that in a future version of the paper.
>
> ---
>
> > In the present form (12) should be re-implemented for each setting? Could this be automatized as well?
>
> You are right that Equation (12) must currently be re-implemented as it is the equation that interfaces with the (automated) Taylor mode and the (automated) collapsing.
>
> Two things which are currently not automated are:
>
> 1. We automated computing the coefficients $\gamma_{\mathbf{i},\mathbf{j}}$, but did not design an interface that allows to specify a PDE operator by specifying the multi-index $\mathbf{i} = (i_1, \dots, i_I)$, i.e. the frequency of the directions. We believe this is possible.
> 2. Using symmetries in the multi-index $\mathbf{i}$ to simplify the number of jets. We believe fully automating this is non-trivial as it requires symbolic understanding of the math.
>
> We will make sure to mention these aspects in the next revision.
>
> ---
>
> > In (12), the outer sum is not pulled inside. [...] Any hope to pull this inside?
>
> Yes, because Equation (12) has the same structure as Equation (5) for which we show how to perform the collapsing (pull the sum inside) in detail.
> We will add the collapsed version of Equation (12) to the next revision, as we did not have space for it.
>
> ---
>
> > Could the authors provide a minimal working code example [...]?
>
> Here is a minimal example (imports omitted) how to work with our code base to compute (collapsed) Laplacians.
> We hope it provides further clarification.
>
> ```python
> # 1) Define the function whose Laplacian we want to compute
> def f(x: Tensor) -> Tensor:
>     return sin(x) # could also be a neural net
>
> # 2) Apply Taylor-mode to compute 2-jets
> jet2_f = jet(f, k=2) # signature: (Tensor, Tensor, Tensor) -> (Tensor, Tensor, Tensor)
>
> # 3) Vectorize to compute multiple 2-jets in parallel
> multijet2_f = vmap(jet2_f) # signature: (BatchedTensor, BatchedTensor, BatchedTensor) -> (BatchedTensor, BatchedTensor, BatchedTensor)
>
> # 4) Use the multi-2-jet to compute the Laplacian
> def laplacian_f(x: BatchedTensor) -> Tensor:
>     x0, x1, x2 = x, eye(...), zeros(...) # Equation (7b) in the paper
>     f0, f1, f2 = multijet2_f(x0, x1, x2)
>     return sum(f2, dim=0)
>
> # 5) Rewrite the compute graph (collapse)
> collapsed_laplacian_f = simplify(laplacian_f)
>
> # ---
>
> # Sanity check
> x = rand(...)
> vanilla = laplacian_f(x)
> collapsed = collapsed_laplacian_f(x) # same result, but less time and memory
> assert vanilla.allclose(collapsed)
> ```
>
> ---
>
> > Line 89: "composing f of atomic functions with known derivatives and the chain rule" what does it mean?
>
> By this, we mean that implementing Taylor mode requires specifying the Faa-di-Bruno formula for basic operations (e.g. `+`, `sin`, `tanh`, ...) which users can then use to programmatically specify functions whose jets should be computed.
> This is a common building principle for AD and makes it extensible, as new operations can be added one-by-one.
>
> ---
>
> > Does Figure 4 contain all coefficients?
>
> Yes. We will describe in more detail in the appendix how to manually compute these coefficients (our code also ships with a utility function to compute the interpolation coefficents $\gamma_{\mathbf{i},\mathbf{j}}$).
>
> Thanks again for your time and effort!

---

> > ### Comment · Reviewer_GZnh · 2025-08-01
> > **Response after rebuttal**
> >
> > I have read the authors comments and they are very satisfactory, thank you.
> > Very minor if you could just precise a bit which part of [14] is used in Section 3.3, this would be very helpful.

---

> > > ### Author Response · Authors · 2025-08-01
> > >
> > > Thanks a lot for getting back to us!
> > >
> > > The interpolation formula we use in Section 3.3 (our Equation (11)) corresponds to Proposition 4.1 in Reference 14.

---

### Official Review · Reviewer_SLnK · 2025-06-24

**Clarity:** 3
**Significance:** 2
**Originality:** 3
**Rating:** 3
**Confidence:** 3

**Summary:**

The paper introduces a method to improve the efficiency of computing higher-order derivatives using Taylor-mode automatic differentiation (AD). The key idea is a graph-rewrite rule that exploits the linearity of highest-order derivative terms, enabling a collapsed representation that reduces memory usage and computational cost. Specifically, it is observed that in Taylor-mode AD, the propagation of the highest-degree coefficient is linear in the input, which allows the summation over multiple directions (used in computing operators like Laplacians) to be moved inside the AD graph. As a result, the system computes only one higher-order jet instead of many, saving time and memory.
The method is implemented as a graph-level transformation, and the rewrite rule identifies when a Taylor jet can be collapsed based on the operator structure. The  experimental results demonstrate improvements in speed and memory saving.

**Questions:**

Suggestions

1. Add a formal analysis of time and space complexity that would clarify the cost savings to guide on when the collapsing method pays off.
2. Broaden the experimental benchmarks to demonstrate robustness, scalability, and generality beyond toy MLPs.
3. Compare with Reverse-Mode and Mixed-Mode AD.
4.  Since high-order derivatives are often sensitive to numerical errors,  include a numerical stability analysis and show that collapsing is not unstable.

**Ethical Concerns:**

["NO or VERY MINOR ethics concerns only"]

**Final Justification:**

Following the discussions, I raised my score to 4, I believe that the paper should be accepted.

**Limitations:**

Yes

**Quality:**

2

**Strengths And Weaknesses:**

Strengths
1. Clear theoretical insight: The observation that in Faà di Bruno’s formula, the highest-order Taylor term is linear in its inputs leads to a simple graph-rewrite rule that reduces significantly the number of propagated jets.
2.  A substantial computational savings is demonstrated in speedups and memory reductions.
3. The collapsing rule is general-purpose, applicable to multiple higher-order operators, which makes it broadly applicable across SciML workflows.
4. Simple integration with existing code, resulting in a minimal barrier to adoption.


Weaknesses
1. The collapsing method only applies when the higher-order operator being computed is a linear functional of the highest-order Taylor coefficient, hence it accelerates a narrow class of derivative computations.
2. The speedups are measured empirically and the complexity reductions are not quantified formally, hence it lacks guarantees or limits on where collapsing breaks down or ceases to be beneficial.
3. All benchmarks are performed on small MLPs with synthetic data, and there is no demonstration of  scalability or broad utility.
4. The paper focuses on forward Taylor mode and compares primarily against unoptimized nested AD and not reverse mode or hybrid AD, thus it is not clear that this is the best method overall for higher-order AD.
5. The paper does not analyze how collapsing affects numerical precision. Combining Taylor coefficients before propagation could change round-off behavior or accumulate truncation errors in high-order derivatives.

---

> ### Author Rebuttal · Authors · 2025-07-30
>
> Dear Reviewer SLnK,
>
> Thanks for your comments and thoughtful feedback.
> In the following, we answer to your raised concerns.
>
> ---
>
> > accelerates a narrow class of derivative computations.
>
> We respectfully disagree.
> While our method indeed relies on the assumption that the PDE operator is a linear functional of the highest-order Taylor coefficient, this structural requirement is satisfied by many important PDE operators:
>
> 1. **The Laplacian.** Among its diverse applications, we stress the importance of computing Laplacians in high dimensions in quantum physics. This is a major bottleneck in the variational Monte Carlo method for the Schroedinger equation.
> 2. **Weighted Laplacians.** These govern the diffusive terms in Fokker–Planck/Kolmogorov-type PDEs. Again, these are PDE operators in high dimensions and efficient schemes to compute them are paramount.
> 3. **Higher-order elliptic operators:** This includes the biharmonic operator, which arises in elasticity and thin plate theory.
>
> As these examples span diverse application domains, and efficient derivative computation is important for all of them, we do not consider our proposed methodology's scope to be narrow.
>
> Clearly, PDE operators of different structure are worthwhile to explore as well.
> We think our scheme can also apply to more PDE operators but leave this for future work as it will require more mathematical formalization.
>
> ---
>
> > The speedups are measured empirically and the complexity reductions are not quantified formally
>
> > Add a formal analysis of time and space complexity that would clarify the cost savings to guide on when the collapsing method pays off.
>
> Thank you for your insightful comment. You are right that a formal complexity analysis would strengthen the presentation. We will incorporate your suggestion in the following way (we make the below argument concrete for a linear layer in our response to Reviewer rNn6):
>
> The key component for our analysis is computing sums of $K$th-order derivatives
> $$\sum_{r=1}^R \langle \partial^K f(\mathbf{x}), \mathbf{v}_r^{\otimes K}\rangle
> $$ via Taylor mode, which must also compute the derivatives of lower degree.
>
> **Time complexity:** The cost of computing the lower-order coefficients ($0$ to $K - 1$) is identical for both collapsed and vanilla Taylor mode. The key difference lies in the highest coefficients ($K$th order), which can be directly understood from the chain rule:
> - In **vanilla Taylor mode**, for each of the $R$ directions, we compute at each layer
> $$
> \sum_{\sigma \in \text{part($K$)}\setminus \\{\sigma_t\\}} \nu(\sigma)\langle\partial^K g(h_{0}), \otimes_{s \in \sigma}h_{s, r} \rangle + \langle \partial g(h_0), h_{K, r}\rangle
> $$ followed by a vector summation over $r$ after the final layer.
> - In **collapsed Taylor mode** we instead compute *once* at each layer
> $$
> \sum_{r=1}^R\left(\sum_{\sigma \in \text{part($K$)}\setminus \\{\sigma_t\\}} \nu(\sigma)\langle\partial^K g(h_{0}), \otimes_{s \in \sigma}h_{s, r} \rangle\right) + \langle \partial g(h_0), \sum_{r=1}^Rh_{K, r}\rangle
> $$
> Note that the summation inside the second term is performed in the previous layer.
>
> **So in total, collapsed Taylor mode trades $R - 1$ Jacobian-vector products (matrix-vector multiplications!) for $R$ vector summations at each layer**.
>
> **Space Complexity:** The space complexity is similarly reduced. **Vanilla Taylor mode** stores the highest coefficient for every direction at each layer ($g_{K, r}$ for $r=1, \dots, R$ in our example). Instead, **collapsed Taylor mode** stores the summed coefficient ($\sum_{r} g_{K, r}$) at each layer.
>
> **This reduces space complexity from storing $R$ $K$th-order coefficients to storing $1$ $K$th-order coefficient at each layer.**
>
> ---
>
> > All benchmarks are performed on small MLPs with synthetic data, and there is no demonstration of scalability or broad utility.
>
> **MLPs are common in the PINN literature**, which is one of the prime application fields for our developed method, explaining our choice and rendering the considered networks meaningful.
>
> Based on our theory, we expect the performance difference between vanilla and collapsed Taylor mode to carry over to a wide array of other architectures.
> **The methodology is completely architecture-agnostic.**
>
> If you have suggestions for additional architectures we should show in the paper, do let us know and we would be happy to try to include them in a future revision.
> However, this will likely require supporting new operations in our Taylor mode library, and we might not be able to provide these results during the discussion.
>
> ---
>
> > compares primarily against unoptimized nested AD and not reverse mode or hybrid AD, thus it is not clear that this is the best method overall for higher-order AD.
>
> Could you elaborate what you mean by hybrid AD?
>
> We use forward-over-reverse mode for Hessian-vector products, which is the recommended way in PyTorch (see e.g. the docstring of `torch.func.hessian`).
> Other modes like reverse-over-reverse are less favourable and there exist empirical (Dagreou et al., 2023) and theoretical (Griewank et al., 2008) arguments for that.
>
> We are not aware of any AD implementation for Python that allows going beyond combining forward and reverse mode.
> Therefore, **we believe that our 'nested first-order AD' baseline is the best currently achievable implementation in PyTorch.**
>
> ---
>
> > The paper does not analyze how collapsing affects numerical precision.
>
> > include a numerical stability analysis and show that collapsing is not unstable.
>
> We appreciate your request to address numerical stability and would like to answer it from a theoretical and empirical perspective:
>
> - **Theoretical:** While we are unaware of existing stability proofs for higher‑order Taylor mode AD, reference [1] showed **forward- and backward‑stability for first‑order AD**. Since the basic structure for propagating derivatives of any order remains the same, **we are strongly convinced that numerical stability can be proven also for the vanilla and the collapsed Taylor mode**. However, this is a very technical/mechanical task and we will not be able to provide general results during the discussion. To illustrate the idea for a corresponding proof, we will add a small example to the paper and highlight that collapsed Taylor mode is likely to reduce rounding errors due to its smaller number of floating point operations.
> - **Empirical:** In our experiments, we verify consistency between all implementations (nested first-order AD versus vanilla Taylor versus collapsed Taylor) by comparing the results  with `torch.allclose` with default arguments (`rtol=1e-5, atol=1e-8`).
> We noticed **no discrepancies when computing in `float64`** (which is standard for applications like PINNs).
>
> We hope this addresses your concern. Please don’t hesitate to let us know if further clarification would be helpful.
>
> Thanks again for your time and effort!
>
> 1. A. Griewank, K. Kulshreshtha und A. Walther: On the numerical stability of algorithmic differentiation. Computing 94:125--149 (2012)

---

> ### Comment · Reviewer_SLnK · 2025-08-02
>
> Thanks for the reply and the changes that will be made. I raise the score to to 4.

---

### Official Review · Reviewer_zJuU · 2025-06-30

**Clarity:** 3
**Significance:** 3
**Originality:** 3
**Rating:** 5
**Confidence:** 3

**Summary:**

This paper proposes an optimized method to compute higher-order differential operators with Taylor-mode automatic differentiation. While typical automatic differentiation propagates only first-order derivatives (which requires nesting for higher orders), Taylor mode propagates a tuple of $K$ derivatives (a so-called “jet”). Leveraging the linearity of common PDE operators like the Laplacian, the authors show that the highest derivative term in the jet can be compressed when multiple perturbations are propagated simultaneously. This yields a theoretical speedup of roughly $K/(K-1)$, which is significant especially for low differentiation order $K$.
As a side product, this method recovers previously-suggested optimizations, like the forward Laplacian, while being easy to automate. The authors demonstrate it with a PyTorch implementation based on computational graph rewriting, whose performance compares favorably to nested first order and standard Taylor mode.

**Questions:**

Here are the main additions which would improve the paper:

- Explain the connection between collapsed Taylor mode and the individual optimizations it generalizes (especially the forward Laplacian).
- Provide the experimental code to the reviewer, mainly assess the reusability of the underlying software for a wider audience (which will determine impact).
- Explain why collapsed Taylor mode can lead to increased memory use for the biharmonic operator.

Here are suggestions related to the presentation:

- Since the key ingredient for collapsing to work is linearity in the last term of the Faa di Bruno formula, might it improve readability to push these details to the appendix and instead go with a simpified version in the main text? Maybe something like “the $K$\-th derivative of $g \circ h $ is $\partial g$ times the $K$\-th derivative of $h$ plus other lower-order terms in $h$”? I’m not sure this is a relevant suggestion to be honest.
- In Equation (12), shouldn’t the two red sums be inside the inner product?
- In Figure 5, what are “non-differentiable computations”, and should “opaque” be replaced by “transparent”?

**Ethical Concerns:**

["NO or VERY MINOR ethics concerns only"]

**Final Justification:**

My score was already high and the rebuttal was entirely satisfactory, so I will keep it as is.

**Limitations:**

Yes.

**Paper Formatting Concerns:**

None.

**Quality:**

4

**Strengths And Weaknesses:**

## Strengths

### Quality

- The suggested approach is technically sound and insightful.
- Experimental results on two differential operators support the use of collapsed Taylor mode, both for exact and stochastic computations.

### Clarity

- The paper demonstrates real effort to make theory and algorithms accessible, with the use of colors and diagrams to emphasize the salient points.

### Significance

- I am especially enthusiastic about the PyTorch library which automates graph rewriting based on torch.fx. Although the main contribution of the paper is algorithmic in nature, I assume that the open-sourcing of said library will be the actual driver of adoption for this otherwise complex method.

### Originality

- Collapsed Taylor mode gives a more generic perspective on previous isolated optimizations like the forward Laplacian, which is important for our understanding of automatic differentiation’s complexity.

## Weaknesses

### Quality

- The main missing aspect is an explanation of how collapsed Taylor mode ties into existing methods like the forward Laplacian or the Hutchinson trace estimator. A short introduction to both of these techniques and a proof of the connection with collapsed Taylor mode would make the paper complete, but appendix D does not contain them.

### Clarity

- Through no fault of the authors, the entry cost into this paper is rather high due to the combination of tensor calculus and high-order derivatives with the Faa di Bruno formula. Perhaps some notational or exposition tricks could be used to facilitate the reader’s task (see below).

### Significance

- Without access to the code of the graph rewriting utility during review, it is hard to assess how generic and reusable this method will be for people unfamiliar with the underlying mathematics or implementation tricks.

---

> ### Author Rebuttal · Authors · 2025-07-30
>
> Dear Reviewer zJuU,
>
> thanks a lot for your strong support and constructive feedback!
> Please find our responses below and let us know if you have follow-up questions.
> We would be happy to discuss.
>
> ---
>
> > missing [...] explanation of how collapsed Taylor mode ties into existing methods like the forward Laplacian or the Hutchinson trace estimator.
>
> > Explain the connection between collapsed Taylor mode and the individual optimizations it generalizes (especially the forward Laplacian).
>
> Thanks for pointing this out.
> We agree that making the connections of our method to existing approaches clear is essential for it to be adopted by the community.
> Due to the space limitations, these connections are currently hard to reconstruct because we frame everything in our tensor notation which focuses on generality.
>
> To address your concern, **we will add a self-contained appendix that starts with equations** (5-7) from the forward Laplacian paper **and gradually transforms them into our tensor notation to show their equivalence** with collapsed Taylor mode.
> We will do the same for the Hutchinson trace estimator, starting from its formulation via vector-Hessian-vector products.
>
> Let us know if this makes sense to you and whether you want all the details, and we will send a separate message about this.
>
> ---
>
> > it is hard to assess how generic and reusable this method will be
>
> To answer this, we summarize the core components of our code from a user and developer perspective (in our response to Reviewer GZnh, we also include a small code snippet that shows how these steps look like in code for the Laplacian):
> 1. *(Start)* The entry point is a function `f` that implements the forward pass.
>   It **can use arbitrary PyTorch functions**, as long as `f` remains trace-able with `torch.fx`.
>   `f` could be an MLP implemented as `torch.nn.Sequential`, but can also contain `torch.nn.functional`s **and is therefore general**.
> 2. *(Taylor mode)* To use Taylor mode, users can create the $K$-jet of `f` by applying the function transformation `jet` to `f` (implemented by our code).
>   **`jet` is extensible**: to add more operations, one has to implement their Taylor mode arithmetic according to the Faa-di-Bruno formula.
> 3. *(Vectorize)* The PDE operators we discuss require evaluating $K$-jets along multiple directions. Going from evaluating one to multiple $K$-jets is done with `vmap`.
> 4. *(Assemble PDE operator)* The vmap-ed $K$-jet can then be used to write a function that computes the desired PDE operator.
>    This function usually consists of three parts: (i) generate the ingoing Taylor coefficients
>    $\\{\mathbf{\mathbf{x}\_{0,r}}\\}$, $\\{\mathbf{\mathbf{x}\_{1,r}}\\}$, ..., along the directions $r$,
>    (ii) evaluate the $K$-jets along all directions using the vmap-ed $K$-jet, and (iii) combine the derivatives to obtain the final PDE operator.
> 5. *(Collapse)* The last step is to simplify the PDE operator function via graph rewrites.
>    For this, our code provides a `simplify` function, which analyzes the passed function's compute graph for rewrites that lead to collapsed coefficients.
>    **`simplify` is also extensible**: it relies on rules that require implementing (i) detecting a pattern, and (ii) specifying its substitution.
>
>
> Please let us know if this clarifies your concerns regarding generality and re-usability.
> We would be happy to follow up on this.
>
> ---
>
> > Explain why collapsed Taylor mode can lead to increased memory use for the biharmonic operator.
>
> Great question, thanks for asking!
>
> **Recap:** You are right that in our submission we found nested first-order AD to use less memory than vanilla/collapsed Taylor mode for computing the exact biharmonic operator (collapsing improved vanilla Taylor mode though, as expected).
>
> **Update:** We looked closer into this and identified an inefficiency in our implementation of Faa-di-Bruno (details below).
> **After fixing this inefficiency, collapsed Taylor mode now also uses less memory than nested first-order AD for the biharmonic operator.**
> Specifically, collapsed Taylor mode **went from 9.3 MiB (submission) to 5.9 MiB (now)** per datum, outperforming nested 1st-order AD (7.9 MiB).
> The inefficiency mostly affected higher derivatives, but we also got some small improvements for the Taylor mode Laplacians and weighted Laplacians.
>
> **Details:** In Faa-di-Bruno, we need to contract the derivative tensor against Taylor coefficients.
> Often, we contract multiple times against the *same* coefficient, but our submitted implementation did not leverage this.
> Think of multiplying multiple vectors `a * b * b * b`.
> We originally did this with `einsum(a, b, b, b)`, which does not exploit that `b` appears 3 times and treats each operand as independent.
> `einsum`'s pair-wise evaluation then leads to a lot of intermediate tensors being saved for backpropagation.
> Now, our code instead computes `a * b ** 3`, which reduces the number of saved intermediates and therefore memory, while also giving a small speed-up.
>
> ---
>
> > Maybe something like “the $K$-th derivative of $g \circ h$ is  $\partial g$ times the $K$-th derivative of $h$ plus other lower-order terms in $h$”?
>
> Thanks for this suggestion.
> We think your phrasing provides a nice complementary description of the math above Equation (6), which will help guide the reader, and will incorporate it.
>
> ---
>
> > In Equation (12), shouldn’t the two red sums be inside the inner product?
>
> The red sums are outside to highlight that Equation (12) has the same structure as Equation (5) for which we demonstrate in detail how the collapsing works.
> We did not spell out the collapsed version of Equation (12) due to space limitations and will add this lengthier version to the next revision.
>
> ---
>
> > In Figure 5, what are “non-differentiable computations”, and should “opaque” be replaced by “transparent”?
>
> Regarding opaque you are right, thanks, consider it fixed.
> What we mean by 'differentiable' versus 'non-differentiable' computations in code is the following:
> ```python
> with torch.no_grad(): # non-differentiable computation
>     laplacian = ... # no intermediate results need to be be stored
>
> with torch.enable_grad(): # differentiable computation
>     laplacian = ... # stores intermediates for later backpropagation
> ```
> Both modes of computation are relevant for applications.
> Let us know if this clarifies your question.
>
> Thanks again for your time and effort!

---

> > ### Comment · Reviewer_zJuU · 2025-08-04
> >
> > Thank you for your responses, I have no further questions.

---

### Official Review · Reviewer_rNn6 · 2025-06-30

**Clarity:** 4
**Significance:** 3
**Originality:** 3
**Rating:** 4
**Confidence:** 4

**Summary:**

The paper proposes a modification of the Taylor mode method [1] for propagating higher-order derivatives to enable more (compute and memory) efficient computation of linear PDE operators. They note that PDE operators can be expressed as sums of contractions of derivative tensors with some coefficient tensor and use this to modify what is propagated. Specifically, instead of propagating separate tuples of derivatives (K-jets) for each term of the sum they note that one can propagate the sum of the highest-order derivative in the K-jet instead which could lead to a memory usage saving. They apply this to a Laplacian operator as well as a biharmonic operator going into detail as to how their approach would work for these operators. Their results show that they obtain improved memory and computational efficiency when compared to vanilla Taylor mode and the more naive nested first-order AD.

Refs:
[1] Griewank, A. and Walther, A. Evaluating derivatives: principles and techniques of algorithmic differentiation. SIAM, 2008

**Questions:**

- There seems to be a significant typo in eqn (6) where $\bigotimes_{s \in \sigma} h_{K,r}$ should be $\bigotimes_{s \in \sigma} h_{s,r}$
- Adding an explanation/discussion for why collapsed Taylor mode has improved compute efficiency over vanilla mode would better help in understanding the full capabilities of their approach
- The weighted Laplacian seems to be just the same experiment as the Laplacian (lines 241-242: "For the weighted Laplacian’s coefficient matrix, we choose a full-rank diagonal matrix") and so it isn't adding new information about the collapsed Taylor mode. Using a dense matrix would probably be the ideal setting for this experiment.

**Ethical Concerns:**

["NO or VERY MINOR ethics concerns only"]

**Final Justification:**

So I have decided to leave my score as is because I feel that even though their approach can give compute and memory savings their lack of exploration of the applicability of the method makes the contributions of the paper not very significant. Specifically, they only apply it to Laplacian-type equations for moderately-sized dimensions which is a scenario of limited utility. Describing what use cases these computational savings unlock would definitely strengthen the paper

**Limitations:**

They have discussed limitations of their implementation but a wider discussion of the conceptual limitations of the collapsed Taylor mode is missing. Some suggestions:

- Discussion of the compute/memory tradeoffs for different kinds of linear PDE operators would make it clear to the reader as to whether they can expect their approach to give them any significant gains over vanilla Taylor mode.
- Related to the previous point, their experimental evaluation is a bit limited in that they only consider two linear PDEs that are very related (the biharmonic equation is just a Laplacian of a Laplacian) and use of the propagation of the sum of the highest-order Taylor coefficients (as highlighted in the "Weaknesses" above) isn't significantly utilized. Evaluating on a wider range of linear PDEs would give a clearer picture of their method.

**Quality:**

3

**Strengths And Weaknesses:**

Strengths:
- The explanation of their contribution is very thorough and clear and well elucidates the core ideas of their approach. Content in the appendix to aid in understanding of how their approach would work practically was very helpful. Also, the use of color coding and schematic diagrams are very useful additions to the paper which would otherwise make the details of the approach difficult to understand.
- Making the connection between the forms of linear PDEs and the Faà Di Bruno formulae to derive a more efficient way to compute the PDEs is a notable contribution and is something that ought to be of interest to those working on PINNs or more broadly interested in AD. In addition, this connection could make it easier for other researchers to derive other efficiencies for operators not considered in this work.
- The improved compute and memory performance over vanilla Taylor mode as shown in their results clearly demonstrates that their approach would be the preferable way to compute linear PDE operators.

Weaknesses:
- Their key contribution is a minor modification of vanilla Taylor mode and it doesn't seem like the key aspects of their approach i.e. propagating the sum of the highest-order Taylor coefficients instead of summing the separate propagations are used in their experiments. Specifically, for the Laplacian it is not clear as to what the difference in computation between standard and collapsed Taylor mode is (D15 and D16) other than the reduced memory saving that comes from storing a single vector instead of multiple. The same issue also looks to be present for the biharmonic equation as in their theory and experiments they don't seem to examine the case when non-zero high-order coefficients are propagated (they only consider the case when the zeroth and first-order coefficients are non-zero) which is when the sum of the highest-order Taylor coefficients would actually be used.

---

> ### Author Rebuttal · Authors · 2025-07-30
>
> Dear Reviewer rNn6,
>
> thanks for your support and for catching the typo in Equation (6)!
> Please find our responses to your questions below.
> Let us know if you have further questions; we would be happy to discuss.
>
> ---
>
> > for the Laplacian it is not clear as to what the difference in computation between standard and collapsed Taylor mode is
>
> > why collapsed Taylor mode has improved compute efficiency over vanilla mode
>
> Thanks for insisting on further clarification.
>
> The difference between standard and collapsed Taylor mode is that the collapsed version propagates fewer coefficients by computing the sum of the highest coefficients directly instead of propagating them separately, which reduces run time and memory.
>
> **Concrete example:** Let us illustrate this by looking at the *Laplacian propagation* at a point $\mathbf{x}_0$ through a single linear layer $\mathbf{h}_0 = \mathbf{W} \mathbf{x}_0 + \mathbf{b}$ with weight matrix $\mathbf{W}$ and bias $\mathbf{b}$:
>
> - **Commonalities:** Both vanilla and collapsed Taylor mode propagate the function value $\mathbf{h}\_0$ and the first derivatives $\mathbf{h}\_{1,r}$ along directions $r=1, \dots, R$ by computing
> \\begin{align}
> \mathbf{h}\_0 &= \mathbf{W} \mathbf{x}\_0 + \mathbf{b}\\,,
> \\\\
> \mathbf{h}\_{1,r} &= \mathbf{W} \mathbf{x}\_{1,r}\\,.
> \\end{align} If we neglect the cost to add the bias, this costs $1 + R$ matrix-vector multiplications with the weight matrix, and storing the $1+R$ result vectors.
> - **Difference:** For the second derivatives, vanilla Taylor mode propagates $R$ vectors
>     $$
>     \mathbf{h}\_{2,r} = \mathbf{W} \mathbf{x}\_{2,r}\\,.
>     $$ In contrast, collapsed Taylor mode propagates only a single vector
>     $$
>     \underbrace{\left(\sum\_r \mathbf{h}\_{2,r}\right)}\_{\mathbf{\tilde{h}}\_2}
>     = \mathbf{W} \underbrace{\left(\sum\_r \mathbf{x}\_{2,r}\right)}\_{\mathbf{\tilde{x}}\_2}\\,.
>     $$
>
> **From this example, we can see that collapsing reduces both run time and memory:**
> - **Vanilla:** Store $1 + 2R$ vectors, apply $1 + 2R$ matmuls.
> - **Collapsed:** Store $2 + R$ vectors, apply $2 + R$ matmuls.
>
> We will add this illustration to the appendix to improve the manuscript's clarity.
>
> ---
>
> > it doesn't seem like the key aspects of their approach i.e. propagating the sum of the highest-order Taylor coefficients instead of summing the separate propagations are used in their experiments
>
> **We do propagate the sums (the key aspect) in all our experiments.**
> This is reflected in the computational results:
> Take the linear layer example from above.
> For large $R$, the ratio of matmuls approaches $(2 + R) / (1 + 2R) \to 0.5$, and we thus expect the run time of collapsed Taylor mode to be half of vanilla Taylor mode.
> Our empirical results (Table 1) confirm that the time ratio is indeed close to $0.5$ for the Laplacian, and collapsed Taylor mode also uses less memory.
>
> ---
>
> > they don't seem to examine the case when non-zero high-order coefficients are propagated
>
> This is a fair point, because we limit the discussion to linear PDE operators that are 'pure' in the sense that they are **linear combinations of derivatives that are all of the same order**.
> This assumption leads to all propagated coefficients of order larger than 2 to be zero ($\mathbf{x}_{s\ge2, r} = \mathbf{0}$) such that each $K$-jet computes pure $K$-th order derivatives.
> As described in the paper, this covers a lot of practically relevant operators, like (weighted) Laplacians (e.g. Schrödinger and Fokker-Planck equations) and the biharmonic operator (e.g. plate equation).
>
> We agree that generalizing the class of PDE operators (say towards linear combinations of mixed-order derivatives) is a promising direction for future work.
> This may require some non-trivial intermediate steps though to formalize it mathematically.
>
> ---
>
> > The weighted Laplacian seems to be just the same experiment as the Laplacian [...] Using a dense matrix would probably be the ideal setting for this experiment.
>
> You are right that the weighted Laplacian we considered is exactly as expensive as the standard Laplacian.
> The relevant quantity that determines this cost is not the sparsity of the weightings $\mathbf{D}$, but their **rank** (Equation (8a)):
> - the standard Laplacian has $\mathbf{D} = \mathbf{I} = \operatorname{diag}(1,1,1,\dots)$ with $\operatorname{rank}(\mathbf{D}) = D$
> - our example weighted Laplacian has $\mathbf{D} = \operatorname{diag}((1,2,3,\dots))$ which also has $\operatorname{rank}(\mathbf{D}) = D$.
>
> Hence, they share the same cost.
> Note that we could have obtained the same result with any full-rank positive definite matrix $\mathbf{D}$ that need not be diagonal/sparse.
> While we believe this is an interesting result in itself (dense and sparse combinations of second derivatives can be done at the same cost), we agree with your criticism that the results are to some extent redundant.
>
> **Additional experiment:** To address this, we experimented with weighted Laplacians with rank-deficient weightings ($\operatorname{rank}(\mathbf{D}) < D$) and will add this to the appendix.
> Please find below a comparison between full-rank ($D$) and half-full rank ($D / 2$).
> Based on our theoretical analysis in the paper, we expect the performance ratio between vanilla and collapsed Taylor mode to be
> $(1 + 2 \operatorname{rank}(\mathbf{D})) / (2 + \operatorname{rank}(\mathbf{D})) \approx 2$.
> **As you can see in the table, going from full-rank to half-full rank indeed approximately halves run time and memory.** (Table notation: *full-rank vs. half-full rank*)
>
> | Implementation | Per-datum time [ms] | Per-datum mem [MiB] (differentiable) | Per-datum mem [MiB] (non-diff.) |
> |-|-|-|-|
> | Nested 1st-order | 0.62 vs. 0.32 | 4.4 vs. 2.3 | 2.2 vs. 1.1 |
> | Standard Taylor  | 0.60 vs. 0.30 | 5.5 vs. 2.8 | 1.2 vs. 0.61 |
> | Collapsed Taylor | **0.31 vs. 0.15** | **2.4 vs. 1.3** | **0.92 vs. 0.47** |
>
> We believe this addresses your concern regarding redundancy.
> Let us know if this makes sense to you.
> Thanks again for your time and effort!

---

> ### Comment · Reviewer_rNn6 · 2025-08-06
>
> Thanks for addressing my concerns.
>
> On the fact the propagation of sums isn't used in the Laplacian experiments, I do agree with your response but what I forgot to say was that the sum that is being propagated $\sum_{d=1}^{D} x_{2,d}$ is zero for the Laplacian case and so the vanilla and collapsed Taylor mode should essentially be the same other than the reduced memory footprint. I would assume that the improved run time performance is due to the fact that the vanilla mode is still doing the computation with $\sum_{d=1}^{D} x_{2,d}$ even though this should be optimized out given that it is zero.
>
> For the weighted Laplacian experiment, what I was trying to say is that if the weighted Laplacian $D$ is a diagonal matrix then the compute/memory result should essentially be the same as the Laplacian case (even though the numerical values may be different). The formulation given for the weighted Laplacian case allows for a dense $D$ and so it seems like their ought to be an experiment that examines this case i.e. when $\sigma$ are dense matrices themselves. However, as you said this may be redundant as the computations are effectively the same as the Laplacian (diagonal $D$) case. Also, the additional experiment seems redundant as it restates the results of the computational cost analysis in the paper. The memory/compute is halved for half-rank because there are now $R/2$ vectors being propagated instead of $R$.
>
> In light of the rebuttal I will leave my score as is.

---

> > ### Author Response · Authors · 2025-08-07
> >
> > Thank you again for your feedback and for taking the time to reconsider our submission.
> >
> > >so the vanilla and collapsed Taylor mode should essentially be the same other than the reduced memory footprint
> >
> > It appears that our explanation of the differences between collapsed and vanilla Taylor mode may have been unclear.
> >
> > After propagation through the first nonlinear operation, the second-order coefficients will be non-zero, resulting in the differences between vanilla and collapsed Taylor mode.
> >
> > To clarify, consider a two-layer network of the form $g \circ \tanh \circ h$, with $\sum_d x_{2, d} = 0$. In the first layer, we indeed have:
> > $$
> > \sum_d h_{2, d} = W_h \left(\sum_d x_{2, d} \right) = 0.
> > $$
> >
> > The second-order coefficients after the nonlinearity remain non-zero due to the presence of first-order terms. Specifically:
> > $$
> > \sum_d \tanh_{2, d} = \sum_d \langle \partial \tanh(h_0), h_{1,d} \otimes h_{1,d} \rangle + \langle \partial^2 \tanh(h_0), \sum_d h_{2,d} \rangle,
> > $$
> > and since $\sum_d h_{2, d} = 0$, this reduces to:
> > $$
> > \sum_d \tanh_{2, d} = \sum_d \langle \partial \tanh(h_0), h_{1,d} \otimes h_{1,d} \rangle.
> > $$
> >
> > Thus, in the next layer, we have:
> > $$
> > \sum_d g_{2, d} = W_g \left(\sum_d \tanh_{2, d} \right),
> > $$
> > which remains non-zero and must be computed.
> >
> >
> > We hope this clarifies the distinction between the two approaches. Please let us know if further explanation would be helpful — we are happy to elaborate further.
> >
> >
> > ---
> >
> >
> > > their ought to be an experiment that examines this case i.e. when $\sigma$ are dense matrices themselves.
> >
> >
> > The experiment we posted in the rebuttal reflects the situation where $\mathbf{D} = \mathbf{\sigma} \mathbf{\sigma}^\top$ with **dense** $\mathbf{\sigma}$.
> >
> > > the additional experiment seems redundant as it restates the results of the computational cost analysis in the paper.
> >
> > We think the experiment differs from the ones shown in the original submission in that it uses a rank-deficient coefficient tensor and *empirically confirms* it and hence provides additional evidence.
> >
> > Do you have a suggestion which experiment with weighted Laplacians is currently missing?
> > We would be happy to include that in a future version.

---

> > > ### Comment · Reviewer_rNn6 · 2025-08-07
> > >
> > > Your responses have addressed my concerns. I will leave the score as is.

---

### Decision · Program_Chairs · 2025-09-17

**Decision:**

Accept (poster)

**Comment:**

I recommend acceptance of this paper. All reviewers agree that the core idea, collapsing Taylor‐mode AD by rewriting the compute graph, leads to a clear practical benefits. In particular, reviewer rNn6 finds that "the explanation of their contribution is very thorough and clear and well elucidates the core ideas of their approach", reviewer zJuU finds that "the suggested approach is technically sound and insightful", reviewer GZnh finds that "the reported improvements are very clean, consistent and convincing". Even the more critical reviewers acknowledge the paper’s strengths. The paper makes a solid contribution.